# TEACHING ARITHMETIC TO SMALL TRANSFORMERS

**Nayoung Lee**[*]
University of Wisconsin-Madison
nayoung.lee@wisc.edu

**Kartik Sreenivasan**[*]
University of Wisconsin-Madison
ksreenivasa2@wisc.edu

**Jason D. Lee**
Princeton University
jasonlee@princeton.edu

**Kangwook Lee**
University of Wisconsin-Madison
kangwook.lee@wisc.edu

**Dimitris Papailiopoulos**
University of Wisconsin-Madison
dimitris@papail.io

## ABSTRACT

Large language models like GPT-4 exhibit emergent capabilities across general-purpose tasks, such as basic arithmetic, when trained on extensive text data, even though these tasks are not explicitly encoded by the unsupervised, next-token prediction objective. This study investigates how even small transformers, trained from random initialization, can efficiently learn arithmetic operations such as addition, multiplication, and elementary functions like square root, using the next-token prediction objective. We first demonstrate that conventional training data is not the most effective for arithmetic learning, and simple formatting changes can significantly improve accuracy. This leads to sharp transitions as a function of training data scale, which, in some cases, can be explained through connections to low-rank matrix completion. Building on prior work, we then train on chain-of-thought style data that includes intermediate step results. Even in the complete absence of pretraining, this approach significantly and simultaneously improves accuracy, sample complexity, and convergence speed. We also study the interplay between arithmetic and text data during training and examine the effects of few-shot prompting, pretraining, and parameter scaling. Additionally, we discuss the challenges associated with length generalization. Our work highlights the importance of high-quality, instructive data that considers the particular characteristics of the next-word prediction loss for rapidly eliciting arithmetic capabilities.[1]

## 1 INTRODUCTION

Large language models like GPT-3/4, PaLM, LaMDA (Brown et al., 2020; Chowdhery et al., 2022; Thoppilan et al., 2022) have demonstrated general-purpose properties, often referred to as *emergent abilities* (Wei et al., 2022a), for a wide range of downstream tasks like language and code translation, compositional reasoning, and basic arithmetic operations (Webb et al., 2022; Nye et al., 2021; Wei et al., 2022b; Shi et al., 2022; Wang et al., 2022; Srivastava et al., 2022; Chen et al., 2023). What is perhaps surprising, is that these tasks are not explicitly encoded in the model's training objective, which typically is an auto-regressive, next-token-prediction loss.

Prior research has delved into exploring these capabilities and how they emerge as the scale and of training compute, type of data, and model size vary (Wei et al., 2022a; Chung et al., 2022; Tay et al., 2022). Untangling the factors, however, remains challenging due to the data complexity and the variety of tasks examined. Driven by the curiosity to understand the factors that elicit these capabilities in next-token predictors, we set out to pinpoint the key contributors that accelerate the

---

[*]Authors contributed equally to this paper.
[1]Our code is available at https://github.com/lee-ny/teaching_arithmetic

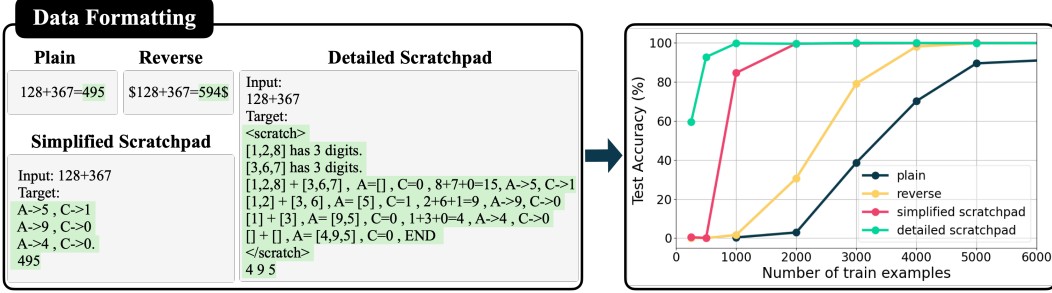

Figure 1: We investigate four data formatting approaches: **(i) Plain:** standard addition formatting (Section 4), **(ii) Reverse:** reversing the output (Section 4), **(iii) Simplified Scratchpad**: recording the digit-wise sum and carry-ons (Section 6), and **(iv) Detailed Scratchpad:** providing detailed intermediate steps (Section 6). We train small decoder-only transformers from scratch on addition data in these formats. The results (right) highlight the crucial role of data formatting in accuracy and sample efficiency. Plain never reaches 100% accuracy and the sample complexity for the remaining methods steadily improves with the level of details in the data format.

emergence of such abilities. These contributors may include the format and scale of data, model scale, the presence of pre-training, and the manner of prompting.

To provide a more precise examination of these factors, our study is conducted in a controlled setting: we first focus on teaching arithmetic to small decoder-only transformer models, such as NanoGPT and GPT-2, when trained from random initialization. Starting with a model of 10.6M parameters and scaling up to 124M parameters, we use the standard autoregressive next-token prediction loss. Our objective is to understand if and to what degree these models can efficiently learn basic arithmetic operations like addition, subtraction, multiplication, square root, and sine, thereby providing a clearer lens through which to view the elicitation of emergent abilities. Below, we summarize our findings.

**Data format and sampling plays a significant role.** We first observe that teaching a model addition (or any other operation) using standard addition samples, *i.e.*, '$A_3A_2A_1 + B_3B_2B_1 = C_3C_2C_1$', is suboptimal, as it requires the model to evaluate the most significant digit $C_3$ of the result first, which depends globally on all the digits of the two summands. By training on samples with reversed results, *i.e.*, '$A_3A_2A_1 + B_3B_2B_1 = C_1C_2C_3$', we enable the model to learn a simpler function, significantly improving sample complexity. Additionally, balanced sampling of different "variants" of addition, based on the number of carries and digits involved, further enhances learning. Even in this simple setting, we observe relatively sharp phase transitions from 0 to 100% accuracy as a function of the size of the training data. Although this may seem surprising, we observe that learning an addition map on $n$ digits from random samples is equivalent to completing a low-rank matrix. This connection allows us to offer a reasonable explanation for such phase transitions.

**Chain-of-thought data during training.** Building on these findings, we then explore the potential benefits of chain-of-thought (CoT) data during training. This format includes step-by-step operations and intermediate results, allowing the model to learn the individual components of compositional tasks. This format is directly borrowed from related literature, e.g., Ling et al. (2017); Wei et al. (2022b); Zhou et al. (2022a;b). We find that CoT-type training data significantly improved learning in terms of both sample complexity and accuracy in agreement with CoT fine-tuning literature (Nye et al., 2021; Chung et al., 2022), but *even in the complete absence of pretraining.* We conjecture that this is because breaking down the required compositional function to be learned into individual components allows the model to learn a higher-dimensional but easier-to-learn function map, in agreement with recent theoretical findings (Li et al., 2023; Malach, 2023). In Figure 1, we provide examples of the data formatting methods explored in our work.

**Training on text and arithmetic mixtures and the role of few-shot prompting.** We also explore the interplay between arithmetic and text data during training, as LLMs are trained on massive amounts of data scraped from the internet (Bubeck et al., 2023; Peterson et al., 2019), where it is impractical to carefully separate different types of data. We observe how the model's perplexity and accuracy vary with the ratio of text to arithmetic data. We find that jointly training on all the arithmetic operations discussed earlier can improve the individual performance of each task and that going from zero-shot to 1-shot prompting (showing one arithmetic example) yields a large accuracy improvement, but there is no significant improvement in accuracy by showing more examples.

**The role of pre-training and model scale.** We further investigate the role of pretraining by fine-tuning pretrained models like GPT-2 and GPT-3 (`davinci`) and observe that while the zero-shot

performance on arithmetic operations is poor, prior "skills" acquired during pretraining facilitate quick learning of some basic arithmetic tasks, even with a small number of finetuning samples. However, finetuning on non-standard data, such as those that result from reverse formatting, can interfere with the model's performance when pretrained, leading to decreased accuracy. We finally share our observations on how performance in arithmetic changes with scale, and although we find that scale does aid when finetuning for these tasks, it is not a necessary trait.

**Compositional and length generalization.** One might question if our trained models truly grasp arithmetic. Our findings present a nuanced answer. We find that length generalization beyond trained digit lengths is still challenging. For instance, if a model is trained on all $n$-digit lengths, excluding a specific length, it still struggles to accurately calculate this missing digit length. Consequently, the models achieve high accuracy within trained digit lengths but struggle significantly beyond this range. This suggests that the models learn arithmetic not as a flexible algorithm, but as a mapping function constrained to trained digit lengths. While this significantly surpasses memorization, it falls short of comprehensive arithmetic "understanding".

**Novelty over prior work.** Our approach heavily builds upon prior work that uses reasoning-augmented data to enhance model performance, and we do not purport originality in the types of training data used, nor in achieving the highest performance with the smallest model parameters possible. What sets our work apart is the primary focus on meticulously ablating our settings and extensive studies on various sampling techniques, training data formats, data source mixing ratios, and model scales. Our goal is to pinpoint the factors that contribute to the fast emergence of arithmetic capabilities. In the process, we also provide several straightforward yet novel and insightful theoretical explanations for some of the phase transition phenomena we observe. Our emphasis on arithmetic is not due to its intrinsic significance — one can easily delegate calculations to external tools (Schick et al., 2023; Gao et al., 2023). Instead, arithmetic serves as an emergent skill, easy to isolate and test, facilitating a more precise exploration of emergent phenomena.

## 2 RELATED WORKS

**Instructional data/chain-of-thought.** Detailed reasoning in training data has roots predating Transformers (Vaswani et al., 2017). Ling et al. (2017); Cobbe et al. (2021) use natural language to generate reasoning steps while Roy & Roth (2016); Reed & De Freitas (2015); Chen et al. (2017); Cai et al. (2017); Nye et al. (2021) show that symbolic reasoning may suffice. Nogueira et al. (2021) stress the importance of large number of small-digit samples (Yuan et al., 2023). Razeghi et al. (2022) observe a correlation between the frequency of numbers in the dataset and the performance involving them. In contrast, we find that transformers can learn to add numbers that were not seen during training. Chain-of-thought (Wei et al., 2022b) refers to the model's improved accuracy when prompted to produce intermediate reasoning steps. Zhou et al. (2022b) show that this can be achieved by providing sufficiently informative exemplars as a few-shot prompt (Brown et al., 2020). Zhou et al. (2022a) showed that *least-to-most* prompting can help GPT-3 solve problems decomposable into simpler sub-problems, by sequentially solving these subproblems. We extend this notion to simple addition and show that asking the model to output the least significant bit first has a similar effect.

**Arithmetic using Transformer models.** Our work focuses on decoder-only models as they are widely used in LLMs (Brown et al., 2020; Touvron et al., 2023; MosaicML, 2023). However, encoder-decoder models have also been extensively studied in the context of learning arithmetic operations (Kim et al., 2021; Wang et al., 2021). Wallace et al. (2019) on the other hand, focus on the impact of the learned embeddings. Ontanón et al. (2021) extensively study the problem of compositional generalization on benchmark datasets, such as SCAN (Lake & Baroni, 2018; Drozdov et al., 2022), and conclude that design choices, like relative position encoding (Shaw et al., 2018), can improve performance. Charton (2022; 2021) show that Transformers can learn linear algebra operations with carefully chosen encodings. Hanna et al. (2023) use mechanistic interpretability techniques to explain the limited numerical reasoning capabilities of GPT-2. Dziri et al. (2023); Jelassi et al. (2023); Yang et al. (2023) focus on the challenges of length generalization. A recent line of work explores finetuning techniques to improve arithmetic capabilities in pretrained models (Qian et al., 2022; Lightman et al., 2023; Uesato et al., 2022).

**Beyond Transformers.** While we focus our attention on GPT-like models, there is a rich literature studying other seq-to-seq models such as recurrent neural networks (RNNs) (Bowman, 2013; Bowman et al., 2014; Zaremba et al., 2014). Zaremba & Sutskever (2014) show that RNNs can learn how to

execute simple programs with for-loops provided they are trained with curriculum learning. Sutskever et al. (2014) show that LSTMs show improved performance on text-based tasks such as translation when the source sentences are reversed, which is closely related to what we observe in addition. Kaiser & Sutskever (2015) propose Neural GPUs which outperform prior RNNs on binary arithmetic tasks and even show length generalization *i.e.*, they can perform arithmetic on inputs of lengths that were unseen during training. This is yet to be seen even in modern pre-trained models (Bubeck et al., 2023) and therefore it is interesting to see if we can leverage some of these techniques and apply them to existing modern architectures. Dehghani et al. (2018) propose Universal Transformers (UTs) which introduce a recurrent transition function to apply recurrence over revisions of the vector representation at each position as opposed to the different positions in the input. They show that on the tasks from Zaremba & Sutskever (2014), UTs outperform traditional Transformers and RNNs.

## 3  PRELIMINARIES AND EXPERIMENTAL SETUP

In this section, we provide a detailed description of our experimental setup, including the model architecture and an overview of the various data formatting and sampling techniques used.

**Model and Data.**  To examine the individual factors at play, we use NanoGPT (Karpathy, 2022), a lightweight implementation of the GPT family of models. NanoGPT is a decoder-only transformer with six self-attention layers, six heads, and an embedding dimension of $384$, resulting in approximately 10.6M parameters. Unless stated otherwise, we use character-level tokenization and absolute position encoding. We train NanoGPT from random initialization, which we refer to as *training from scratch*, using the conventional next-token prediction objective. To study the effect of scale, we extend our experiments to GPT-2 and GPT-3 in Section 8. We investigate both training from scratch as well as fine-tuning using a pretrained GPT-2, whereas, for GPT-3, we only consider fine-tuning pretrained models. Refer to Appendix I for more details on the models and data used.

For arithmetic tasks like addition, subtraction, and multiplication, we define the training dataset for a binary operator $f(\cdot)$ as $\mathcal{D}_{\text{train}} = \{(a_i, b_i), y_i\}_{i=1}^{N}$ where $y_i = f(a_i, b_i)$. For unary operations like sine, the training dataset is formulated as $\mathcal{D}_{\text{train}} = \{a_i, y_i\}_{i=1}^{N}$, where $y_i = f(a_i)$. The test dataset $\mathcal{D}_{\text{test}}$ is constructed by randomly sampling pairs of operands not included in $\mathcal{D}_{\text{train}}$. We then apply different *data formatting* techniques on each data sample from the training dataset, creating the final sequence that serves as the model's input. Note that while we view $a_i$ as a single integer, the model will see it as a sequence of digits after character-level tokenization.

**Data Formatting.**  In the following sections, we will delve into the four data formatting approaches in our arithmetic experiments. See Figure 1 and Appendix J for examples. In Section 4, we explore the limitations of the conventional plain-format data and demonstrate how a simple reversal of the output order can lead to substantial performance improvements and enhanced sample efficiency. We introduce two Lemmas to support and explain these findings. Additionally, in Section 6, we present results on the simplified and detailed scratchpad formats, highlighting significant enhancements in sample efficiency for learning addition. We also emphasize the importance of carefully designing the intermediate steps in the detailed scratchpad method. Note that the scratchpad formats are largely adopted from the literature of chain-of-thought (CoT) training (Nye et al., 2021; Zhou et al., 2022b).

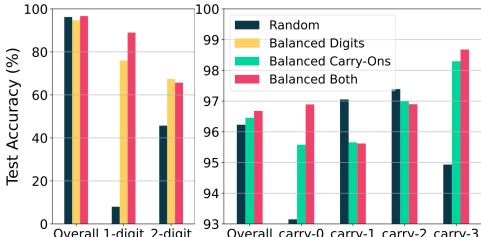

Figure 2: Performance of 3-digit addition on various data sampling methods used: **(i) Random**: uniform sampling of operands; **(ii) Balanced digits**: assigning higher sampling weights to operations involving 1 and 2-digit numbers; **(iii) Balanced carry**: balancing the dataset to contain an equal number of carry-on operations; **(iv) Balanced both**: balancing digits and carry-ons. We observe that *balanced data* improves accuracy compared to random sampling. Experiments on addition with the '$' symbol wrapped for each sample.

**Structured Data Sampling.**  While data formatting plays a crucial role, we also discover that choosing the samples carefully is also essential. When sampling operands for $n$-digit addition uniformly at random between 1 to $10^n - 1$, the dataset inevitably becomes highly skewed in terms of the number of samples with (i) operands containing a certain number of digits and (ii) operands

resulting in a certain number of *carry-on*[2] operations. For instance, in the case of 3-digit addition, random sampling results in a meager $0.01\%$ probability of selecting a 1-digit number. Additionally, 1 or 2 carry-on operations are more likely to occur than 0 or 3. To address this *imbalance*, we employ a structured sampling approach. Specifically, we aim to **(i) balance digits** by assigning higher weights to lower-digit numbers during the sampling process and **(ii) balance carry-ons** by ensuring an equal distribution of examples with $0, 1, \ldots, n$ carry-on operations. When sampling $10,000$ examples of 3-digit addition, we include all 100 1-digit additions, 900 2-digit samples and 9000 3-digit samples. Note that while the number of samples increases, the fraction of all possible $k-$digit additions that we sample for $k = 2, 3$ decreases due to the inherent skew. The split was chosen to ensure we saw a "reasonable" fraction of all possible $k-$digit samples for all $k$. Similarly, we ensure that the number of samples with $0, 1, 2,$ or $3$ carry-ons are all approximately 2500.

Figure 2 reveals the importance of balancing. We observe improvements in accuracy across the board while using *balanced* data when compared to random sampling. Further, random sampling performs relatively poorly even for the simple task of $2-$digit addition, possibly due to the fact that the model has not seen enough of these examples. For the remaining experiments, we set the default dataset for addition to be one that has both balanced digits and carry-ons.

## 4 DATA FORMAT CHALLENGES AND ARITHMETIC EMERGENCE

We start by examining *integer addition*. We first focus on 3-digit addition, *i.e.*, where the two summads have at most 3 digits ($\leq 999$). Later, in Section 7, we extend our findings to numbers with up to 10 digits. Surprisingly, teaching addition can be more complex than expected.

**Training on Conventional Data.** We start by training NanoGPT on standard addition data represented as '$A_3A_2A_1 + B_3B_2B_1 = C_3C_2C_1$', termed the *plain* format. However, as shown in Figure 1, this leads to fairly poor performance. We suspect that this is because the next-token prediction objective outputs the most significant digit (MSB) first. The following lemma clarifies the necessity to access all operand digits for outputting the MSB first.

**Lemma 1.** *Let $A$ and $B$ be two $n$-digit numbers, and let $C = A + B$. Suppose an algorithm $\mathcal{A}$ outputs the digits of $C$ in decreasing order of significance, then $\mathcal{A}$ must have access to all digits of $A$ and $B$ starting from the first digit that it outputs.*

The lemma suggests that to train a model for addition and to output the MSB first, it is necessary to emulate a *"global"* algorithm. Unlike the standard *"local"* algorithm for addition, which consists of computing digit-wise sums and carry-ons, approximating the global algorithm would require learning a more complicated function than necessary. The increased complexity results in decreased accuracy, as observed in our experiments. Liu et al. (2023) refer to this phenomenon as *attention glitches*.

**Reversing the Output.** We propose that the *reverse* format '$\$A_3A_2A_1 + B_3B_2B_1 = C_1C_2C_3\$$'[3] is more suitable for next-word prediction models. The rationale behind this is that when generating the sum by starting with the least significant digit (LSB), the model only needs to learn a local function of three inputs per digit – the two relevant digits of the operands and the carry-on from the previous digit. This local operation simplifies the function to be learned. The following lemma formalizes this:

**Lemma 2.** *There exists an algorithm that computes $C = A + B$ for two $n$-digit numbers $A$ and $B$ and outputs its digits in increasing order of significance such that, at each position $i$, the algorithm only requires access to the $i^{th}$ digits of $A$ and $B$, as well as the carry-on from the previous position.*

Lemma 2 directly follows from the *standard* algorithm for addition, which performs the sum and carry-on operations digit by digit. The implications of these lemmata are evident in our experiments when comparing the accuracy of the *plain* and *reverse* formats. As shown in Figure 1, training on reversed outputs significantly enhances accuracy, with considerably fewer samples. What is particularly remarkable is the *rapid emergence* of addition occurring between 1k to 4k samples for reverse. During that, the model rapidly transitions from being unable to add two numbers to being capable of perfectly adding. This leads us to ask:

---

[2]In this paper, we adopt the definition that a carry-on operation involves transferring information from one digit position to another position of higher significance. Therefore, we refer to the "borrow" operation in subtraction as a carry operation.

[3]We use '$\$$' symbol for data delimiter for the reverse format. Refer to Appendix B.1 for details.

*Why does addition rapidly emerge as the number of training examples increases?*

## 5 MATRIX COMPLETION: AN INCOMPLETE TALE OF EMERGENCE

Although the rapid transition observed in the previous section may initially seem surprising, closer examination reveals a fascinating equivalence – learning an addition map on $n$ digits from random samples can be considered as completing a rank-2 matrix. Establishing this connection with low-rank matrix completion (LRMC) provides meaningful insights into the observed phenomenon. However as we explain in this Section, this connection does not tell the complete story, and Transformers possess generalization capabilities far beyond what LRMC would predict.

**Learning addition tables is Matrix Completion.** Learning addition from samples $i + j$ can be formulated as a rank-2 Matrix Completion (MC) problem, where we partially observe an $n \times n$ matrix $\boldsymbol{M}$, whose $(i, j)$-th entry represents $i + j$. $\boldsymbol{M}$ can be decomposed into the sum of two rank-1 matrices, $\boldsymbol{N1}^T + \boldsymbol{1N}^T$, where $\boldsymbol{N}$ is a vector with entries $\{1, \ldots n\}$, and $\boldsymbol{1}$ is the vector of ones. Recovering a rank-2 matrix, in the absence of noise, can be sample-optimally performed by a simple iterative algorithm from Király et al. (2015) (Algorithm 1 in Appendix B.2). As depicted in Figure 3a, a *rapid transition* occurs at $\mathcal{O}(n)$, a well-known matrix recovery phenomenon (Recht, 2011).

We notice a similar rapid transition in NanoGPT. To investigate it, we focus on 2-digit addition (*i.e.*, $n = 100$) and evaluate the performance of learning addition through NanoGPT and LRMC (Figure 3a) by constructing train data as the revealed entries of the $\boldsymbol{M}$ matrix. Note that the dataset is no longer *balanced*, as the revealed entries are randomly sampled for LRMC experiments, to match the standard MC probabilistic settings Recht (2011). In Figure 3b, both NanoGPT and LRMC exhibit rapid transitions at approximately 1500 samples.

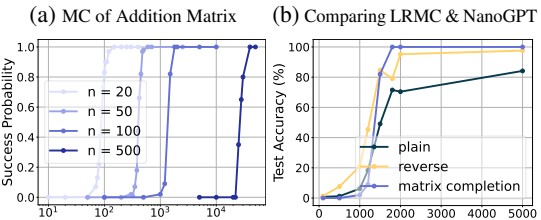

Figure 3: (a) We run Algorithm 1 (Király et al., 2015) on the addition matrix for $n = 20, 50, 100, 500$ and report the success probability while varying the number of revealed entries. As expected, a sharp transition occurs when approximately $\mathcal{O}(n)$ entries are revealed. (b) We compare the performance of NanoGPT trained on a dataset containing $n = 100$ samples (*i.e.*, 2-digit addition) to that of the corresponding LRMC problem using the same sample set. Remarkably, at $\approx 1500$ samples, both NanoGPT and Algorithm 1 begin learning addition almost flawlessly.

While the observed rapid transition can be attributed to the principles of LRMC, shedding light on the emergent arithmetic skill in NanoGPT, this connection falls short of capturing the full generalization capabilities displayed by NanoGPT.

**NanoGPT generalizes better than Matrix Completion solutions.** Upon further investigation, we find that NanoGPT exhibits capabilities beyond LRMC. Notably, LRMC is constrained by its inability to generalize in the presence of missing rows or columns. In our context, this equates to certain numbers being omitted from the training data. To assess NanoGPT's ability to overcome this, we deliberately exclude specific numbers or digits from our training data and assess the model's ability in learning addition. Can the model still generalize to unseen numbers?

As shown in Table 1, the answer to this question is a resounding *Yes!* The model achieves almost perfect accuracy even when excluding half of all possible $3-$digit numbers. NanoGPT can successfully learn 3-digit addition even when numbers or digits are intentionally excluded from the training data, thereby exhibiting generalization capabilities that far exceed what standard LRMC would predict.

Table 1: Impact of excluding numbers on addition task: NanoGPT models trained with $100/200/500$ excluded operands show no significant drop in accuracy and in some cases, the performance even improves.

| | No Exclusion | | Excluding 100 numbers | | Excluding 200 numbers | | Excluding 500 numbers | |
| --- | --- | --- | --- | --- | --- | --- | --- | --- |
| | Plain | Rev | Plain | Rev | Plain | Rev | Plain | Rev |
| Overall Acc. | 92.65%($\pm$2.53) | 99.87%($\pm$0.24) | 93.36%($\pm$2.62) | 99.82%($\pm$0.29) | 93.61%($\pm$2.77) | 99.78%($\pm$0.17) | 93.47%($\pm$3.22) | 100.0%($\pm$0.0) |
| Exclusion Acc. | - | - | 94.43%($\pm$1.70) | 99.87%($\pm$0.24) | 95.25%($\pm$1.60) | 99 88%($\pm$0.08) | 94.39%($\pm$2.05) | 100.0%($\pm$0.0) |

Specifically, we randomly choose $100/200/500$ numbers and exclude them from the training data. We then evaluate the trained models using two metrics: (i) *Overall accuracy:* which measures the

accuracy over a random set of $10,000$ examples and (ii) *Exclusion accuracy:* which measures the accuracy only over the excluded set (where either of the two operands is one of the excluded numbers). Remarkably, excluding numbers from the training data sometimes leads to improved performance. We conjecture that this may be due to a regularization effect, similar to random masking or cropping images in vision tasks. In Appendix B.2.1, we further find that NanoGPT models can even generalize to *unseen digits*, where a digit is absent from a particular ordinal position.

## 6 TRAINING ON CHAIN-OF-THOUGHT DATA EXPEDITES EMERGENCE

So far, we observed that simply reversing the output can result in remarkable performance, exceeding that of LRMC in learning addition. Here, we investigate if it is possible to expedite the emergence of addition by further enhancing the data format. As addition is a multi-step process, we explore the idea of incorporating additional information about each intermediate step. We adopt a CoT style approach, where we guide the model to learn addition step-by-step. We explore two levels of detail in the provided instruction steps, as shown in Figure1: **(i) Simplified Scratchpad** with minimal information – the sum and carry information for each digit/step. **(ii) Detailed Scratchpad** with comprehensive information on detailed traces of execution for each intermediate step.

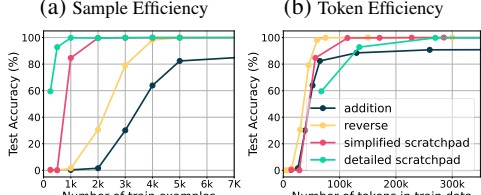

The results in Figure 4a show that the model trained on simplified scratchpad achieves $100\%$ accuracy with only 2000 samples, whereas reverse requires more than twice as many. Detailed scratchpad, which provides even more fine grained information, achieves perfect addition with just 1000 samples. This indicates a clear message: incorporating more information enables the model to learn addition with far fewer examples. We conjecture that this is because breaking down the required compositional function to be learned into individual, simpler components allows the model to learn a higher-dimensional but easier-to-learn function map, in agreement with recent theoretical work (Li et al., 2023; Malach, 2023).

We note that while CoT-style training enhances sample efficiency, it may not necessarily be the most "token-efficient" approach. To account for the cost associated with training and inference, we conduct a cost analysis based on the number of tokens (number of training samples $\times$ number of tokens per sample – see Appendix G for details) in the train data. encountered during training. The result in Figure 4b shows that reverse is the most efficient in terms of *token usage* for model training. The scratchpad methods, although more *sample-efficient*, require more tokens per sample.

Figure 4: (a) Comparison of sample efficiency: evaluating performance on training dataset with different numbers of addition samples. While all variants other than plain achieve $100\%$ accuracy, they differ in terms of sample complexity. (b) Number of tokens in train dataset required by NanoGPT to learn addition. Reverse is the most efficient in terms of token usage for model training, as the scratchpad methods, although more sample-efficient, require more tokens per sample.

In summary, incorporating scratchpad data and decomposing the addition task into steps offer a promising strategy to improve the performance and efficiency of small models in learning addition from scratch. Nevertheless, for practical usage, it is crucial to evaluate both the number of samples for achieving the desired performance and the actual token requirements during training and inference.

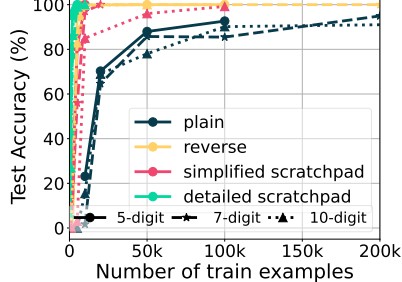

## 7 LONGER DIGITS AND BLENDING ARITHMETIC WITH SHAKESPEARE

In this section, we go beyond 3-digit addition to encompass a wider range of arithmetic tasks and longer digits to show that our insights on data sampling and formatting hold true even in this regime. We also explore the effect of mixing arithmetic with text data, and few-shot prompting.

Figure 5: Comparison of sample efficiency for 5, 7, and 10-digit additions. Training on plain requires an increasing number of samples for higher digits, while the sample complexity for other data formats remains relatively consistent.

**Extending to longer digit addition.** We repeat the experiment from Section 3 with up to 10 digit integers. Figure 5 shows that the behavior of all data formats remains similar across varying number of digits. In fact, the performance gap between the modified formats and plain grows with longer digits. While plain requires an increasing number of samples to learn higher-digit additions, the reverse and scratchpad formats maintain a consistent sample complexity. We also observe similar results in the *fine-tuning* setting, where we fine-tune a model initially trained on $k$-digits on $k+1$-digit data. See Appendix C for details on the experimental setup and additional results.

**Mixing Text with Arithmetic Data.** While the models so far were trained exclusively on arithmetic tasks, in practice, LLMs utilize a combination of arithmetic and *text* data for training. How does that affect the emergence of arithmetic skills? To explore that we incorporate both addition samples and text into our train data and evaluate the models with few-shot prompting (showing a few examples of addition in the prompt) to see if it is able to be effectively conditioned for the appropriate context (arithmetic/text generation). As we see in Figure 6, we find that few-shot prompting improves the performance of the model, allowing it to perform addition accurately even in the plain format.

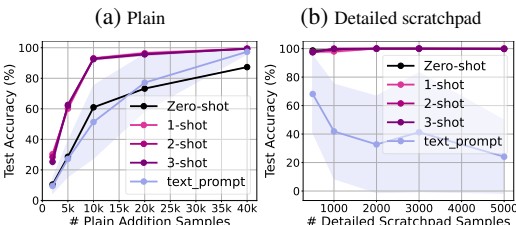

Figure 6: Performance of NanoGPT model trained with the Shakespeare dataset, addition dataset in plain, and detailed scratchpad format. The number of plain (left) and detailed scratchpad (right) formatted samples are varied. Performance is evaluated on zero-shot, few-shot, and text prompts, with the shaded area representing the standard deviation across various prompt exemplar sets.

Intriguingly, accuracy remains high using *plain* even with the inclusion of a text prompt preceding "A+B=". This is likely due to the structure of our mixed dataset where addition examples are interspersed within Shakespeare text. With the incorporation of more addition examples, instances where addition follows Shakespeare text increases, leading to a decrease in potential inconsistencies when text content is present during addition test queries. We further analyze the effect of text on prompting for both cases with and without text in the training data in Appendix E.

**Teaching arithmetic operations beyond addition.** We also note that the results on addition also hold for broader mathematical operations such as *subtraction*, *multiplication*, *sine*, and *square root* operations where each operation entails its unique challenges and intricacies. Refer to Appendix D for detailed settings and results.

## 8 FINE-TUNING, SCALING, AND PRETRAINING IN LARGER MODELS

We extend our study from NanoGPT to larger models like GPT-2 and GPT-3 to explore the impact of pretraining and model size. Initially, we compare the performance of NanoGPT and GPT-2, both trained from scratch. This highlights the advantages of larger model scales, especially in zero-shot scenarios. Subsequently, we delve into the impact of tokenization methods and model pretraining in GPT-2 models. We then fine-tune a pretrained GPT-3 on various arithmetic tasks using different data formats, reaffirming the importance of data formatting for larger pretrained models.

**Comparing NanoGPT and GPT-2: Tokenizer and Training Pretrained Models** We repeat our experiments on a GPT-2 model with 85M parameters, with twice as many layers, heads, and embedding size compared to NanoGPT. The transition to a GPT-2 setup necessitates several modifications. Firstly, we shift to OpenAI's Tiktoken BPE tokenizer. We also

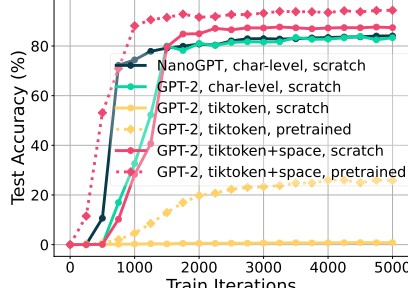

Figure 7: Performance of various configurations of the GPT-2 model on the addition task. We compare the effects of tokenization methods, specifically character-level tokenization versus Tiktoken (OpenAI's BPE tokenizer), training initialization (training from scratch versus training from a pretrained GPT-2 model), and the inclusion or exclusion of spaces between numbers. The results highlight the significance of utilizing pretrained models and incorporating spaces for consistent tokenization of numbers when training a model for arithmetic tasks.

examined two different training approaches: training the model from random initialization (scratch) and fine-tuning the pretrained model sourced from Huggingface. To circumvent potential inconsistent tokenization of numbers, alterations were made in data formatting to include spaces between numbers.

Figure 7 shows that GPT-2 demonstrates high performance in addition tasks with both character-level tokenization and Tiktoken with spaces between digits. This aligns with the results by Wallace et al. (2019), suggesting that character-level tokenization exhibits stronger numeracy capabilities compared to word or sub-word methods. Furthermore, comparing the models trained from scratch and the models trained from the pretrained model, we observe that fine-tuning a pretrained model results in better performance compared to training a model from scratch.

**GPT-3 experiments.** We consider three GPT-3 variants: `Ada`, `Curie`, and `Davinci` (OpenAI). We fine-tune these models using the same four data formatting methods as our NanoGPT experiments.

Table 2: Evaluation of addition performance for fine-tuned GPT-3 models: Davinci, Curie, and Ada. In each case, the model is finetuned on 1000 samples of addition in the corresponding format.

| GPT-3 Model | Zero-shot | Plain | Reverse | Simplified Scratchpad | Detailed Scratchpad |
|---|---|---|---|---|---|
| Davinci | 2% | 34% | 80.9% | 88.7% | **99.5%** |
| Curie | 0.0% | 1.4% | 12.3% | 10.7% | **99.7%** |
| Ada | 0.0% | 0.3% | 6.3% | 0.6% | **99.8%** |

The results in Table 2 show that starting with pretrained GPT-3 significantly improves performance compared to training NanoGPT or GPT-2 from scratch with only 1000 examples (Figure 4a). Similar to the result of training NanoGPT from scratch, the modified formats all outperform the plain format. Detailed scratchpad data achieves near-perfect accuracy, albeit with increased training and inference costs due to higher context length requirements. For our detailed experimental setup and further experiments on larger models, including fine-tuning GPT-3 refer to Appendix F.

## 9 LIMITATIONS

**Length generalization.** In our experiments, we did not observe any instances where the model could predict beyond the number of digits it had been trained on (see Appendix H). Shaw et al. (2018); Sun et al. (2022) reported similar difficulties and proposed approaches such as relative positional encodings. Anil et al. (2022) suggests that models can only perform out-of-distribution tasks by combining fine-tuning, prompting, and scratchpad techniques.

**Model/Data scale.** Due to the smaller scale of our experiments, we were able to thoroughly examine the impact of individual components on the model's arithmetic learning capabilities. Our model was limited to decoder-only architectures, primarily focusing on character-level tokenization. Although we have some preliminary results on scaling up and incorporating BPE-based tokenization, it is not clear if all our findings can be generalized to the scale of LLMs being used in practice.

**Beyond elementary arithmetic.** We choose to analyze simple arithmetic operations in order to carefully isolate factors that contribute to emergence. While the existing literature has already demonstrated the emergence of complicated abilities in practice, our work seeks to provide a better understanding of this behavior by extensive ablations in a controlled setting.

## 10 CONCLUSION

In this study, we investigate teaching arithmetic operations to small randomly initialized transformers using the next-token prediction objective. We carefully ablate different aspects of the training setting so as to isolate the factors that contribute to the emergence of arithmetic capabilities. Our results reveal that traditional training data is sub-optimal for learning arithmetic, and training on data with detailed intermediate steps or even simply reversing the output improves accuracy and sample complexity. We also study the effects of few-shot prompting, pretraining, and model scale. Despite improvements from detailed data, length generalization remains a challenge, highlighting the need for better-curated training data to ensure successful learning of specific algorithms as opposed to just learning an approximate function map. The correct approach for learning multiple arithmetic operations, of different levels of complexity, is still unclear. We anticipate that this research will contribute to a more nuanced understanding of the mechanisms by which transformers (approximately) acquire algorithmic skills.

ACKNOWLEDGMENTS

This work was supported by the Institute for Foundations of Data Science (IFDS), ONR Grant No. N00014-21-1-2806, N00014-23-1-2848, and a grant by FuriosaAI.

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

# Appendix

## Table of Contents

## A   PROOFS

Here, we present the proofs of Lemma 1 and 2.

**Lemma 1.** *Let $A$ and $B$ be two $n$-digit numbers, and let $C = A + B$. Suppose an algorithm $\mathcal{A}$ outputs the digits of $C$ in decreasing order of significance, then $\mathcal{A}$ must have access to all digits of $A$ and $B$ starting from the first digit that it outputs.*

*Proof.* We begin by assuming for contradiction that there does exist an algorithm `Algo` that does not have access to all digits of $A$ and $B$ and still outputs $C = A + B$ correctly for all $n-$ digit numbers $A, B$. Without loss of generality, say `Algo` does not have access to the $k-$th digit of $A$ where $k \in [n]$ represents the position counting from the least significant digit. Then consider the example $B = (10^n - 1)$ and $(A = 000 \ldots A_k 00 \ldots 0)$ where $B$ is just the integer with $n$ 9's and $A$ is just 0's with $A_k$ in the $k$th position. If $A_k = 0$, then $C_{n+1} = 0$, but if $A_k = 1$, then $C_{n+1} = 1$. Therefore, without access to the $k-$th digit of $A$, there exist examples where the algorithm will surely make a mistake. Therefore, by contradiction such an `Algo` cannot exist. $\square$

**Lemma 2.** *There exists an algorithm that computes $C = A + B$ for two $n$-digit numbers $A$ and $B$ and outputs its digits in increasing order of significance such that, at each position $i$, the algorithm only requires access to the $i^{th}$ digits of $A$ and $B$, as well as the carry-on from the previous position.*

*Proof.* First note that the trivial algorithm for addition is exactly the proof of this Lemma. However, we present a more formal argument below for completeness. Let $A$, $B$ be $n-$digit numbers and $C = A + B$ be at most an $(n + 1)$ digit number. Define the digits of $A, B,$ and $C$ as $A_i$, $B_i$, and $C_i$, respectively, for $i \in [n]$ counting from the least significant digit once again. Then, the addition can be performed using the following steps. First, $C_i = (A_i + B_i + carry_i) \mod 10$ where $carry_i$ is the carry-on from the addition of digits at position $i - 1$. If there is no carry from the previous position, then $carry_i = 0$. The carry for the next position is then calculated as $carry_{i+1} = \left\lfloor \frac{A_i + B_i + carry_i}{10} \right\rfloor$.

Putting this together, the algorithm for addition can be described as follows:

**Step 1:** Set $carry_1 = 0$. **Repeat** for $i = \{1, \ldots, n\}$: {**Step 2:** Compute $C_i = (A_i + B_i + carry_i) \mod 10$ and $carry_{i+1} = \left\lfloor \frac{A_i + B_i + carry_i}{10} \right\rfloor$, **Step 3:** Output $C_i$} **Step 4:** Output $C_{n+1} = carry_{n+1}$.

It is easy to see that this algorithm computes the digits of the sum $C$ correctly and requires only the individual digits at position $i$ and the carry from the previous position. Therefore, this algorithm satisfies the conditions of the lemma. $\square$

# B  ADDITIONAL EXPERIMENTS

## B.1  ZERO-PADDING, SYMBOL WRAPPING, AND TESTING

**Zero-padding and Symbol wrapping**  As discussed briefly in Section 3, we found a significant benefit to using padding for multi-digit addition. Throughout our experiments, we use the plain format without any such padding (denoted as "vanilla" below) as the default baseline representing the conventional data format used in training. Nonetheless, we explore modifications to this plain format to enhance performance; zero-padding, and wrapping with a single symbol. Zero-padding ensures a fixed length for operands and the output. In the case of 3-digit addition, this means 3-digit operands and a 4-digit output. For example, '112 + 29 = 141' becomes '112 + 029 = 0141'. As shown in Table 3. this modification significantly improves model performance. Next, we wrap each sample using the '$' symbol as in '$112 + 29 = 141$'. We found this performs on par with zero-padding.

As a result, we adopt the '$' symbol for efficient data delimiter, extending its use to the reverse format. Figure 8 shows '$'-wrapping also enhances the performance of the reverse format. Despite the plain format being improved with the '$' delimiter, it remains short of the reverse format's accuracy and sample efficiency. We continue to maintain the original plain format as a baseline since it not only exemplifies conventional data but further emphasizes the need for improved data formatting to ensure efficient training. As such, for the reverse format, we have incorporated the '$' delimiter in our formatting modifications.

Table 3: Test accuracy of NanoGPT model on 3-digit addition trained on 10, 000 samples of plain format data, comparing (i) vanilla format without modifications, (ii) Zero-padding format, and (iii) '$'-wrapped format. The results show significant performance enhancement through zero-padding for fixed length and similar improvements when deploying a single-symbol wrapping.

| Vanilla | Zero-pad | '$'-Wrapped |
| --- | --- | --- |
| 88.17% | 97.74% | 97.76% |

Figure 8: Performance of NanoGPT (left) and GPT-2 (right) models on 3-digit addition using plain and reverse format, both with and without '$' delimiter. The addition of the '$' symbol noticeably enhances performance in both formats. Nevertheless, the plain format underperforms compared to the reverse format, particularly in terms of sample efficiency. While we maintain the original plain format as a baseline – emphasizing the necessity for improved data formatting for efficient emergence – we incorporate the '$' wrapping in our modified reverse format.

## B.2 LOW-RANK MATRIX COMPLETION

In our Low-Rank Matrix Completion experiment for the addition matrix (which is of rank-2), we employ an iterative algorithm proposed by Király et al. (2015). This algorithm systematically searches for a $2 \times 2$ submatrix in which three entries are known and one entry is unknown. It then fills the unknown entry to ensure that the determinant of the $2 \times 2$ submatrix becomes zero, where the solution is known to be optimal. We present the full pseudo-code in Algorithm 1.

To assess the performance of the algorithm, we generate $n \times n$ addition matrices for various values of $n$ (e.g., 20, 50, 100, 500). We vary the number of revealed entries, randomly sampling a sparse matrix where only a specified number of entries between $n$ and $n \times n$ are known, while the remaining entries are set to zero. We repeat this process 100 times for each number of revealed entries, tracking the algorithm's success or failure in finding the solution. We calculate the average success rate across the trials and present the success probabilities in Figure 3a, where we observe a sharp phase transition when $\mathcal{O}(n)$ entries are observed, as expected.

---

**Algorithm 1:** Iterative $2 \times 2$ Matrix Completion Algorithm

---

**Data:** Data Matrix $M \in \mathbb{R}^{n \times n}$ with partially revealed entries. Assumed to be of Rank 2.
**Result:** $\widehat{M} \in \mathbb{R}^{n \times n}$, Success/Fail.

1   $n_1 \leftarrow 1$ represents number of resolved submatrices.
2   $n_2 \leftarrow 0$ represents number of unresolved submatrices.
3   $\widehat{M} \leftarrow M$
4   **while** $n_1 \geq 1$ **do**
     /* As long as we resolved at least one submatrix in the previous
     iteration      */
5     $n_1 \leftarrow 0$
6     $n_2 \leftarrow 0$
7     **for** $i = 1$ **to** $n$ **do**
8       **for** $j = 1$ **to** $n$ **do**
        /* do something      */
9
10        **if** $\widehat{M}_{i,j}$ *is not revealed **and** all its neighbors are revealed* **then**
11          $\widehat{M}_{i,j} = \frac{\widehat{M}_{i+1,j} \times \widehat{M}_{i,j+1}}{\widehat{M}_{i+1,j+1}}$
12          $n_1 \leftarrow n_1 + 1$
13        **if** $\widehat{M}_{i+1,j}$ *is not revealed **and** all its neighbors are revealed* **then**
14          $\widehat{M}_{i+1,j} = \frac{\widehat{M}_{i,j} \times \widehat{M}_{i+1,j+1}}{\widehat{M}_{i+1,j}}$
15          $n_1 \leftarrow n_1 + 1$
16        **if** $\widehat{M}_{i+1,j+1}$ *is not revealed **and** all its neighbors are revealed* **then**
17          $\widehat{M}_{i+1,j+1} = \frac{\widehat{M}_{i+1,j} \times \widehat{M}_{i,j+1}}{\widehat{M}_{i,j}}$
18          $n_1 \leftarrow n_1 + 1$
19        **if** $\widehat{M}_{i,j}, \widehat{M}_{i+1,j}, \widehat{M}_{i,j+1}, \widehat{M}_{i+1,j+1}$ *are all revealed* **then**
20          **continue**
21        **else**
22          $n_2 \leftarrow n_2 + 1$

23 **if** $n_2 > 0$ **then**
24     **return** $\widehat{M}$, Fail
25 **else**
26     **return** $\widehat{M}$, Success

---

### B.2.1 GENERALIZING TO UNSEEN DIGITS

Building upon the model's robustness to excluded numbers, we further investigate its ability to handle excluded digits, where a digit is absent from a particular ordinal position. Intuitively, this should

be even more challenging since excluding a digit means the model cannot learn directly how to operate in that position. Instead, it would have to generalize and infer that digits act similarly across all positions. We construct datasets with the number **5** excluded in 1st (LSB), 2nd, and 3rd (MSB) positions, and train separate models on each of these datasets. We compare the resulting models by evaluating **overall accuracy** on a test set of $10,000$ randomly sampled numbers, as well as their accuracy specifically on samples with **5** in each position which we call **exclusion accuracy**.

The results presented in Table 4 indicate that the model is not as robust to excluding digits compared to excluding numbers. However, it still achieves more than $66\%$ accuracy on every test and maintains an overall accuracy above $85\%$. Moreover, it appears that excluding a number in the least significant position yields the worst performance. This can be attributed to the fact that learning addition in this position is transferable to other positions since it is unaffected by carry-on operations. Failing to learn addition in this position, however, will have a detrimental impact on other positions as well.

Table 4: Impact of excluding digits on addition task: We investigate whether GPT-based models can infer addition on an excluded digit in a specific position from training data on other positions. We compare NanoGPT models trained with and without an excluded digit and find that excluding digits is harder to learn but not entirely impossible, with the worst performance observed when excluding the least significant digit.

| Excluded position | Input format | Overall Acc | "5" in the 1st (LSB) digit | "5" in the 2nd digit | "5" in the 3rd (MSB) digit |
|---|---|---|---|---|---|
| No exclusion | Plain | $92.65\%_{(\pm2.53)}$ | $92.58\%_{(\pm2.93)}$ | $93.59\%_{(\pm2.57)}$ | $95.16\%_{(\pm1.23)}$ |
| | Reverse | $99.87\%_{(\pm0.24)}$ | $99.89\%_{(\pm0.20)}$ | $99.95\%_{(\pm0.1)}$ | $99.97\%_{(\pm0.04)}$ |
| 1st (LSB) digit | Plain | $93.87\%_{(\pm1.64)}$ | $\mathbf{81.8\%_{(\pm7.01)}}$ | $94.04\%_{(\pm1.45)}$ | $94.22\%_{(\pm1.98)}$ |
| | Reverse | $98.00\%_{(\pm1.30)}$ | $\mathbf{90.32\%_{(\pm6.64)}}$ | $98.22\%_{(\pm1.27)}$ | $97.86\%_{(\pm1.54)}$ |
| 2nd digit | Plain | $92.53\%_{(\pm1.87)}$ | $90.46\%_{(\pm2.64)}$ | $\mathbf{83.87\%_{(\pm3.47)}}$ | $94.80\%_{(\pm0.90)}$ |
| | Reverse | $97.91\%_{(\pm1.47)}$ | $98.13\%_{(\pm1.28)}$ | $\mathbf{91.4\%_{(\pm6.13)}}$ | $98.93\%_{(\pm0.60)}$ |
| 3rd (MSB) digit | Plain | $90.85\%_{(\pm2.77)}$ | $89.23\%_{(\pm3.04)}$ | $90.94\%_{(\pm2.78)}$ | $\mathbf{85.88\%_{(\pm2.45)}}$ |
| | Reverse | $98.84\%_{(\pm0.47)}$ | $98.7\%_{(\pm0.54)}$ | $98.78\%_{(\pm0.59)}$ | $\mathbf{94.47\%_{(\pm1.99)}}$ |

## B.3 SUPPLEMENTARY EXPERIMENTS ON ADDITION

### B.3.1 COMMUTATIVITY IN ADDITION

We explore whether NanoGPT, trained with a plain data format on 10,000 training samples, captures the commutativity property of addition. This inquiry involves testing the equation $(A+B) = (B+A)$ across 9,900 instances.

As shown in Table 5, out of these 9,900 test cases, 8,799 samples exhibit commutative behavior. While a majority (8,700/8,799) of these instances are straightforward — the model correctly computes both $(A + B)$ and $(B + A)$ — it is noteworthy that among 164 cases where both outcomes are incorrect, 99 (60.3%) still preserve commutativity. This suggests a considerable, though imperfect, understanding of commutativity post-full training.

Table 5: Commutativity analysis of NanoGPT on 3-digit addition, examining the validity of $(A+B) = (B+A)$ for 9,900 test examples.

| $P(A + B = B + A$, both correct) | $P(A + B$ and $B + A$ both incorrect) | $P(A + B = B + A \mid$ both incorrect) |
|---|---|---|
| 0.878 (8,700 samples) | 0.016 (164 samples) | 0.603 (99/164 samples) |

### B.3.2 ADDITION INVOLVING SIGNED NUMBERS

While our primary focus has been on adding positive numbers, we posit that these results can extend to signed numbers, which effectively encompass both addition and subtraction operations. We conduct a preliminary experiment, training the model on various signed combinations of two 3-digit numbers.

Figure 9 shows that training on signed numbers presents a more challenging task compared to training on addition or subtraction on only positive numbers. Notably, test accuracy for addition involving same-sign operands, $(+, +)$ and $(-, -)$, surpass those involving opposite-sign operands, $(+, -)$ and

$(-, +)$. This observation aligns with the expectation that learning addition is relatively simpler than learning subtraction.

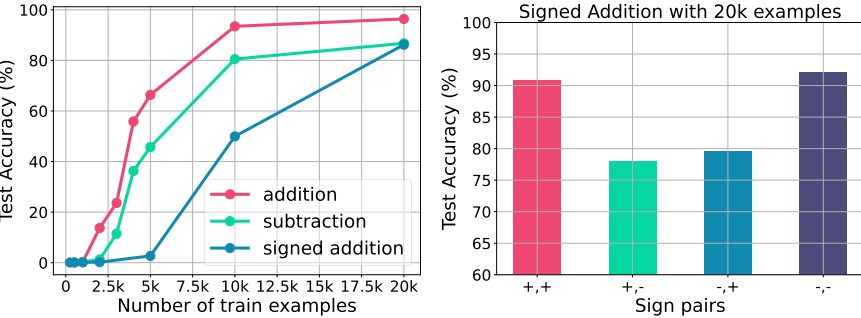

Figure 9: Performance on training NanoGPT on addition of signed numbers. (Left) Training on addition and subtraction tasks using only positive numbers up to 3 digits, contrasted with the signed addition task involving various sign combinations. (Right) Performance of the model trained on signed addition with 20,000 training examples, across different sign combinations in the test data.

### B.3.3 PERFORMANCE BASED ON DIFFERENT DIGIT LENGTHS

We analyze the performance of NanoGPT models by segregating the test accuracy based on the digit count of operands. In Table 6 and Table 7, rows and columns correspond to the digit count in the first and second operands, respectively.

Notably, with the plain format, performance decreases with fewer digits in the operands, and an asymmetry based on operand lengths is observed. We speculate that this asymmetry arises because, despite efforts to ensure balanced sampling, combinations of small digits, such as $(1, 3)$-digit or $(2, 1)$-digit pairs, are underrepresented in our training dataset.

Table 6: Performance by digit length in plain format addition.

|         | 1-digit | 2-digit | 3-digit |
|---------|---------|---------|---------|
| 1-digit | 72.8%   | 42.2%   | 77.0%   |
| 2-digit | 27.8%   | 60.0%   | 91.4%   |
| 3-digit | 38.6%   | 61.2%   | 97.0%   |

In contrast, the reverse format result exhibits notably consistent performance across various digit combinations, indicating near-perfect learning of addition by the model.

Table 7: Performance by digit length in reverse format addition.

|         | 1-digit | 2-digit | 3-digit |
|---------|---------|---------|---------|
| 1-digit | 100.0%  | 99.8%   | 99.0%   |
| 2-digit | 96.4%   | 100.0%  | 100.0%  |
| 3-digit | 94.0%   | 99.4%   | 100.0%  |

### B.4 THE IMPORTANCE OF INTERMEDIATE STEP DESIGN

In this section, we underscore the significance of *meticulously designing the intermediate steps in a Chain-of-Thought manner*. Specifically, we investigate whether the enhanced sample efficiency of NanoGPT in detailed scratchpad format arises from its longer length or from the breakdown of intermediate steps into simpler components.

**Randomizing the intermediate steps**     To discern the impact of length, we modify the intermediate steps, replacing them with either a uniform token "#" or random tokens within the vocabulary (see examples in Figure 10).

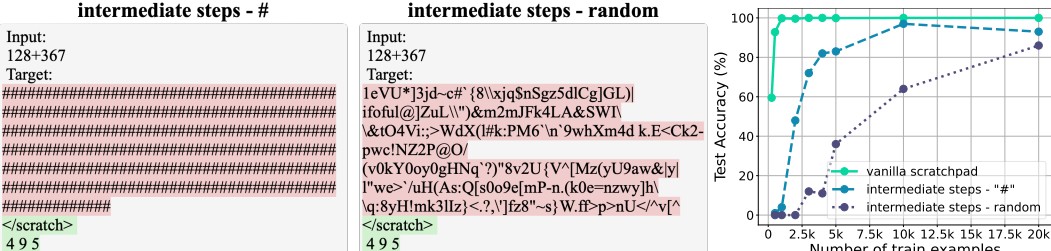

Figure 10: Replacing the intermediate steps with random characters. We ablate whether the sample efficiency gain from the detailed scratchpad stems from the increased length in the format.

The results (Figure 10) indicate that the sample efficiency of the vanilla detailed scratchpad surpasses the modified versions. This suggests that the advantage of detailed scratchpad format stems from the breakdown into simpler functions rather than its increased length.

**Subtraction - two different intermediate step design**     We further focus on the **subtraction** task and conduct experiments to compare two different versions of the detailed scratchpad for this operation (see examples in Figure 11). These trials shed light on the importance of decomposing the subtraction task into *simpler intermediate steps*. Unlike addition, subtraction behaves differently depending on whether the first operand ($a$) is greater than the second operand ($b$) or vice versa.

The first strategy (Version 1 in Figure 11) involves performing digit-wise subtraction starting from the least significant bit (LSB) and considering borrows when necessary. However, this strategy produces incorrect results when the first operand is smaller than the second operand. In such cases, we subtract the number in the most significant bit (MSB) position multiplied by 10 to the power of (number of digits in the output - 1) from the remaining digits in the output. An example illustrating this approach is shown in Version 1, Case 2. Alternatively, we can adopt a more familiar strategy. If the first operand is smaller than the second, we swap the operands and compute the negation of the subtraction of the swapped operands: $a - b = -(b - a)$ (referred to as Version 2).

The results in Figure 12 indicate that Version 2, which involves comparing two operands, performs considerably worse than Version 1. In Version 1, each intermediate step only requires the simpler 1-digit subtraction, along with addition in the final result processing step. Upon analyzing the failure cases of Version 2, we observe that the majority of errors stem from incorrectly identifying which of the two operands is larger, while the intermediate steps are handled correctly. This finding underscores the significance of breaking down arithmetic operations into *simpler intermediate steps*. Unless otherwise specified, we use Version 1 in all detailed scratchpad experiments.

---

**Detailed scratchpad formatting for different arithmetic tasks**

Examples of two variations of detailed scratchpad formatting for subtraction, considering the scenario where the first operand $a$ is greater than the second operand $b$, and vice versa. In Version 1, a result processing step is included in the final stage to handle negative outputs. In Version 2, the operands are compared at the beginning, and if $b$ is larger, their order is reversed.

**Prompt (Case 1.** $a - b \geq 0$ **):**
```
Input:
367-128
Target:
```

Version 1.

```
...
<scratch>
[3,6,7] has 3 digits.
[1,2,8] has 3 digits.
[3,6,7] - [1,2,8] , A=[] , C=0 , 7-8-0+10=9 ,
 A->9 , C->-1
[3,6] - [1,2] , A=[9] , C=-1 , 6-2-1=3 , A->3
 , C->0
[3] - [1] , A=[3,9] , C=0 , 3-1-0=2 , A->2 ,
C->0
[] - [] , A=[2,3,9]
200+39=239 , END # result processing
</scratch>
2 3 9
```

Version 2.

```
...
<scratch>
[3,6,7] has 3 digits.
[1,2,8] has 3 digits.
367>=128 # comparison of two operands
[3,6,7] - [1,2,8] , A=[] , C=0 , 7-8-0+10=9 ,
A->9 , C->-1
[3,6] - [1,2] , A=[9] , C=-1 , 6-2-1=3 , A->3
 , C->0
[3] - [1] , A=[3,9] , C=0 , 3-1-0=2 , A->2 ,
C->0
[] - [] , A=[2,3,9] , END
</scratch>
2 3 9
```

**Prompt (Case 2.** $a - b < 0$ **):**
```
Input:
128-367
Target:
```

Version 1.

```
...
<scratch>
[1,2,8] has 3 digits.
[3,6,7] has 3 digits.
[1,2,8] - [3,6,7] , A=[] , C=0 , 8-7-0=1 , A
->1 , C->0
[1,2] - [3,6] , A=[1] , C=0 , 2-6-0+10=6 , A
->6 , C->-1
[1] - [3] , A=[6,1] , C=-1 , 1-3-1=-3 , A->-3
 , C->-1
[] - [] , A=[-3,6,1]
-300+61=-239 , END # result processing
</scratch>
-2 3 9
```

Version 2.

```
...
<scratch>
[1,2,8] has 3 digits.
[3,6,7] has 3 digits.
128<367 : 128-367=-(367-128) # comparison
[3,6,7] - [1,2,8] , A=[] , C=0 , 7-8-0+10=9 ,
 A->9 , C->-1
[3,6] - [1,2] , A=[9] , C=-1 , 6-2-1=3 , A->3
 , C->0
[3] - [1] , A=[3,9] , C=0 , 3-1-0=2 , A->2 ,
C->0
[] - [] , A=[2,3,9] , END
</scratch>
-2 3 9
```

Figure 11: Two versions of detailed scratchpad formatting for subtraction.

### B.5 THE EFFECT OF NOISY INPUTS ON ACCURACY

**Noisy intermediate steps in the scratchpad data.** We further investigate the significance of providing accurate intermediate steps in the scratchpad during the training process. While this was inspired by the findings of Min et al. (2022), it is inherently different. Min et al. (2022) show that using random labels in ICL demonstrations caused minimal degradation when compared to the gold labels. However, those models were trained on gold labels and then evaluated on multiple downstream tasks. In our setting, the model is trained and evaluated on a single arithmetic task. Further, the final result(or label) is left untouched as the correct answer to the arithmetic operation. We only replace the intermediate steps. The goal of this study is to verify whether the model actually learns to reason using the given intermediate steps or merely uses the scratchpad to improve its expressivity. We compare the performance of training with our simplified scratchpad formatting, which includes accurate $A$ (digit sum) and $C$ (carry) information, with formatting that includes random $A$, random $C$, or random $A$ and $C$ for each intermediate step, as depicted in Figure 1.

The results in Figure 13, demonstrate that the inclusion of noisy labels can impede sample efficiency. However, with enough samples, the model ultimately achieves full accuracy. This suggests that while

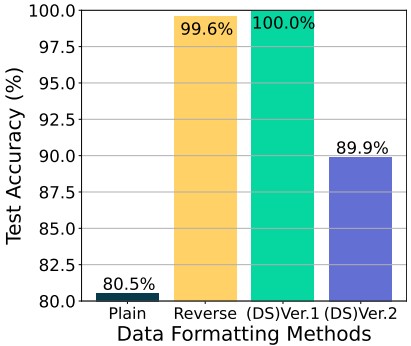

Figure 12: Comparison of performance among various data formatting approaches (plain, reverse, and two versions of detailed scratchpad (DS)) for the subtraction task. The experiments were conducted on a NanoGPT model trained on a dataset of 10,000 examples. Version 2, which incorporates operand comparison, exhibits significantly lower performance compared to Version 1. This observation highlights the substantial impact of the construction of intermediate steps on the model's performance.

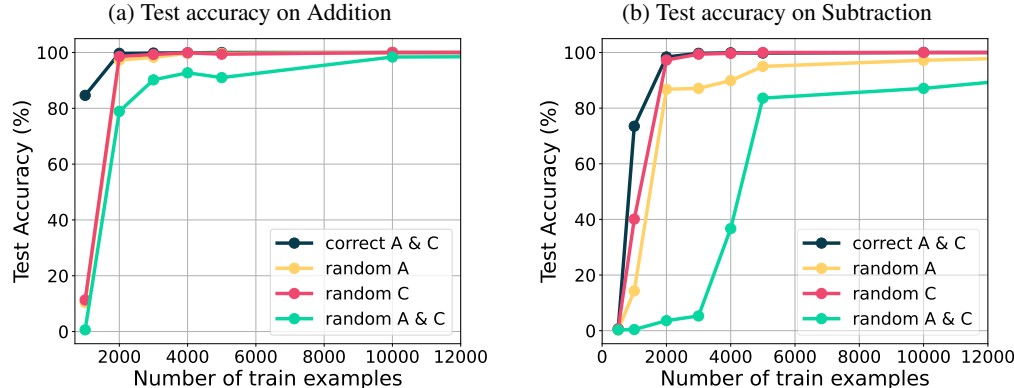

Figure 13: Comparison of training with simplified scratchpad formatting using correct A and C information with formatting using random A/C and their effect on sample efficiency and accuracy. Results show that noisy labels degrade sample efficiency, but with sufficient training data, the model eventually reaches full accuracy.

the model is capable of leveraging the information contained in the intermediate steps, it can also gradually learn how to perform addition while disregarding the presence of noisy intermediate steps.

**Model robustness to noise in the auto-regressive output.** In this analysis, we explore the robustness of models trained on plain or reverse formatted data (without noise) when exposed to noise during an auto-regressive generation process. In particular, we aim to unravel how much the learned mapping of the $i$-th output relies on the operands and preceding tokens in the addition result, given that transformer models generate tokens sequentially in an autoregressive manner, making them prone to error propagation.

For this experiment, we focus on 3-digit addition. We train models on either plain or reverse format data and evaluate the accuracy of next-token predictions when the output sequence contains noise. Specifically, in the plain format setting, we expect a well-performing model to generate the correct output tokens $O_3, O_2, O_1$ sequentially, where $O_3 = C_3$, $O_2 = C_2$, $O_1 = C_1$, and $C_3C_2C_1$ represents the correct answer. We consider two types of perturbation: (i) **random** perturbation, where we modify the first two output tokens $O_3O_2$ to random numbers different from $C_3C_2$, and (ii) **precise** perturbation, where we perturb only the second output token $O_2$ by 1. The second case is particularly relevant since a common error case is where the model misses a digit by 1. We provide the model with an expression of the form "$A_3A_2A_1 + B_3B_2B_1 = O_3O_2$", where $O_3O_2$ can be either (i) a random incorrect number, $i.e.$, $O_3O_2 \neq C_3C_2$, or (ii) $O_2 = C_2 \pm 1 \mod 10$, and observe the next token generated by the model. A corresponding process is deployed for the reverse format, introducing a noisy sequence to models trained on reverse format data.

To evaluate the performance, we define two accuracy criteria for $O_1$: **exact accuracy**, reckoning $O_1$ as accurate only when $O_1 = C_1$, and **relaxed accuracy**, considering $O_1$ correct if it deviates from the original output $C_1$ by at most 1. In other words, $C_1 = O_1$, $C_1 = O_1 + 1 \mod 10$ or $C_1 = O_1 - 1 \mod 10$.

Table 8: Prediction accuracy for the third digit output under different types of noise in the preceding output tokens. **Random** perturbation, applies random flips whereas **precise** perturbation shifts the preceding output tokens by 1. **Relaxed accuracy**, allows for a $\pm 1$ deviation from the true output whereas **Exact accuracy** is strict. Reverse consistently outputs a number that is at most 1 different from the true output, even in the presence of noise. The plain format has high exact accuracy in the presence of precise perturbation, as the noise in the output token has a lower impact on predicting the next token, which is of lower significance. However, with completely random noise, the plain format shows poor performance, suggesting a strong dependence on all digits. (See Lemma 1 and 2).

| Perturbation Type | Random | | Precise | |
|---|---|---|---|---|
| | Plain | Reverse | Plain | Reverse |
| Exact Acc | 49.88% | **81.26%** | **99.85%** | 90.47% |
| Relaxed Acc | 61.55% | **100%** | **100%** | **100%** |

The results presented in Table 8 reveal intriguing findings. We observe that the reverse format consistently outputs a result that *deviates by no more than 1 from the true answer*, regardless of whether the preceding outputs $O_3O_2$ are subjected to random or precise perturbation. This consistency can be explained by Lemma 2, indicating that the reverse format only requires learning a straightforward function of digit-wise addition for each corresponding position, along with the carry-on (0 or 1). Therefore, even with noise in the preceding tokens, the model accurately performs digit-wise addition, albeit with occasional carry-on prediction errors. With an exact accuracy of 81.26% even in the presence of random perturbation, the reverse format demonstrates the model's ability to rely less on the preceding output tokens, indicating a robust learned output mapping.

On the contrary, models using the plain format have to decipher a more intricate function drawing from all digits within the sequence, as described by Lemma 1. Given that in addition, carry operations transition from right to left (*i.e.*, least to most significant digit), the introduction of precise perturbation on preceding output tokens, which possess higher significance, has a minor impact on the output (which has less significance). As a result, models trained using the plain format attain an exact accuracy rate of 99.85% and a relaxed accuracy of 100% for cases involving precise perturbation. Interestingly, under purely random perturbation, the plain format struggles, leading to a reduced relaxed accuracy of 61.55% and exact accuracy of 49.88%. This suggests that the output mapping learned by the plain format is not merely a function of the two operands but rather enmeshed in complex dependencies on preceding output tokens.

### B.6 ANALYZING THE RESULTS ON SINE/SQRT

Since sine and sqrt are arguably more complicated functions than the remaining arithmetic tasks, we decided to more carefully analyze their performance. As shown in Figure 14, sin shows excellent performance across all data formats around $\sin(x) = 0$. We conjecture that this is because $\sin(x) \approx x$ for $x \approx 0$, which is easy to learn. We also note that accuracy once again improves close to $\pm 1$ potentially for similar reasons.

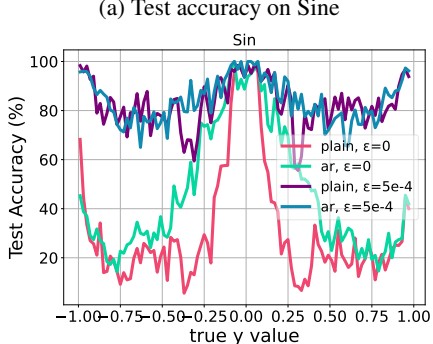
(a) Test accuracy on Sine

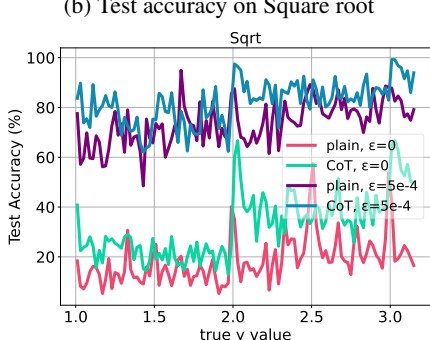
(b) Test accuracy on Square root

Figure 14: Error analysis of sine and square root functions, considering varying error tolerance (eps) thresholds to determine correct output. The sine function demonstrates excellent performance across all data formats, particularly around $\sin(x) = 0$, where $\sin(x) \approx x$ for $x \approx 0$. Additionally, we observe improved accuracy near $\pm 1$.

## C  EXTENDING TO LONGER DIGIT ADDITION

In this section, we extend our experiments beyond 3-digit addition and explore longer-digit settings, ranging up to 10 digits. Our aim is to investigate whether our previous findings regarding the sample efficiency of reverse and scratchpad formats hold true for larger numbers of digits.

We begin by observing that the phase transition behavior observed in previous sections also applies to longer-digit addition. Furthermore, we discover that the advantages of using reverse and scratchpad formats become even more pronounced as the number of digits increases. Next, we examine the number of training samples required to learn $k + 1$ digit addition when fine-tuning a pretrained model trained on $k$ digit addition. We find that while the number of samples needed to further learn $k + 1$ digit addition remains relatively consistent for reverse and scratchpad formats, the plain format requires an increasing number of samples.

**Experimental setup and data generation.**    To explore the performance of the model in higher-digit addition scenarios, we extend the experimental setup described in Section 3. We adopt a balanced sampling approach for training data with $D$ digits, ensuring an equal number $d$ of all combinations of digits for both operands as follows:

We begin by sampling all 100-digit additions. For the remaining number of digits, ranging from 2 to $D$, we generate addition examples of the form "A + B = C". The two operands, A and B, are randomly sampled $d = \lfloor (N - 100)/(D(D + 1)/2 - 1) \rfloor$ times for every $D$, where $N$ is the total number of training examples. Operand A is sampled between $[10^{k_1-1}, 10^{k_1} - 1]$ and operand B is sampled between $[10^{k_2-1}, 10^{k_2} - 1]$, for all $1 \leq k_1 \leq k_2 \leq D$, excluding the case where $k_1 = k_2 = 1$. After sampling the two operands, we randomly interchange them to cover cases where A has fewer digits than B and vice versa.

### C.1  TRAINING FROM RANDOM INITIALIZATION

We repeat the experiment from Section 3 on nanoGPT with longer digits. The results shown in Figure 15 demonstrate a similar behavior to the findings observed in Figure 4a for 3-digit addition. This indicates that our previous observations generalize to longer sequence lengths. Notably, the performance gap between the modified formats (reverse, simplified scratchpad, and detailed scratchpad) and the plain format becomes even more significant in the context of higher digits. While the plain format requires an increasing number of training examples to learn higher-digit additions, the reverse or scratchpad formats exhibit a more consistent requirement in terms of the number of training examples.

This prompts us to explore the differences between each format in a fine-tuning setting. Specifically, we ask whether a model trained on reverse or scratchpad-formatted $k$ digit addition data would find it easier to learn $k + 1$ digit addition compared to a model trained with plain format addition.

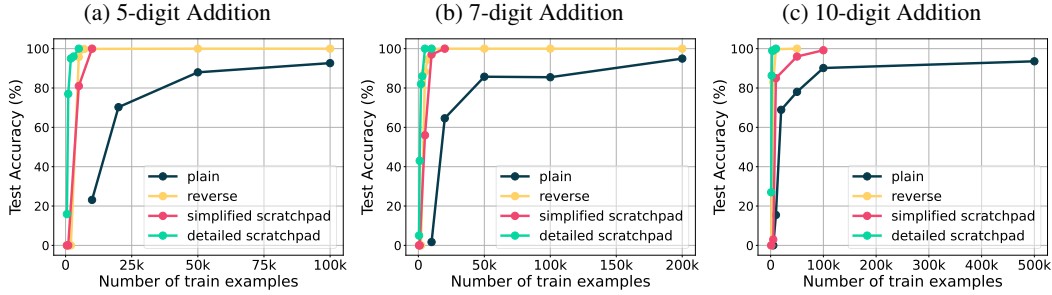

Figure 15: Comparison of sample efficiency for 5, 7 and 10-digit additions: performance of models trained with varying numbers of addition samples on each data format. The plain format data requires an increasing number of training examples for higher digits, while the number of samples required for other methods remains relatively consistent.

## C.2 FINE-TUNING FROM PRETRAINED MODELS

In this section, we investigate the generalization ability of transformer models, specifically focusing on their capacity to learn higher-digit additions based on their knowledge of lower-digit additions. Additionally, we explore how the choice of data format affects the number of samples required to learn higher-digit additions.

**Forgetting of $k$-digit addition when trained on $k + 1$-digit addition.**

We begin by fine-tuning a model that was initially trained on 3-digit addition. We fine-tune this model using 4-digit addition training data, with each data format being used separately. To mitigate the "catastrophic forgetting" phenomenon, we experiment with different learning rates, gradually reducing the magnitude. We continue this process until the learning rate becomes too small for the model to effectively learn 4-digit addition.

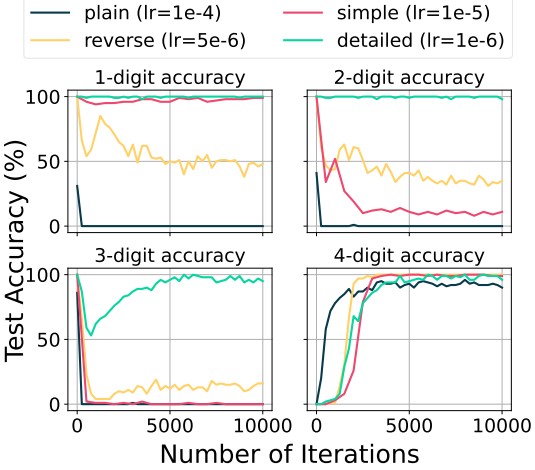

Figure 16: Accuracy of 1 to 4-digit additions during fine-tuning of a pretrained model on 3-digit additions using different data formats. The model is fine-tuned using only 4-digit addition data with corresponding formats. We observe that the plain format 'forgets' 1 to 3-digit additions entirely when learning 4-digit addition. In contrast, the detailed scratchpad method successfully learns 4-digit addition while maintaining high performance on 1 to 3-digit additions.

The results depicted in Figure 16 reveal interesting insights about the fine-tuning process. When training the model using the plain format with only 4-digit addition data, there is an immediate drop in accuracy for 1 to 3 digit additions. This indicates that the model experiences significant forgetting of previously learned additions. In contrast, the reverse and scratchpad methods exhibit a more favorable behavior. The model trained with these methods does not completely forget 1 or 2 digit additions while learning 4-digit addition. Remarkably, the detailed scratchpad method stands out by enabling the model to learn 4-digit addition without compromising its performance on 1 to 3 digit

additions. Although there is a slight decrease in performance for 3-digit additions initially, the model quickly recovers and picks up the knowledge again as it trains on 4-digit additions.

This result can be explained by the hypothesis that learning a $k + 1$ digit addition from a $k$-digit model is an incremental process for the detailed scratchpad method. The model already has a solid foundation in understanding the intermediate steps involved in addition, so it only needs to adapt to longer sequences. In contrast, for the plain format, learning higher-digit additions requires the model to establish new mappings to generate correct outputs, which is a more challenging task.

**Sample efficiency of fine-tuning $k$-digit models with $k + 1$-digit examples.** Building upon our previous findings that fine-tuning a model solely on $k + 1$-digit addition leads to a loss in performance for $k$-digit addition, we modify our approach to prevent the loss of performance in the $k$-digit addition task. Instead of training solely on $k + 1$-digit examples, we construct a dataset that includes all addition tasks from 1-digit to $k + 1$-digit, with the method described in the previous section. By doing so, we aim to maintain the performance of 1 to $k$-digit addition while enabling the model to learn $k + 1$-digit addition during fine-tuning.

In this experiment, we investigate the number of $k + 1$-digit training examples required for the model to effectively learn $k + 1$-digit addition when fine-tuning a pretrained model on $k$-digit addition. It is important to note that this setting differs from the previous section (Section C.1), where we focused on training models from random initialization. Here, we specifically focus on the fine-tuning process. We fine-tune individual models pretrained on each data format (using $k$-digit addition) and further train them using the same data format on a new dataset that includes all addition examples from 1-digit to $k + 1$-digit.

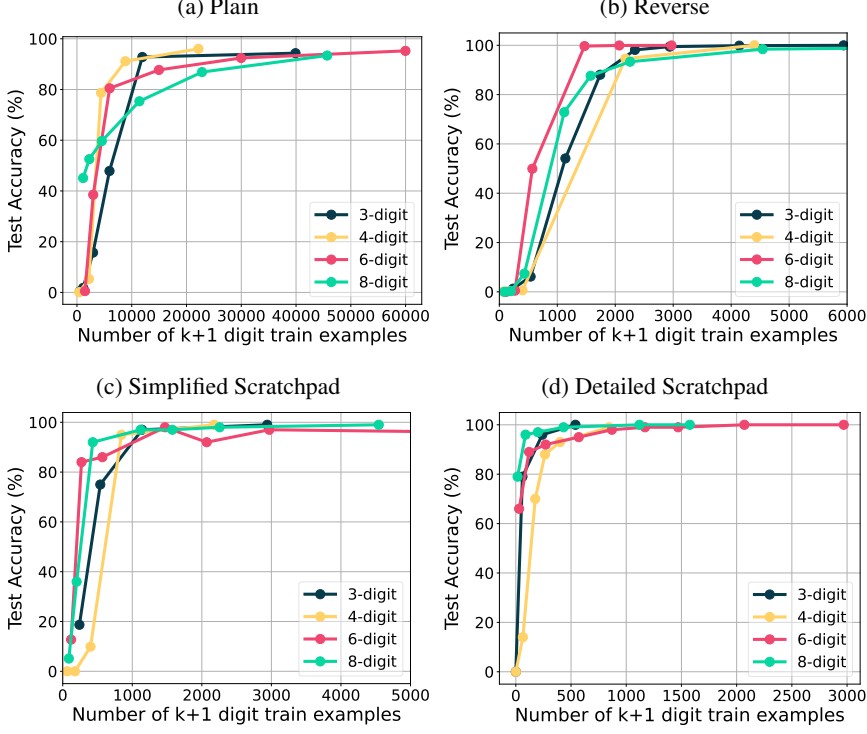

Figure 17: Fine-tuning performance of pretrained $k$-digit models using varying numbers of $k + 1$-digit examples, with corresponding data formats. The plain format requires an increasing number of $k + 1$-digit examples as the number of digits $(k + 1)$ increases. In contrast, the modified formats (reverse, scratchpad) exhibit consistent performance across different numbers of digits, requiring a relatively consistent number of examples to learn the additional digit.

The results in Figure 17 demonstrate the number of $k + 1$-digit addition samples required for a pretrained model capable of performing $k$-digit addition to learn the addition of $k + 1$ digits. The findings reveal that modified formats (reverse, scratchpad) require a relatively small number of samples (between 1000 and 5000) to learn the addition of an extra digit. In contrast, the plain format

necessitates a significantly larger number of training examples, with the requirement increasing as the number of digits grows.

This observation aligns with our previously established Lemma 2 and Lemma 1, which suggest that learning higher-digit addition in the reverse format involves processing the $i$-th digit of the operands and carrying from the previous position. This operation *remains consistent regardless of the number of digits being added*. As a result, the model primarily needs to learn how to handle longer digits to perform addition effectively.

In contrast, the plain addition format requires the model to learn a more complex function that incorporates all digits from both operands. As the number of digits increases, the complexity of this function grows as well. This highlights the greater difficulty faced by the plain format in accommodating additions with a larger number of digits.

### C.3 IMPACT OF FORMATS ON FINE-TUNING

We delve deeper into the impact of different formats on the fine-tuning process. Specifically, we investigate whether training a model in one format helps in learning addition in another format, and vice versa. To conduct this analysis, we begin with a model trained on each data format using 3-digit addition examples. We then individually fine-tune these pretrained models using different data formats, on 4-digit addition examples.

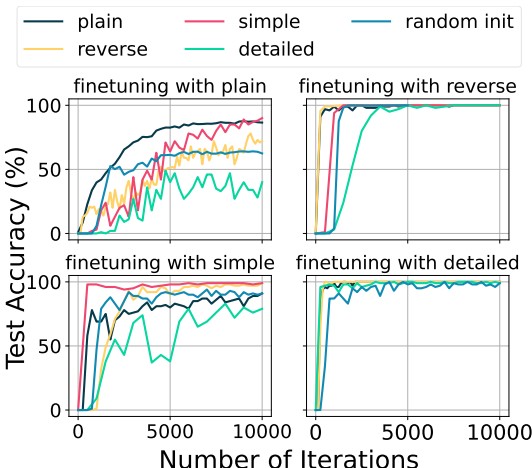

The results depicted in Figure 18 highlight some interesting findings. Firstly, we observe that a model trained with the same format as the fine-tuning format exhibits faster learning in terms of the number of iterations. For instance, training a model with the plain format outperforms training a model pretrained with scratchpad formats. This suggests that the model benefits from the consistency and familiarity provided by the same format throughout the training process.

Additionally, we notice that fine-tuning a detailed scratchpad pretrained model on other formats proves to be more challenging. This observation can be attributed to the need for the model to "unlearn" the intricacies of the verbose detailed scratchpad format and adapt to the new format. For example, the plain format does not involve the use of alphabet characters in the data, so a model pretrained with the plain format would have a low probability of generating alphabetic outputs. In contrast, a detailed scratchpad pretrained model would have encountered various alphabets and may have a tendency to output them. Therefore, adjusting to a new format requires additional effort for the model to "unlearn" the patterns specific to the previous format and effectively learn the new format it is being trained on.

Figure 18: Performance of fine-tuning a 3-digit model trained on different data formats (plain, reverse, simple scratchpad, detailed scratchpad, and random initialization) individually with different data formats of 4-digit addition. The results demonstrate that fine-tuning yields the best performance when the pretrained model and the fine-tuning format are consistent. Notably, fine-tuning a detailed scratchpad format model shows suboptimal performance. We hypothesize that this is due to the need for the model to "unlearn" the rigid and verbose format and adapt to the new format.

These findings highlight the importance of considering format consistency during the fine-tuning process, as it can impact the efficiency and effectiveness of the learning process. We will delve further into this topic in the upcoming section 8, where we fine-tune pretrained GPT-3 models. Notably, we observe that fine-tuning with reverse or simplified scratchpad formats actually yields worse results compared to fine-tuning with plain formats. For a detailed exploration of these observations, please refer to the forthcoming section.

## D  TEACHING ARITHMETIC OPERATIONS BEYOND ADDITION

While this study has a primary focus on the *addition* operation and aims to comprehend the significance of data sampling and formatting, its findings are applicable beyond the realm of addition alone. In this section, we expand our examination to include other arithmetic operations, thus demonstrating the broader applicability of our insights. We consider a mix of arithmetic tasks, including binary operations like *subtraction* and *multiplication*, and unary operations such as *sine* and *square root*. Each operation entails its unique challenges and intricacies. For instance, subtraction introduces the concept of negative numbers, multiplication can generate significantly longer outputs, and sine and square root functions entail computations involving floating-point numbers, which are considered up to four digits of precision in our work.

We acknowledge that while our examination is detailed, it does not encompass all the fundamental arithmetic operations or the entire scope of floating-point arithmetic. Specifically, our focus is primarily on integer arithmetic for binary operations, considering a limited length of digits. Additionally, for unary operations, we confine ourselves to a restricted number of digits below the decimal point.

In Section D.1, we delve into each arithmetic operation individually, exploring the impact of data formatting and determining the relevancy of our insights across disparate tasks. Further, in Section D.2, we perform an analysis of joint training across all five tasks, investigating the potential performance implications for each individual task.

### D.1  EXTENDED ARITHMETIC OPERATIONS

In order to extend our analysis to arithmetic operations beyond addition, we consider the following tasks:

**Subtraction** $(-)$. We consider subtraction of positive numbers up to 3 digits, written as $A_3A_2A_1 - B_3B_2B_1 = C_3C_2C_1$ in (i) plain formatting, and $\$A_3A_2A_1 - B_3B_2B_1 = C_1C_2C_3\$$ in (ii) reverse formatting. As with addition, scratchpad-based methods (iii, iv), present the intermediate steps of digit-wise subtraction and handling of carry-ons. These steps proceed from the least significant bit (LSB) to the most significant bit (MSB). If the final result after computing all the digit-wise subtractions is negative, we subtract the number in the most significant bit (MSB) position multiplied by 10 to the power of (number of digits in the output - 1) from the remaining digits in the output. In Section B.4, we present an alternative version of the detailed scratchpad formatting for subtraction.

**Multiplication** $(\times)$. We consider multiplication of positive numbers up to 2-digits. (i) Plain formatting examples are formatted as $A_2A_1 * B_2B_1 = C_4C_3C_2C_1$, while (ii) reverse formatting is formatted as $\$A_2A_1 * B_2B_1 = C_1C_2C_3C_4\$$. The (iv) detailed scratchpad method simplifies each intermediate step by conducting a series of multiplications between the first operand and each digit of the second operand, starting from the least significant bit (LSB) and moving toward the most significant bit (MSB). For each step, we multiply the result by an exponentiation of 10 corresponding to the relative digit position.

**Sine** $(\sin)$. We consider decimal numbers within the range $[-\pi/2, \pi/2]$, truncated to 4-digit precision. (i) Plain formatting examples are formatted as $\sin(A_0.A_1A_2A_3A_4) = B_0.B_1B_2B_3B_4$. For (iv) detailed scratchpad method, we include the Taylor series expansion steps for sine, which is represented as $\sin(x) = x - \frac{1}{3!}x^3 + \frac{1}{5!}x^5 - \frac{1}{7!}x^7 + \cdots$. These intermediate steps involve exponentiation, which may not be any easier to compute than the sine operation itself.

**Square Root** $(\sqrt{})$. We consider decimal numbers within $[1, 10)$, truncated to 4-digits of precision with the format, written as $\text{sqrt}(A_0.A_1A_2A_3A_4) = B_0.B_1B_2B_3B_4$ for (i) plain formatting. For (iv) detailed scratchpad method, we enumerate each step of Newton's method to compute the square root function. The iterative formula is given by $x_n = \frac{1}{2}(x_{n-1} + \frac{x}{x_{n-1}})$, where $x_0$ is initialized as the floor of the square root value of the operand $x$. These intermediate steps involve a division operation, which can be as complex as the square root operation itself.

For evaluation of sine and square root, we classify the result $\hat{y}_i$ as correct if the absolute difference between $\hat{y}_i$ and the ground truth value $y_i$ is less than or equal to a predefined threshold $\epsilon \geq 0$.

For each arithmetic task, we explore both the plain format and the detailed scratchpad format. The detailed scratchpad formatting for each task is illustrated in Figure 19 and Appendix J. For subtraction, the process involves breaking down the operation into intermediate steps of digit-wise subtraction, including carry-ons when necessary. Unlike addition, subtraction requires an additional step to handle cases where the first operand is smaller than the second. Further details on the detailed scratchpad for subtraction can be found in Section B.4. For multiplication, each intermediate step carries out a 2-digit $\times$ 1-digit multiplication between the first operand and each separate digit of the second operand. For sine and square root, we utilize a sequence of *iterative approximations* instead of algorithmic explanations. Specifically, Taylor's series expansion steps for sine and Newton's method steps for square root are used. It is important to note that while addition, subtraction, and multiplication are broken down into simpler operations at each step, CoT for sine and square root functions requires intermediate steps involving operations like exponentiation or division, which might not be inherently simpler.

---

**Detailed scratchpad formatting for different arithmetic tasks**

```
Examples of detailed scratchpad formatting for different arithmetic
tasks:
(1) Subtraction - includes borrows for intermediate steps, (2)
Multiplication - decomposes the second operand for 2-digit × 1-digit
multiplication at each step, (3) Sine - utilizes Taylor series
expansion, and (4) Square root - employs Newton's method.
```

**Subtraction**

```
Input:
128-367
Target:
<scratch>
[1,2,8] has 3 digits.
[3,6,7] has 3 digits.
[1,2,8] - [3,6,7] , A=[] , C=0 , 8-7-0=1 ,
A->1 , C->0
[1,2] - [3,6] , A=[1] , C=0 , 2-6-0+10=6 ,
A->6 , C->-1
[1] - [3] , A=[6,1] , C=-1 , 1-3-1=-3 , A
->-3 , C->-1
[] - [] , A=[-3,6,1]
-300+61=-239 , END
</scratch>
-2 3 9
```

**Sine**

```
Input:
sin(1.5707)
Target:
<scratch>
x_0=1.5707
x_1: x_0 - 1/3! * (x^3) , x_1=0.9247
x_2: x_1 + 1/5! * (x^5) , x_2=1.0043
x_3: x_2 - 1/7! * (x^7) , x_3=0.9996
x_4: x_3 + 1/9! * (x^9) , x_4=0.9997 , END
</scratch>
0.9997
```

**Multiplication**

```
Input:
12*36
Target:
<scratch>
[1,2] has 2 digits.
[3,6] has 2 digits.
[1,2] * 6 , A=[7,2] , k=1 , B=[7,2] , C
=0+72=72
[1,2] * 3 , A=[3,6] , k=10 , B=[3,6,0] , C
=72+360=432 , END
</scratch>
4 3 2
```

**Sqrt**

```
Input:
sqrt(2.7174)
Target:
<scratch>
x_0=1
x_1: 1/2*(1+2.7175/1)=1.8587, x_1=1.8587
x_2: 1/2*(1.8587+2.7175/1.8587)=1.6603, x_2
=1.6603
x_3: 1/2*(1.6603+2.7175/1.6603)=1.6485, x_3
=1.6485
x_4: 1/2*(1.6485+2.7175/1.6485)=1.6484, x_4
=1.6484 , END
</scratch>
0.6484
```

Figure 19: Examples of the detailed scratchpad format for different arithmetic tasks such as subtraction, sine, multiplication, and square root.

The results depicted in Figure 20 indicate that similar to the findings of addition, the detailed scratchpad format significantly improves performance over *plain* or *reverse* formats and yields efficient results even with few samples for subtraction and multiplication tasks. Interestingly, we find *reverse* is not particularly effective in multiplication. On the other hand, the detailed scratchpad format exhibits reduced efficiency for sin and $\sqrt{\ }$ compared to other operations $(+, -, \times)$. This

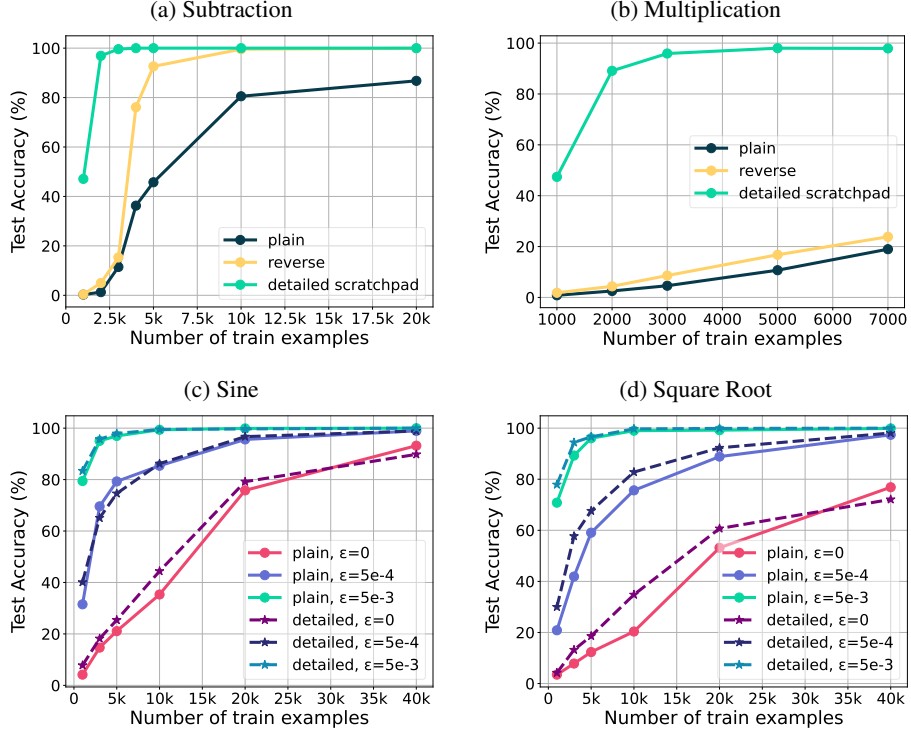

Figure 20: Performance of $3-$digit subtraction, $2-$digit multiplication, $4-$digit precision sine and square root with varying data formats. As with addition, reverse always produces improved sample complexity and performance for all operations. For sine and square root, scratchpad formatting provides limited improvement. This discrepancy can be attributed to the complexity of the intermediate steps involved in the detailed scratchpad.

discrepancy can be traced back to the complexity of the intermediate steps involved in the detailed scratchpad. While addition, subtraction, and multiplication are decomposed into simpler functions, sine and square root operations involve more intricate operations. For a broader analysis of the error profile, see Appendix B.6.

## D.2 JOINTLY TRAINING ON ALL FIVE ARITHMETIC TASKS

So far, we only considered the problem of learning different arithmetic operations individually. In this section, we study the effect of jointly training on all five arithmetic tasks - addition, subtraction, multiplication, sine, and square root. We construct a single train dataset incorporating all task $\mathcal{D}_{\text{train}} = \{\mathcal{D}_{\text{train}}^{+}, \mathcal{D}_{\text{train}}^{-}, \mathcal{D}_{\text{train}}^{\times}, \mathcal{D}_{\text{train}}^{\sin}, \mathcal{D}_{\text{train}}^{\sqrt{}}\}$, and randomize the sequence of tasks in our train samples. For example, a randomly chosen segment of the training data may exhibit a task order such as $(+, -, \sin . -, \times, \times, \sqrt{}, ...)$. We consider $10,000$ training examples for each task of addition, subtraction, sine, and square root and $3,000$ for multiplication.

The model's performance, after training on our joint dataset $\mathcal{D}_{\text{train}}$, is evaluated in both zero-shot and few-shot settings. These results are also compared with the performance of models that were trained separately on each dataset $(\mathcal{D}_{\text{train}}^{+}, \mathcal{D}_{\text{train}}^{-}, \mathcal{D}_{\text{train}}^{\times}, \mathcal{D}_{\text{train}}^{\sin}, \mathcal{D}_{\text{train}}^{\sqrt{}})$, identical to those used to construct $\mathcal{D}_{\text{train}}$. In the few-shot setting, each task is given examples from any of the five arithmetic tasks (not necessarily related to the test task under consideration) or prompt texts, followed by test queries specific to the task of interest. For further details on the few-shot prompting methods used, please refer to Section E.

Table 9 shows that joint training significantly enhances the zero-shot performance for multiplication and square root tasks, yet it slightly reduces the performance for subtraction. Generally, few-shot prompting exhibits improved performance. Notably, the performance of few-shot prompting remains consistent regardless of whether the exemplars provided are from unrelated tasks or are task-specific. We propose that this consistency is due to our randomized task sequence during training, which

presents the model with numerous instances where one task directly follows another, thus simulating few-shot prompting with different tasks. Furthermore, we observe that text prompting performs similar to zero-shot. We conjecture that this is because the training data does not include text data and the model has never encountered text and therefore, text prompting serves as a random prefix attached to our test query.

Table 9: Performance of models trained individually and jointly on five arithmetic tasks. The threshold $\epsilon$ for $\sin$ and $\sqrt{\ }$ functions is set to 0. For the models trained jointly on all five tasks, we evaluate their performance in both a zero-shot setting and a few-shot setting. In the few-shot setting, each task is presented with exemplars from one of the five arithmetic tasks or prompted with text, followed by task-specific test queries. The results show that few-shot prompting with any arithmetic operators (even unrelated to the test task) generally improves performance. However, text prompting shows performance similar to the zero-shot setting.

| | Trained on individual task | Trained jointly on all 5 tasks | | | | | | |
| | | Zero-shot | Few-shot exemplar format | | | | | |
| | | | + | − | × | sin | sqrt | text |
|---|---|---|---|---|---|---|---|---|
| + | 84.06 | 87.96 | 96.45 | 96.90 | 96.92 | 97.06 | 97.01 | 88.71 |
| − | 79.97 | 72.83s | 81.28 | 79.59 | 81.39 | 81.84 | 81.74 | 68.91 |
| × | 4.58 | 14.28 | 18.86 | 18.96 | 15.43 | 19.20 | 19.59 | 15.48 |
| sin | 35.03 | 34.74 | 34.35 | 34.31 | 34.34 | 32.64 | 33.42 | 33.96 |
| sqrt | 19.85 | 27.37 | 26.65 | 26.74 | 26.70 | 25.60 | 25.61 | 26.02 |

# E    MIXING TEXT WITH ARITHMETIC DATA

Until now, our focus was primarily on models trained exclusively on arithmetic tasks. However, in practice, large language models (LLMs) utilize a combination of arithmetic and *text* data for training. In this section, we broaden our scope by incorporating both addition samples and text into our pretraining data. We then evaluate the trained models with various few-shot prompts to analyze if the model is able to effectively identify the correct context.

**Experimental Setup.**    We mix addition and text data in our experiment using the Shakespeare dataset (Karpathy, 2015) that includes $1,115,394$ tokens of text, $10,000$ plain addition examples ($120,027$ tokens) without the $ delimiter, and $3,000$ detailed scratchpad formatted addition examples ($813,510$ tokens). We fix the number of detailed scratchpad examples and plain addition examples ($3,000$ and $10,000$ respectively) while varying the number of each example type in the training process. The Shakespeare text is segmented into dialogue chunks, with a random number of consecutive plain addition data and detailed scratchpad data inserted between them. We use a character-level tokenizer with a vocabulary size of $80$, containing all characters present in the dataset, including alphabets, digits, and certain symbols like $+, =$ and \n.

## E.1    FEW-SHOT PROMPTING

Given the mixed nature (arithmetic and text) of our dataset, introducing relevant examples seems an effective strategy to prime the model to generate the desired type of output. To assess the performance of such few-shot ($1/2/3-$shot) prompting, we provide task-specific exemplars as illustrated in Figure 21. Plain addition formatted exemplars are used for testing plain addition inputs, while detailed scratchpad formatted exemplars are utilized for assessing performance on detailed scratchpad formatted inputs. Additionally, we experiment with demonstrating text (see Appendix E.3. for details) before querying addition (which we denote, Text-prompt). For each 1/2/3-shot and text prompting, average performance is reported over a fixed set of exemplars. Standard deviations of these prompts are denoted by shaded areas in the plots. The term "few-shot" refers to the reported mean of all 1/2/3-shot prompting results.

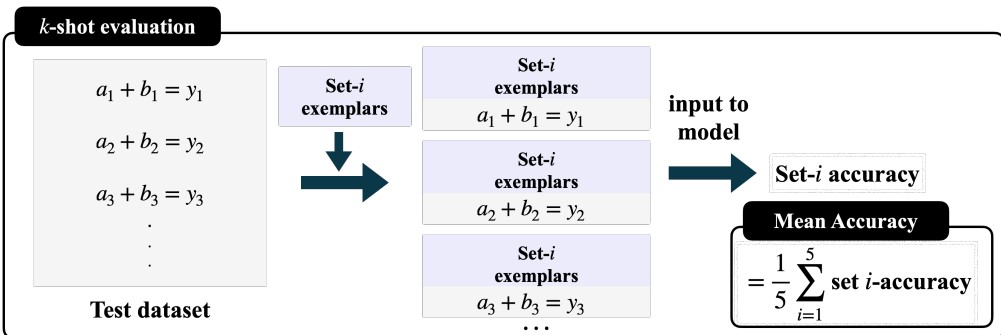

Figure 21: Few-shot prompting method. Few-shot prompting performance is evaluated by presenting relevant exemplars of addition and detailed scratchpad formatted inputs. Each 1/2/3-shot prompting is tested on a fixed five set of exemplars, and the accuracy is averaged over these evaluations.

Figure 22 shows that few-shot prompting directs the enhancement of performance, thereby allowing plain addition to perform almost perfectly with 40,000 train samples. Intriguingly, performance remains high on plain addition even with the inclusion of a text prompt, given a substantial number of addition examples. We hypothesize that this is due to the structure of our mixed dataset where addition examples are interspersed within Shakespeare data. With the incorporation of more addition examples, instances where addition examples directly follow Shakespeare text increase, leading to a decrease in potential inconsistencies when text content is present during addition test queries.

## E.2    DISENTANGLING THE EFFECT OF TEXT ON PROMPTING

To disentangle the effects of the textual content in the training data, we train a model strictly on plain addition, utilizing an enlarged vocabulary that also includes alphabet characters, thereby enabling text

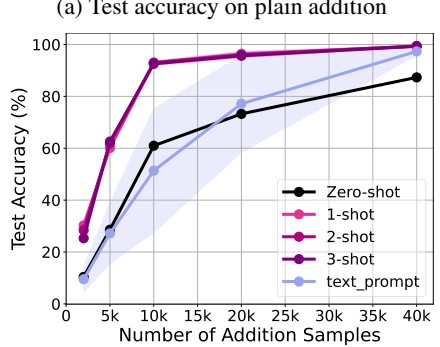 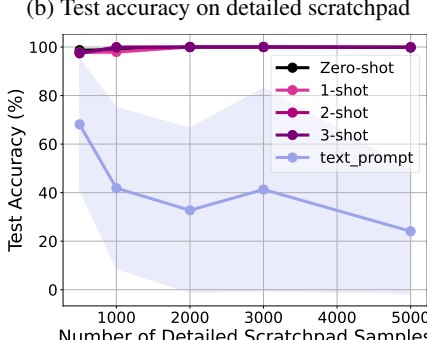

Figure 22: Performance of NanoGPT model trained with the Shakespeare dataset, addition dataset in plain, and detailed scratchpad format. The number of plain (left) and detailed scratchpad (right) formatted addition samples are varied. Performance is evaluated on zero-shot, few-shot, and text prompts, with the shaded area representing the standard deviation across various prompt exemplar sets. The results indicate a consistent enhancement in model performance using few-shot prompting.

prompting. (Note that previous experimental settings on plain formatted additions used a vocabulary size of 13, which only includes 10 numerals and 3 symbols - "+","=","\n"). We introduce a variant of few-shot prompting, termed as **noisy-prompt**, which prompts the model with erroneous addition exemplars, *i.e.*, , $A + B = C$, where $C \neq A + B$.

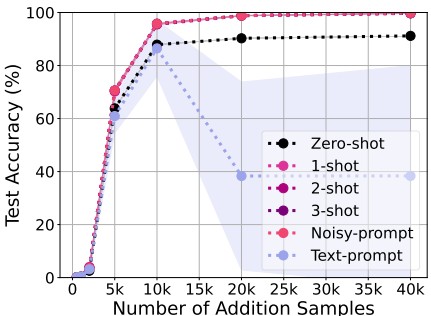

Figure 23: Performance of NanoGPT model trained exclusively on plain addition, but with an extended vocabulary including both addition and alphabets (vocabulary size = 80). Few-shot prompting, using both correct addition examples (1, 2, 3-shot) and incorrect addition examples (noisy-prompt) leads to enhanced performance, while the use of text prompts results in a degradation of performance when the model is trained solely on addition.

Figure 23 shows that few-shot prompting contributes to performance enhancement even when the model is confined to training on a single plain addition task. Even in the presence of noisy prompting, simply providing the model with the $A + B = C$ format yields performance nearly identical to few-shot prompting, aligning with the result observed by Min et al. (2022). Conversely, we notice that text prompts negatively influence performance when the model is trained only on addition. This finding reinforces our earlier observation in Figure 6 that the advantageous impact of text prompts originates from the combined text and addition data.

## E.3 PROMPTING WITH TEXT

To extend on the few-shot prompting experiments from Section D.2, we also evaluate the effect of prompting the model with pure-text prompts. If few-shot prompting with addition samples improves accuracy through in-context learning, we expect few-shot prompting with text to hurt accuracy since the text exemplars are *out-of-context*. We use five different types of text exemplars: (i) **Prompt1:** a short text prompt that is not present in the Shakespeare dataset, (ii) **Prompt2:** a short text prompt extracted from within Shakespeare dataset, (iii) **Prompt3:** a longer form text prompt extracted from within the Shakespeare dataset, (iv) **Prompt4:** a prompt that includes numbers, and (v) **Prompt5:** a

long text prompt that is not present in the Shakespeare dataset. More details on the text prompts can be found in Figure 24.

---

### Text prompts for few-shot experiments

Examples of the different text prompts used in the few-shot experiment. Each exemplar is separated by '---'.

#### Prompt 1. Short, $\notin$ Shakespeare

```
et tu brute
---
hello, world
---
how are you doing?
---
agi is coming
---
boom! stability
```

#### Prompt 2. Short, $\in$ Shakespeare

```
JULIET:
Romeo!
---
All:
Resolved. resolved.
---
VOLUMNIA:
Why, I pray you?
---
CORIOLANUS:
Nay! prithee, woman,--
---
MENENIUS:
I mean, thy general.
```

#### Prompt 3. Long, $\in$ Shakespeare

```
JULIET:
Romeo!
ROMEO:
My dear?
---
MENENIUS:
This is good news:
I will go meet the ladies. This Volumnia
Is worth of consuls, senators, patricians,
---
LADY ANNE:
Foul devil, for God's sake, hence, and trouble us
 not;
For thou hast made the happy earth thy hell,
Fill'd it with cursing cries and deep exclaims.
---
BUCKINGHAM:
I fear he will.
How now, Catesby, what says your lord?
---
CATESBY:
Bad news, my lord: Ely is fled to Richmond;
And Buckingham, back'd with the hardy Welshmen,
Is in the field, and still his power increaseth.
```

#### Prompt 4. Has number, $\notin$ Shakespeare

```
I go 16-12
That's the code to my heart, ah
I go 1-6-1-2
Star
---
Like a river flows 17-23
Surely to the sea 15-22
Darling, so it goes 46-92
Some things are meant to be
---
I got my first real 6-string
Bought it at the five and dime
Played it 'til my fingers bled
Was the summer of '69
---
I think someday I might just 5-3-2-1 get a real job
I spent half of my life 1-2-3 in a bus or on a flight
I'm getting off 17-36-8-2 the road and in a real job
---
Every time that 27-67-29 I look in the mirror
All these lines on my 1-3-92-5 face getting clearer
The past 45-5-3 is gone
```

#### Prompt 5. Long, $\notin$ Shakespeare

```
Is this the real life? Is this just fantasy? Caught in a landside, no escape from reality.
Open your eyes, look up to the skies and see.
I'm just a poor boy, I need no sympathy. Because I'm easy come, easy go,
Little high, little low,
Any way the wind blows doesn't really matter to me, to me.
---
It's my life
And it's now or never
I ain't gonna live forever
I just want to live while I'm alive
My heart is like an open highway
Like Frankie said, I did it my way
---
Destruction leads to a very rough road but it also breeds creation
```

```
And earthquakes are to a girl's guitar, they're just another good vibration
And tidal waves couldn't save the world from Californication
---
I want to stay
But I need to go
I want to be the best for you
But I just don't know what to do
'Cause baby, say I've cried for you
The time we have spent together
Riding through this English whether
---
Lorem ipsum dolor sit amet, consectetur adipiscing elit. Vestibulum mattis in leo vel
gravida.
Pellentesque libero elit, scelerisque varius vehicula a, hendrerit et tellus.
Proin convallis neque nisl, nec lobortis est scelerisque tincidunt.
Nunc venenatis auctor urna.
Class aptent taciti sociosqu ad litora torquent per conubia nostra.
```

Figure 24: Text prompt exemplars for few-shot experiments.

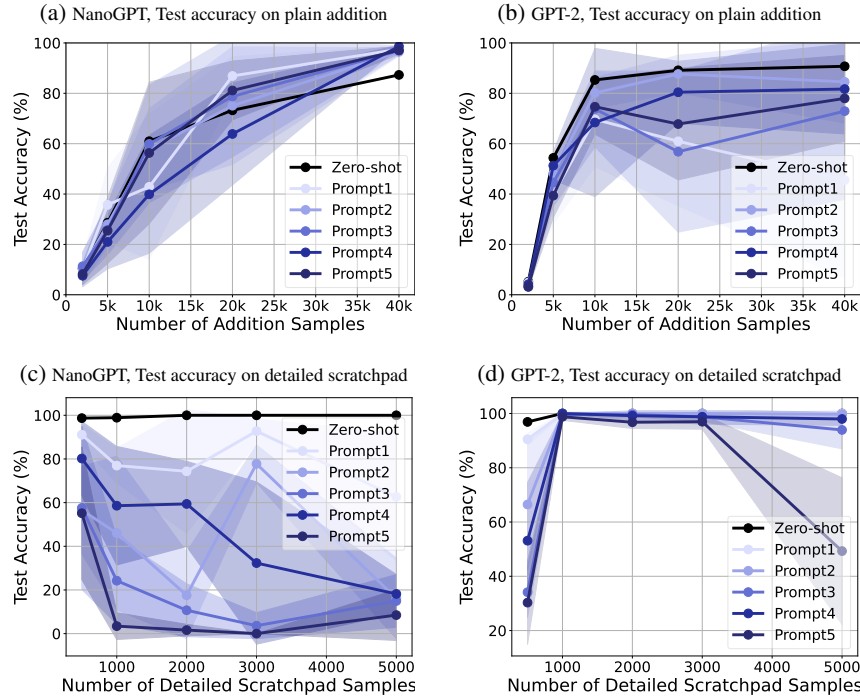

Figure 25: Experiments on few-shot prompting with different text prompts: (i) Prompt1: short text not in Shakespeare dataset (ii) Prompt2: short text within Shakespeare dataset (iii) Prompt3: long text within Shakespeare dataset (iv) Prompt4: text with numbers (v) Prompt5: long text not in the Shakespeare dataset. Each prompt (Prompt 1-5) consists of five distinct exemplars. The solid lines represent the mean performance across the five exemplars, while the shaded area indicates the standard deviation. We observe that the effectiveness of text prompts varies greatly depending on the exemplars used.

The results presented in Figure 25 show notable variations in evaluation accuracy for addition, depending on the chosen text prompts. Longer text prompts (Prompt 5) typically result in a more significant decline in performance. With the exception of NanoGPT trained on plain addition, the result in Figure 26 indicates that employing text prompts followed by test addition queries tends to have an adverse impact on the overall model performance, whereas incorporating relevant few-shot exemplars (1/2/3-shot) is beneficial. This aligns well with our intuition on the benefits on in-context learning.

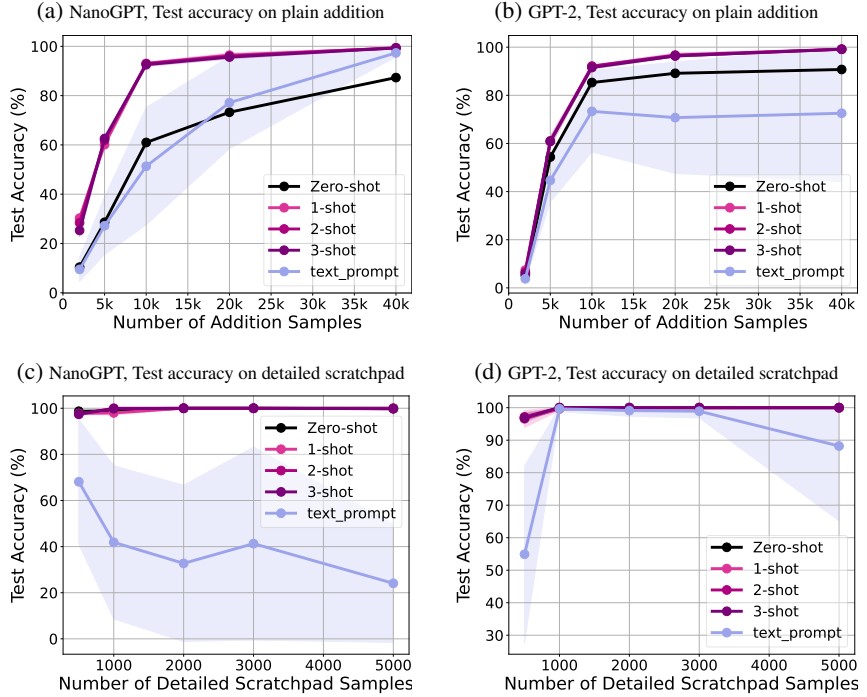

Figure 26: Performance of NanoGPT and GPT-2 model trained with entire Shakespeare dataset and a varying number of samples of plain addition, and addition with detailed scratchpad dataset. Performance is evaluated on test prompts formatted as plain addition and detailed scratchpad. Few-shot experiments are based on an average of 5 exemplars, while text prompts involve an average of 25 exemplars. The shaded area represents the standard deviation. Our observations indicate that few-shot prompting consistently improves performance, whereas test prompts generally have a negative impact.

## F   FINE-TUNING, SCALING, AND PRETRAINING IN LARGER MODELS

This section focuses on bridging the gap between our experiments on NanoGPT and the more realistic setting of larger language models like GPT-2 and GPT-3. We begin by comparing the performance of NanoGPT and GPT-2 models when trained from random initialization. This comparison highlights the improved performance achieved with the larger model scale, especially in the zero-shot setting. Subsequently, we delve into the impact of tokenization methods and model pretraining in GPT-2 models. Our exploration reveals the crucial role of pretrained models and the consistent tokenization of numbers (achieved by introducing spaces) during the training phase for arithmetic tasks. Building on these findings, we proceed to fine-tune a pretrained GPT-3 model on various arithmetic tasks, employing different data formats.

**Comparing NanoGPT and GPT-2.**   To examine the impact of scale on arithmetic performance, we explore a larger GPT-2 model with 85 million parameters, featuring twice as many self-attention layers, heads, and embedding size compared to the previously used NanoGPT model. We train the GPT-2 model from scratch using character-level tokenization, jointly on text and addition tasks, adopting both plain and detailed scratchpad formats; an approach mirroring the setting in Section 7. The results depicted in Figure 27 demonstrate that the larger model outperforms in both plain and detailed scratchpad evaluations. For a comprehensive analysis of GPT-2, including few-shot learning and the influence of text prompts, refer to Figure 25 and Figure 26.

**Going from character-level tokenization to BPE.**   The transition to a GPT-2 setup necessitates several modifications. Firstly, we shift to OpenAI's Tiktoken BPE tokenizer, which is the default tokenizer for the pretrained GPT-2 model, featuring a vocabulary size of 50,257. We also examined two different training approaches: training the model from random initialization (scratch) and fine-tuning the pretrained model sourced from Huggingface. To ensure uniform digit tokenization,

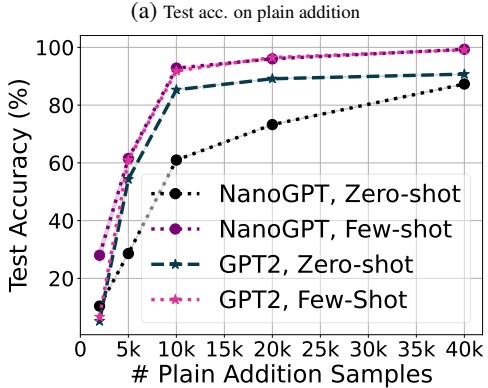 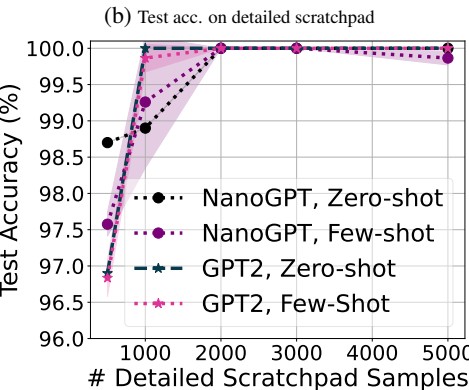

Figure 27: Performance of NanoGPT and GPT-2 model trained with entire Shakespeare dataset and a varying number of samples of plain addition, and addition with detailed scratchpad dataset. Performance is evaluated on test prompts formatted as plain addition and detailed scratchpad. Few-shot experiments are based on an average of 5 exemplars, while text prompts involve an average of 25 exemplars. The shaded area represents the standard deviation. Our observations indicate that few-shot prompting consistently improves performance, whereas test prompts generally have a negative impact.

alterations were made in data formatting to include spaces between numbers. This change aims to circumvent potential inconsistent tokenization of numbers while utilizing the Tiktoken tokenizer.

Figure 7 shows that GPT-2 demonstrates high performance in addition tasks with both character-level tokenization and Tiktoken with spaces between digits. This aligns with the results by Wallace et al. (2019), suggesting that character-level tokenization exhibits stronger numeracy capabilities compared to a word or sub-word methods. Furthermore, comparing the models trained from scratch and the models trained from the pretrained model, we observe that fine-tuning a pretrained model results in better performance compared to training a model from scratch.

**GPT-3 experiments: Supervised fine-tuning.** We extend our experiments to verify if our observations hold while fine-tuning larger pre-trained models. In the following, we consider three GPT-3 variants: Ada, Curie, and Davinci. Note that since we perform fine-tuning using the OpenAI APIs, by default *only the completions are loss generating tokens*. Therefore, these experiments are slightly different when compared to the previous settings. We fine-tune these models using the same four data formatting methods as our NanoGPT experiments: (i) *plain* formatting, (ii) *reverse* formatting, (iii) *simplified scratchpad*, and (iv) *detailed scratchpad*. These formats are identical to those from our NanoGPT experiments except for one aspect. We introduce spaces between numbers in *plain* and *reverse* formatting to ensure consistent tokenization.

Due to budget constraints, all experiments were conducted using a fine-tuning dataset of $1,000$ examples, and models were trained for 4 epochs. Performance evaluation was carried out on $1,000$ examples that were disjoint from the training dataset. Note that this training scale is significantly smaller than our experiments on NanoGPT, which employed $10,000$ training examples for $5,000$ iterations, with evaluations conducted on $10,000$ test examples. However, given these models' extensive pretraining on large data corpora, this scale can be deemed rational.

The results for addition and subtraction tasks are presented in Table 2 and Table 10, respectively. We observed that initiating with a pretrained GPT-3 model significantly improves performance compared to training NanoGPT or GPT-2 models from random initialization with only 1000 samples. This indicates the utility of leveraging pretrained models for improved arithmetic performance. Interestingly, while reverse formatting and simplified scratchpad formats improve addition performance, they adversely affect subtraction performance. This observation is consistent with our earlier finding depicted in Figure 18, wherein transitioning from one data format to another often results in lower performance compared to initiating training from random initialization. We postulate that this discrepancy may be due to the pretrained GPT-3 model's requirement to adapt to the reversed approach and "unlearn" its knowledge of plain formatting arithmetic, thereby introducing additional complexity. On

the other hand, the detailed scratchpad method achieves excellent performance, albeit with increased training and inference costs due to higher token requirements.

Table 10: Evaluation of subtraction performance for fine-tuned GPT-3 models: Davinci, Curie, and Ada. In each case, the model is finetuned on 1000 samples of addition in the corresponding format.

| GPT-3 Model | Zero-shot | Plain | Reverse | Simplified Scratchpad | Detailed Scratchpad |
|---|---|---|---|---|---|
| Davinci | 0.1% | 84.8% | 66.0% | 15.4% | **99.5%** |
| Curie | 0.1% | 24.1% | 6% | 3.8% | **92.5%** |
| Ada | 0.0% | 3.7% | 2.6% | 3.4% | **81.5%** |

Table 11: Evaluation of sine and square root performance for fine-tuned GPT-3 models: Davinci, Curie, and Ada. In each case, the model is finetuned on 1000 samples of addition in the corresponding format.

| GPT-3 Model | eps | Sine | | | Square Root | | |
|---|---|---|---|---|---|---|---|
| | | Zero-shot | Plain | Detailed Scratchpad | Zero-shot | Plain | Detailed Scratchpad |
| Davinci | 0 | 0% | **11.0%** | 10.3% | 0% | 0.7% | **4.6%** |
| | 5e-4 | 0% | **35.9%** | 29.7% | 0% | 7.5% | **17.2%** |
| | 5e-3 | 0.4% | **85.5%** | 72.8% | 0% | 59% | **60.5%** |
| Curie | 0 | 0.0% | **8.6%** | 1.2% | 0.0% | 0.7% | **2.1%** |
| | 5e-4 | 0.4% | **32.7%** | 5.4% | 0.1% | **6.5%** | 6.0% |
| | 5e-3 | 0.9% | **80.8%** | 15% | 0% | **52.7%** | 30.2% |
| Ada | 0 | 0.0% | **5.8%** | 4.3% | 0.0% | 0.3% | **2.7%** |
| | 5e-4 | 0.0% | **21.4%** | 9.1% | 0.0% | 3.8% | **11.9%** |
| | 5e-3 | 0.3% | **67.8%** | 25.2% | 0.0% | 32.2% | **45.8%** |

For the more complex sine and square root tasks as shown in Table 11, we found that training with only 1000 samples is insufficient to generate exact answers (eps=0). The GPT-3 model, fine-tuned with 1,000 samples, performs worse than the NanoGPT model trained with 10,000 samples. Further experiments with larger training datasets are necessary for deeper insights and improved performance on these tasks.

It is worth mentioning that while few-shot prompting notably improves the performance of all three GPT-3 models, their zero-shot performance is quite poor (as shown in the leftmost column of the tables). However, post-training, few-shot prompting becomes less effective as OpenAI's fine-tuning process trains the model on individual prompts and desired completions serially, rather than in concatenation with multiple examples like in our NanoGPT experiments. Consequently, our comparisons primarily focus on the **zero-shot performances** of each task.

## G  TOKEN EFFICIENCY ACROSS DATA FORMATS

Figure 4a demonstrates that more detailed training data leads to improved sample efficiency. However, this comparison does not account for the cost associated with training and inference. To address this, we conduct a cost analysis based on (i) the number of tokens within the train dataset (measuring the efficiency of the training dataset), and (ii) the number of tokens encountered during training. For (i) token-efficiency of train dataset, we calculate the number of tokens within the dataset which can be derived by *number of samples × number of tokens per sample*. For instance, the mean token count for a single training example in a 3-digit addition task is 13 for plain format, 15 for reverse format, 64 for simplified scratchpad format, and 281 for detailed scratchpad format. For (ii) token-efficiency for training, we calculate the number of tokens encountered by the model which is *number of iterations × context length of the model × batch size*. This approach ensures our cost calculation accounts for a vanilla implementation of attention with no additional optimizations (Pope et al., 2023). Table 12 presents the number of tokens required for prompting and completion in each data format, per example. Evidently, the detailed scratchpad method uses considerably more tokens compared to other techniques.

The result in Figure 4b indicates that reverse formatting is the most token-efficient approach for training dataset construction. While detailed scratchpad training is more sample efficient, it necessitates a larger number of tokens per sample, both during training and inference. Given that the inference cost for commercial models is determined by the number of tokens utilized per inference call (sum of prompting and completion tokens), the use of models trained on detailed scratchpad formats may escalate overall costs. Furthermore, since the cost of a single forward pass is quadratic in the number of tokens, this is important to consider. On the other hand, the result in Figure 28 shows that for the same number of tokens input to the model to be trained on, the model is trained faster with detailed scratchpad data. Therefore, for practical usage, it is crucial to evaluate both the number of samples needed for achieving the desired performance and the actual token demands during training and inference.

Table 12: Token requirements for prompting and completion per single example of 3-digit addition.

|            | Plain | Reverse | Simplified Scratchpad | Detailed Scratchpad |
|------------|-------|---------|-----------------------|---------------------|
| Prompt     | 8     | 9       | 23                    | 23                  |
| Completion | 5     | 6       | 41                    | 258                 |
| **Total**  | **13**| **15**  | **64**                | **281**             |

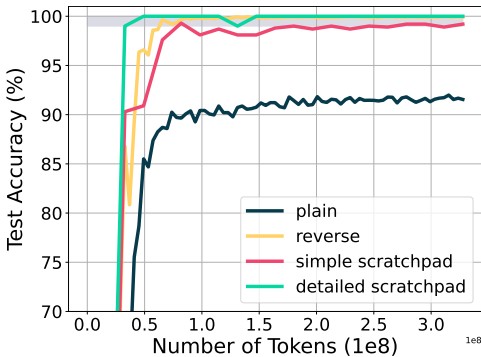

Figure 28: Model performance by the number of tokens input to the model during training.

## H  LENGTH GENERALIZATION

In this section, we present results from experiments conducted to assess the model's ability to generalize across different digit lengths. Initially, we exclude training examples featuring 2-digit operands from the 10,000-sample addition dataset, yielding a reduced dataset of 7,655 samples, consisting solely of 1 or 3-digit operands. The model is trained with reverse format and its performance is evaluated on test dataset containing 100 random samples of 1-digit, 2-digit, 3-digit, and 4-digit additions. The results in Figure 29 demonstrate that the NanoGPT model is incapable of performing 2-digit and 4-digit additions. This suggests an inherent necessity for exposure to all digit combinations to perform accurate calculations and lacks generalization capabilities for unseen digit lengths.

Additionally, we investigate the model's ability to extrapolate over larger digit lengths. The model is trained on 7-digit plain-formatted additions (each digit addition comprises 16650 samples, except 1-digit addition, which is trained on 100 samples). Its ability to add add 8-digit numbers is then put to test. The results in Figure 29 show that the model is unable to generalize to a greater number of digits beyond what it has been trained on. Similarly, when training the model on 10-digit binary numbers, it fails to generalize to 11-digit binary additions, further confirming its limited ability to handle unseen digit combinations.

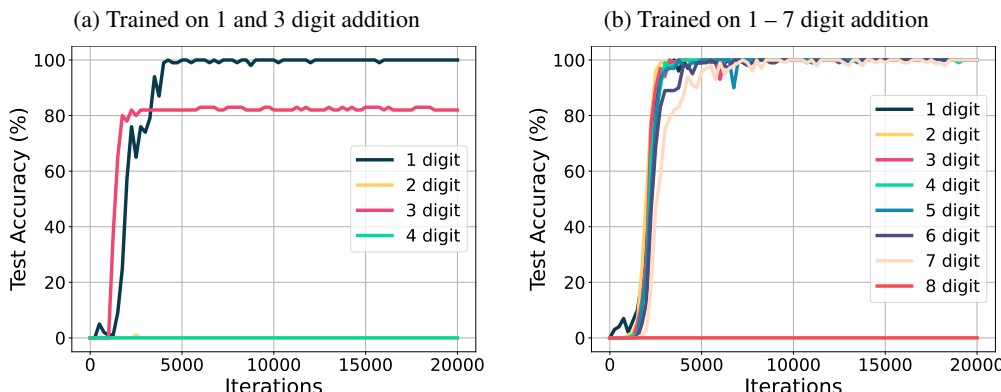

Figure 29: Generalization experiments testing NanoGPT's performance on unseen numbers of digits in addition tasks. (Left): NanoGPT trained on reverse formatted addition with 1 and 3 digits, and tested on additions ranging from 1 to 4 digits. (Right): NanoGPT trained on up to 7-digit plain formatted addition and tested on additions ranging from 1 to 8 digits. In both cases, NanoGPT exhibits an inability to perform addition on digits it has not been exposed to.

We further explore the impact of detailed scratchpad formatting. The model trained on additions of up to 3 digits, struggles to generalize to 4-digit additions. Notably, it randomly drops a single digit from the 4-digit number, erroneously perceiving it as a 3-digit number. We illustrate this difficulty in Figure 30 through multiple detailed error cases, ranging from instances in which only the test query is provided (Case 1) to scenarios where all intermediate steps are provided except only the final answer (Case 5). The prompts are highlighted in light grey and the responses generated by our trained NanoGPT model are highlighted in light green. These cases emphasize the model's shortcomings in accurately managing larger digit lengths.

---

**Examples for length generalization prompts**

Results obtained by prompting the NanoGPT model with larger digits than those it was trained on.  The model is trained using detailed scratchpad formats with 3-digit numbers.  We evaluate its performance on 4-digit numbers, with varying levels of provided information.  The prompt input is highlighted in a light blue box, while the model's output is highlighted in a light green box.

**Case 1: Just asking the question**

---

```
Input:
8465+3541
Target:
<scratch>
[8,4,6] has 3 digits.
[3,5,1] has 3 digits.
[8,4,6] + [3,5,1] , A=[] , C=0 , 6+1+0=7 , A->7 , C->0
[8,4] + [3,5] , A=[7] , C=0 , 4+5+0=9 , A->9 , C->0
[8] + [3] , A=[9,7] , C=0 , 8+3+0=11 , A->1 , C->1
[] + [] , A=[1,9,7] C=1 , END
</scratch>
1 1 9 7
```

### Case 2: Giving number of digits

```
Input:
8465+3541
Target:
<scratch>
[8,4,6,5] has 4 digits.
[3,5,4,1] has 4 digits.
[8,4,6] + [3,5,4] , A=[] , C=0 , 6+4+0=10 , A->0 , C->1
[8,4] + [3,5] , A=[0] , C=1 , 4+5+1=10 , A->0 , C->1
[8] + [3] , A=[0,0] , C=1 , 8+3+1=12 , A->2 , C->1
[] + [] , A=[2,0,0] C=1 , END
</scratch>
1 2 0 0
```

### Case 3: Giving 1 intermediate steps

```
Input:
8465+3541
Target:
<scratch>
[8,4,6,5] has 4 digits.
[3,5,4,1] has 4 digits.
[8,4,6,5] + [3,5,4,1] , A=[] , C=0 , 5+1+0=6 , A->6 , C->0
[8,4] + [3,5] , A=[6] , C=0 , 4+5+0=9 , A->9 , C->0
[8] + [3] , A=[9,6] , C=0 , 8+3+0=11 , A->1 , C->1
[] + [] , A=[1,9,6] C=1 , END
</scratch>
1 1 9 6
```

### Case 4: Giving all but one intermediate steps

```
Input:
8465+3541
Target:
<scratch>
[8,4,6,5] has 4 digits.
[3,5,4,1] has 4 digits.
[8,4,6,5] + [3,5,4,1] , A=[] , C=0 , 5+1+0=6 , A->6 , C->0
[8,4,6] + [3,5,4] , A=[6] , C=0 , 6+4+0=10 , A->0 , C->1
[8,4] + [3,5] , A=[0,6] , C=1 , 4+5+1=10 , A->0 , C->1
[8] + [3] , A=[0,0,6] , C=1 , 8+3+1=12 , A->2 , C->1
[] + [] , A=[2,0,6] C=1 END
</scratch>
1 0 0 6
```

### Case 5: Giving all intermediate steps, all it has to do is aggregate the answer

```
Input:
8465+3541
Target:
<scratch>
[8,4,6,5] has 4 digits.
[3,5,4,1] has 4 digits.
[8,4,6,5] + [3,5,4,1] , A=[] , C=0 , 5+1+0=6 , A->6 , C->0
[8,4,6] + [3,5,4] , A=[6] , C=0 , 6+4+0=10 , A->0 , C->1
[8,4] + [3,5] , A=[0,6] , C=1 , 4+5+1=10 , A->0 , C->1
[8] + [3] , A=[0,0,6] , C=1 , 8+3+1=12 , A->2 , C->1
[] + [] , A=[2,0,0,6] C=1 END
</scratch>
1 2 0 6
```

### Case 6: Giving 1 in-context example

```
Input:
8465+3541
Target:
<scratch>
```

```
[8,4,6,5] has 4 digits.
[3,5,4,1] has 4 digits.
[8,4,6,5] + [3,5,4,1] , A=[] , C=0 , 5+1+0=6 , A->6 , C->0
[8,4,6] + [3,5,4] , A=[6] , C=0 , 6+4+0=10 , A->0 , C->1
[8,4] + [3,5] , A=[0,6] , C=1 , 4+5+1=10 , A->0 , C->1
[8] + [3] , A=[0,0,6] , C=1 , 8+3+1=12 , A->2 , C->1
[] + [] , A=[2,0,0,6] C=1 , END
</scratch>
1 2 0 0 6
Input:
1946+3598
Target:
<scratch>
[1,9,4] has 3 digits.
[3,5,8] has 3 digits.
[1,9,4] + [3,5,8] , A=[] , C=0 , 4+8+0=12 , A->2 , C->1
[1,9] + [3,5] , A=[2] , C=1 , 9+5+1=15 , A->5 , C->1
[1] + [3] , A=[5,2] , C=1 , 1+3+1=5 , A->5 , C->0
[] + [] , A=[5,5,2] C=0 , END
</scratch>
5 5 2
```

**Case 7: Giving 1 In-context example, and all intermediate steps**

```
Input:
8465+3541
Target:
<scratch>
[8,4,6,5] has 4 digits.
[3,5,4,1] has 4 digits.
[8,4,6,5] + [3,5,4,1] , A=[] , C=0 , 5+1+0=6 , A->6 , C->0
[8,4,6] + [3,5,4] , A=[6] , C=0 , 6+4+0=10 , A->0 , C->1
[8,4] + [3,5] , A=[0,6] , C=1 , 4+5+1=10 , A->0 , C->1
[8] + [3] , A=[0,0,6] , C=1 , 8+3+1=12 , A->2 , C->1
[] + [] , A=[2,0,0,6] C=1 , END
</scratch>
1 2 0 0 6
Input:
1946+3598
Target:
<scratch>
[1,9,4,6] has 4 digits.
[3,5,9,8] has 4 digits.
[1,9,4,6] + [3,5,9,8] , A=[] , C=0 , 6+8+0=14 , A->4 , C->1
[1,9,4] + [3,5,9] , A=[4] , C=1 , 4+9+1=14 , A->4 , C->1
[1,9] + [3,5] , A=[4,4] , C=1 , 9+5+1=15 , A->5 , C->1
[1] + [3] , A=[5,4,4] , C=1 , 1+3+1=5 , A->5 , C->0
[] + [] , A=[5,5,4,4] C=0 , END
</scratch>
5 5 4
```

Figure 30: Example results on the model's output when prompted with a larger number of digits than those it was trained on.

# I EXPERIMENTAL SETUP

In this section, we summarize the datasets, models and hyperparameters used for experiments. All of our experiments on NanoGPT and GPT-2 models are run using PyTorch 2.1 and CUDA 11.7 on Nvidia 2808 TIs and NVIDIA 3090s. Detailed dependencies are provided on our github repository[4].

## I.1 DATASET

In this section, we explain the details of the datasets used for our experiments. For arithmetic tasks, we construct our own datasets as described below while we use the standard shakespeare (Karpathy, 2015) dataset for text.

**Arithmetic Tasks** As mentioned above, for all arithmetic tasks, we prepare our own datasets. We refer to the training dataset for a binary operator $f(\cdot)$ as $\mathcal{D}_{\text{train}} = \{(x_i^1, x_i^2), y_i\}_{i=1}^N$ where $y_i = f(x_i^1, x_i^2)$. Similarly, the test dataset $\mathcal{D}_{\text{test}}$ is constructed by randomly sampling pairs of operands that do not appear in $\mathcal{D}_{\text{train}}$. During both training and inference, we then apply different formatting techniques (see Section 3), to construct the final sequence that is input to the model. We would like to repeat that both the careful choice of samples in the training dataset as well as their formatting play a crucial role in the final performance of the model.

**Text** For text data, we use the Shakespeare dataset which was introduced by Karpathy (2015) originally featured in the blog post "The Unreasonable Effectiveness of Recurrent Neural Networks". It consists of 40,000 lines of dialogue carefully curated from William Shakespeare's plays. The dataset comprises of a total of 1,115,394 characters and 64 unique tokens(when using the character-level tokenizer that we employed in all NanoGPT experiments).

### I.1.1 DATA BALANCING

As mentioned in Section 3, we carefully sample our data to ensure that they are "*balanced*" with respect to the number of carries and number of digits. As mentioned earlier, sampling the operands uniformly at random would lead to an extremely skewed dataset. To avoid this, we try to **(i) Balance digits** by sampling lower-digit numbers with higher weights and **(ii) Balance carry-ons** by sampling such that we have equal number of examples with 0, 1, 2 and 3 carry-on operations.

Specifically, we create a balanced dataset of $10,000$ samples. This dataset includes all 100 1-digit additions and a random sampling of 900 2-digit additions (including both $(2+1)$ and $(1+2)$ digit additions) and $9,000$ 3-digit additions. For the 3-digit addition samples, we employ *rejection sampling* to ensure an equal distribution of carry-ons $(0, 1, 2, \text{or } 3)$. For the test dataset, we uniformly sample $10,000$ addition examples that do not overlap with the train dataset. Results in Figure 2 and Table 13 demonstrate a clear advantage of the employed data balancing methods.

For the train dataset, we follow a specific approach based on the number of examples. For sample sizes smaller than $10,000$ (*e.g.*, $500, 1,000, 2,000, 3,000, 4,000, 5,000$), we include all 1-digit additions and a proportionate number of 2-digit samples (*e.g.*, for a total of $5,000$ samples, we include $900 \times 5,000/10,000 = 450$ two-digit additions). The remaining samples are filled with 3-digit additions from the constructed train dataset of 10,000 samples. For sample sizes larger than 10,000 (*e.g.*, 20,000, 40,000), we include all examples from the 10,000-sample train dataset and then add additional samples as needed. Similar to before, we perform rejection sampling to maintain an equal number of carry operations. Table 14. provides detailed information on the number of samples with 1-digit, 2-digit, and 3-digit additions, as well as the number of carry-ons.

For the other arithmetic operations (subtraction, multiplication, sine, and square root), we construct the train dataset using the following approach: (i) For subtraction, we use the same pairs of operands that were used for addition. (ii) For multiplication, we include all 100 cases of a 1-digit number multiplied by a 1-digit number. Additionally, we randomly sample multiplications involving operands of up to 2 digits. (iii) For sine, we sample a random number in $[\pi/2, \pi/2]$ and truncate it to 4 decimal places. (iv) For square root, we sample a random number between $[1, 10]$ and truncate it to 4 decimal

---

[4]https://github.com/lee-ny/teaching_arithmetic

places. For the test dataset, we sample $10,000$ data points ($7,000$ for multiplication) that do not overlap with the train dataset.

Table 13: Performance of addition on various data sampling methods used: (i) Random - uniform sampling of operands; (ii) Balanced digits - sampling more 1 and 2-digit operations ; (iii) Balanced carry - balancing the dataset to contain an equal number of carry-on operations. Experiments on addition with zero-padding each operand and output to have 3 and 4 digits, respectively. We observe that balancing the dataset can significantly improve the performance or arithmetic operations.

| Data Sampling | Overall | 1-digit | 2-digit | Carry-0 | Carry-1 | Carry-2 | Carry-3 |
|---|---|---|---|---|---|---|---|
| Random | 97.74 | 98.00 | 96.20 | 95.88 | 98.61 | 98.74 | 94.98 |
| Balanced Digits | 98.13 | **100.00** | **99.70** | **98.87** | **98.64** | 98.13 | 95.93 |
| Balanced Carry-Ons | **98.29** | **100.00** | **99.70** | 98.38 | 97.56 | **99.02** | **98.22** |

Table 14: Number of examples of digit $1/2/3$ and $0/1/2/3$ carry-ons for NanoGPT experiments on addition for different number of samples varying from 500 to $40,000$.

| Total number | 1-digit | 2-digit | 3-digit | 0-carry-ons | 1-carry-ons | 2-carry-ons | 3-carry-ons |
|---|---|---|---|---|---|---|---|
| 500 | 100 | 45 | 355 | 163 | 141 | 97 | 99 |
| 1000 | 100 | 90 | 810 | 283 | 268 | 236 | 213 |
| 2000 | 100 | 180 | 1720 | 535 | 502 | 481 | 482 |
| 3000 | 100 | 270 | 2630 | 781 | 782 | 748 | 689 |
| 4000 | 100 | 360 | 3540 | 1020 | 1016 | 958 | 1006 |
| 5000 | 100 | 450 | 4450 | 1279 | 1271 | 1229 | 1221 |
| **10000** | **100** | **900** | **9000** | **2500** | **2500** | **2500** | **2500** |
| 20000 | 121 | 1937 | 17942 | 5000 | 5000 | 5000 | 5000 |
| 40000 | 132 | 3939 | 35929 | 10000 | 10000 | 10000 | 10000 |

### I.1.2 DATA FORMATTING

For each of the four formatting techniques, as applied to each arithmetic operation we provide the details below. **(i) Plain** refers to the simplest formatting where we simply create a sequence as the mathematical representation of the corresponding operation (*e.g.*, $A_3A_2A_1 + B_3B_2B_1 = C_3C_2C_1$). For **(ii) Reverse**, we simply reverse the digits of the output so that they appear in increasing order from LSB to MSB (*e.g.*, $\$A_3A_2A_1 + B_3B_2B_1 = C_1C_2C_3\$$). **(iii) Simplified Scratchpad** and **(iv) Detailed Scratchpad** provide algorithmic reasoning steps like (Nye et al., 2021; Zhou et al., 2022b) so as to help the model get more "information" per sample. Our intuition is that this approach nudges the model towards actually learning the algorithm of addition or subtraction rather than merely trying to fit the training examples. Refer to Appendix J for detailed examples of data formatting for each arithmetic operation.

---

### Different data formatting methods for addition

Four input formatting methods used for the addition task:
**(i) Plain**: standard formatting of addition
**(ii) Reverse**: flips the order of the output and encapsulates each data sample with the '$' symbol at the start and end.
**(iii) Simplified Scratchpad**: provides carry and digit-sum information for each step of addition, from the LSB to the MSB[5].
**(iv) Detailed Scratchpad**: provides explicit details of intermediate steps of addition.

| Plain | Detailed Scratchpad |
|---|---|

**Plain**

```
128+367=495
```

**Reverse**

```
$128+367=594$
```

**Simplified Scratchpad**

```
Input:
128+367
Target:
A->5 , C->1
A->9 , C->0
A->4 , C->0.
495
```

**Detailed Scratchpad**

```
Input:
128+367
Target:
<scratch>
[1,2,8] has 3 digits.
[3,6,7] has 3 digits.
[1,2,8] + [3,6,7] , A=[] , C=0 , 8+7+0=15 , A->5 , C->1
[1,2] + [3,6] , A=[5] , C=1 , 2+6+1=9 , A->9 , C->0
[1] + [3] , A=[9,5] , C=0 , 1+3+0=4 , A->4 , C->0
[] + [] , A=[4,9,5] C=0 , END
</scratch>
4 9 5
```

Figure 31: The four input formatting methods used for the addition task. We progressively increase the amount of detail with each format.

Note that we wrap each data sample in the reverse format with the '$' symbol at the beginning and end as a delimiter. We originally observed improved performance in both the plain and reverse formats when the operands and outputs were zero-padded to a fixed length (*e.g.*, 3 and 4 digits, respectively, for 3-digit addition). But later realized that a single symbol can effectively replace zero-padding. While we maintain the original plain format without padding as a baseline – emphasizing the necessity for improved data formatting for efficient emergence – we incorporate the '$'-encapsulation in our modified reverse format. For further details, refer to Appendix B.1.

**Addition** ($+$).  We focus on additions of positive numbers up to 3-digits, in which the plain formatting would look like $A_3A_2A_1 + B_3B_2B_1 = C_3C_2C_1$. For experiments on comparing data sampling presented in Figure 2, we pad the two operands and the output with zero, to be of length 3 and 4 respectively. For all other experiments, we **do not utilize zero-padding.** For Scratchpad-based methods **(iii, iv)**, we provide the digit-wise addition (denoted as $A$) and carry-on (denoted as $C$) information for intermediate steps from the least significant bit (LSB) to the most significant bit (MSB).

**Subtraction** ($-$).  We consider subtraction of positive numbers up to 3 digits, written as $A_3A_2A_1 - B_3B_2B_1 = C_3C_2C_1$ in (i) plain formatting, and $\$A_3A_2A_1 - B_3B_2B_1 = C_1C_2C_3\$$ in (ii) reverse formatting. As with addition, scratchpad-based methods (iii, iv), present the intermediate steps of digit-wise subtraction and handling of carry-ons. These steps proceed from the least significant bit (LSB) to the most significant bit (MSB). If the final result after computing all the digit-wise subtractions is negative, we subtract the number in the most significant bit (MSB) position multiplied by 10 to the power of (number of digits in the output - 1) from the remaining digits in the output. In Section B.4, we present an alternative version of the detailed scratchpad formatting for subtraction.

**Multiplication** ($\times$).  We consider multiplication of positive numbers up to 2-digits. (i) Plain formatting examples are formatted as $A_2A_1 * B_2B_1 = C_4C_3C_2C_1$, while (ii) reverse formatting is

---

[5]We deviate from the strict definition of "most significant bit" (MSB) and "least significant bit" (LSB), typically associated with binary numbers, and reinterpret them for the purpose of this paper as the most significant "digit" and least significant "digit", respectively.

formatted as $A_2A_1 * B_2B_1 = C_1C_2C_3C_4\$$. The (iv) detailed scratchpad method simplifies each intermediate step by conducting a series of multiplications between the first operand and each digit of the second operand, starting from the least significant bit (LSB) and moving toward the most significant bit (MSB). For each step, we multiply the result by an exponentiation of 10 corresponding to the relative digit position.

**Sine** (sin ). We consider decimal numbers within the range $[-\pi/2, \pi/2]$, truncated to 4-digit precision. (i) Plain formatting examples are formatted as $\sin(A_0.A_1A_2A_3A_4) = B_0.B_1B_2B_3B_4$. For (iv) detailed scratchpad method, we include the Taylor series expansion steps for sine, which is represented as $\sin(x) = x - \frac{1}{3!}x^3 + \frac{1}{5!}x^5 - \frac{1}{7!}x^7 + \cdots$. These intermediate steps involve exponentiation, which may not be any easier to compute than the sine operation itself.

**Square Root** ($\sqrt{}$). We consider decimal numbers within $[1, 10)$, truncated to 4-digits of precision with the format, written as sqrt$(A_0.A_1A_2A_3A_4) = B_0.B_1B_2B_3B_4$ for (i) plain formatting. For (iv) detailed scratchpad method, we enumerate each step of Newton's method to compute the square root function. The iterative formula is given by $x_n = \frac{1}{2}(x_{n-1} + \frac{x}{x_{n-1}})$, where $x_0$ is initialized as the floor of the square root value of the operand $x$. These intermediate steps involve a division operation, which can be as complex as the square root operation itself.

For evaluation of sine and square root, we classify the result $\hat{y}_i$ as correct if the absolute difference between $\hat{y}_i$ and the ground truth value $y_i$ is less than or equal to a predefined threshold $\epsilon \geq 0$.

## I.2 MODEL

For all experiments, we use a Decoder-only Transformer architecture. Specifically, we primarily use the NanoGPT model, a scaled-down variant of the GPT-2 model with half the number of self-attention layers, heads, and embedding dimension. Note that we use character-level tokenization instead of using the OpenAI's BPE tokenizer (Tiktoken) of vocabulary size 50257, making the vocabulary size significantly smaller. We use a learnable absolute positional embedding initialized randomly, following the GPT-2 model. Are results are generated using a temperature of 0.8.

In the case of arithmetic tasks performed on plain and reverse formatting, we set a context length of 256 for NanoGPT experiments. The length of a single train example falls within the range of 13 to 15, approximately. However, when conducting experiments on scratchpad formatting, we increase the context length to 1024. This adjustment allows us to accommodate more examples per batch. In the case of simplified scratchpad, the length of each train example is approximately 64, while the detailed scratchpad has a length of approximately 281. For GPT-2 experiments we fix the context length to 1024 for all experiments. See Table 15 for details on model configuration.

For experiments on fine-tuning a pretrained large language model, we use OpenAI's GPT-3 model - Ada, Curie, and Davinci.

Table 15: NanoGPT and GPT-2 model configuration

| Model | Input Formatting | Context Length | Self-Attn Layers | Num Heads | Embedding Dim |
|---|---|---|---|---|---|
| NanoGPT | Plain, Reverse | 256 | 6 | 6 | 384 |
|  | Scratchpad | 1024 | 6 | 6 | 384 |
| GPT-2 | Plain, Reverse | 1024 | 12 | 12 | 768 |
|  | Scratchpad | 1024 | 12 | 12 | 768 |

## I.3 TRAINING

Our overall experimental setup closely follows the standard training procedures for language models Karpathy (2022); Brown et al. (2020). We train models using the autoregressive loss for all tokens, including the prompts, rather than limiting the training to only the answers. In addition, unlike popular approaches for synthetic tasks, we pack multiple training examples to fill up the entire context window. We also randomly sample the starting position, allowing the model to process shifted inputs.

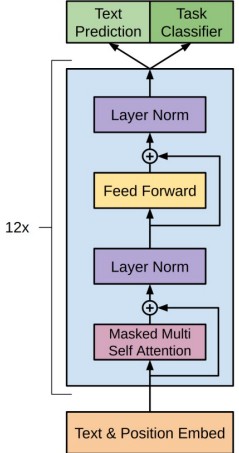

Figure 32: The GPT-2 Architecture. Image from (Radford & Narasimhan, 2018). NanoGPT model is a smaller model with half the number of self-attention layers, multi-heads, and embedding dimensions.

## I.4    HYPERPARAMETER CONFIGURATIONS

In this section, we provide a detailed overview of the hyperparameter configuration used in our experiments in Table 16 and 17. To enhance memory efficiency and training speed, we employ flash attention. For most experiments, we utilize the bfloat16 data type. However, when working with Nvidia 2080 GPUs, which do not support bfloat16, we switch to float16. It is worth noting that we did not observe significant differences in training and evaluation performance between the two data types.

The learning rate is chosen from {1e-3, 5e-4, 1e-4, 5e-5} based on validation loss. For the scratchpad format, NanoGPT is trained longer since the number of tokens per sample is higher and it requires more iterations to converge.

For the GPT-2 experimentation, we reduced the batch size to 8 to accommodate the GPU memory limitations. However, to mitigate the impact of the smaller batch size, we employed gradient accumulation steps. This approach involves taking multiple steps between gradient updates, effectively increasing the *effective* batch size to 64. For specific hyperparameter details, please refer to Table 17.

Table 16: Hyper Parameters used for NanoGPT experiments on arithmetic tasks

| Input Format | Batch Size | Optimizer | LR | Betas | Iterations | Warmup Iter | Wt decay | Dropout |
|---|---|---|---|---|---|---|---|---|
| Plain, Reverse | 256 | AdamW | 0.001 | $(0.9, 0.99)$ | 5000 | 100 | 0.1 | 0.2 |
| Scratchpad | 16 | AdamW | 0.001 | $(0.9, 0.99)$ | 50000 | 0 | 0.1 | 0.2 |

Table 17: Hyper Parameters used for GPT-2 experiments on arithmetic tasks

| Input Format | Batch Size | Optimizer | LR | Betas | Iterations | Warmup Iter | Wt decay | Dropout |
|---|---|---|---|---|---|---|---|---|
| Plain, Reverse | 64 | AdamW | 0.0005 | $(0.9, 0.99)$ | 5000 | 100 | 0.1 | 0.2 |
| Scratchpad | 64 | AdamW | 0.0005 | $(0.9, 0.99)$ | 20000 | 0 | 0.1 | 0.2 |

Table 18: Hyper Parameters used for tandem training experiments in Section 7.

| Model | Batch Size | Optimizer | LR | Betas | Iterations | Warmup Iter | Wt decay | Dropout |
|---|---|---|---|---|---|---|---|---|
| NanoGPT | 16 | AdamW | 0.001 | $(0.9, 0.99)$ | 5000 | 0 | 0.1 | 0.2 |
| GPT-2 | 40 | AdamW | 0.0006 | $(0.9, 0.95)$ | 50000 | 2000 | 0.1 | 0.2 |

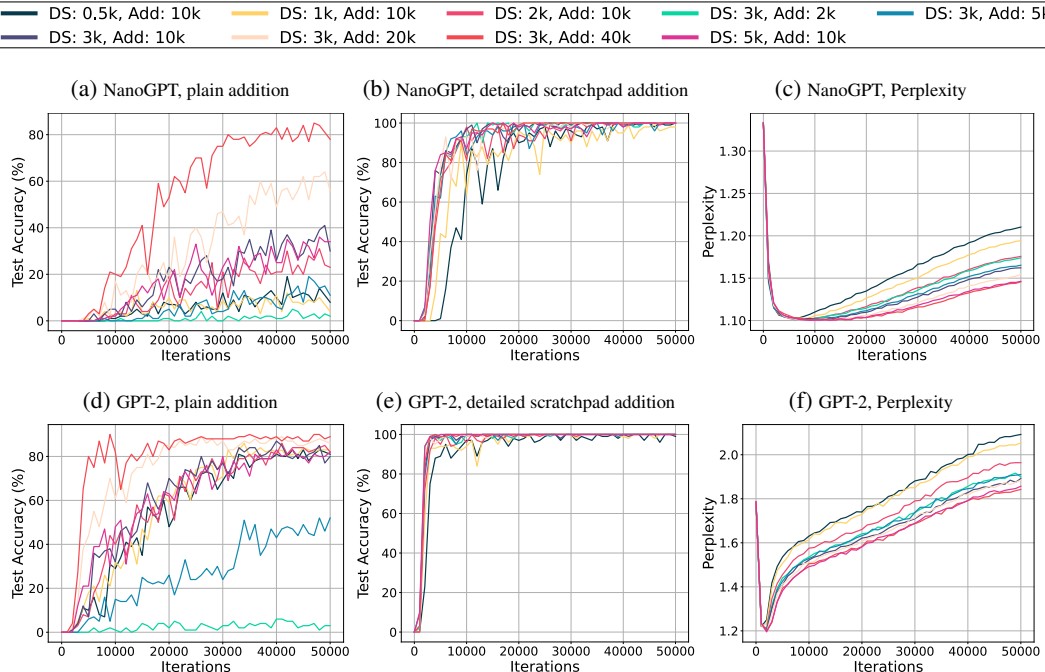

Figure 33: Training loss curves for NanoGPT and GPT-2 trained with varying numbers of plain (Add) and detailed scratchpad (DS) samples as well as the shakespeare dataset as described in Section 7. As we can see, the model continues to improve in addition accuracy as the number of iterations increases. However, the training perplexity on Shakespeare also tends to increase, which indicates some overfitting. However, we note that the model still outputs "reasonable" text when prompted with shakespeare text.

# J  PROMPT EXAMPLES

In this section, we provide three examples of each formatting (plain, reverse, simplified scratchpad, detailed scratchpad) of arithmetic operations $(+, -, \times, \sin, \sqrt{})$.

## J.1  ADDITION

---

**Addition Examples**

### Plain

```
266+738=1004
980+743=1723
41+34=75
```

### Reverse

```
$913+524=1437$
$226+598=824$
$35+58=93$
```

### Simplified Scratchpad

```
Input:
922+244
Target:
A->6 , C->0
A->6 , C->0
A->1 , C->1.
1166
Input:
285+43
Target:
A->8 , C->0
A->2 , C->1
A->3 , C->0.
328
Input:
993+849
Target:
A->2 , C->1
A->4 , C->1
A->8 , C->1.
1842
```

### Detailed Scratchpad

```
Input:
396+262
Target:
<scratch>
[3,9,6] has 3 digits.
[2,6,2] has 3 digits.
[3,9,6] + [2,6,2] , A=[] , C=0 , 6+2+0=8 , A->8 , C->0
[3,9] + [2,6] , A=[8] , C=0 , 9+6+0=15 , A->5 , C->1
[3] + [2] , A=[5,8] , C=1 , 3+2+1=6 , A->6 , C->0
[] + [] , A=[6,5,8] C=0 , END
</scratch>
6 5 8
Input:
796+890
Target:
<scratch>
[7,9,6] has 3 digits.
[8,9,0] has 3 digits.
[7,9,6] + [8,9,0] , A=[] , C=0 , 6+0+0=6 , A->6 , C->0
[7,9] + [8,9] , A=[6] , C=0 , 9+9+0=18 , A->8 , C->1
[7] + [8] , A=[8,6] , C=1 , 7+8+1=16 , A->6 , C->1
[] + [] , A=[6,8,6] C=1 , END
</scratch>
1 6 8 6
Input:
788+989
Target:
<scratch>
[7,8,8] has 3 digits.
[9,8,9] has 3 digits.
[7,8,8] + [9,8,9] , A=[] , C=0 , 8+9+0=17 , A->7 , C->1
[7,8] + [9,8] , A=[7] , C=1 , 8+8+1=17 , A->7 , C->1
[7] + [9] , A=[7,7] , C=1 , 7+9+1=17 , A->7 , C->1
[] + [] , A=[7,7,7] C=1 , END
</scratch>
1 7 7 7
```

## J.2 SUBTRACTION

### Subtraction Examples

#### Plain

```
266-738=-472
980-743=237
41-34=7
```

#### Reverse

```
$913-524=983$
$226-598=273-$
$35-58=32-$
```

#### Simplified Scratchpad

```
Input:
396-262
Target:
A->4 , C->0
A->3 , C->0
A->1 , C->0
100+34=134.
134
Input:
796-890
Target:
A->6 , C->0
A->0 , C->0
A->-1 , C->-1
-100+6=-94.
-94
Input:
788-989
Target:
A->9 , C->-1
A->9 , C->-1
A->-3 , C->-1
-300+99=-201.
-201
```

#### Detailed Scratchpad

```
Input:
396-262
Target:
<scratch>
[3,9,6] has 3 digits.
[2,6,2] has 3 digits.
[3,9,6] - [2,6,2] , A=[] , C=0 , 6-2-0=4 , A->4 , C->0
[3,9] - [2,6] , A=[4] , C=0 , 9-6-0=3 , A->3 , C->0
[3] - [2] , A=[3,4] , C=0 , 3-2-0=1 , A->1 , C->0
[] - [] , A=[1,3,4]
100+34=134 , END
</scratch>
1 3 4
Input:
796-890
Target:
<scratch>
[7,9,6] has 3 digits.
[8,9,0] has 3 digits.
[7,9,6] - [8,9,0] , A=[] , C=0 , 6-0-0=6 , A->6 , C->0
[7,9] - [8,9] , A=[6] , C=0 , 9-9-0=0 , A->0 , C->0
[7] - [8] , A=[0,6] , C=0 , 7-8-0=-1 , A->-1 , C->-1
[] - [] , A=[-1,0,6]
</scratch>
-9 4
Input:
788-989
Target:
<scratch>
[7,8,8] has 3 digits.
[9,8,9] has 3 digits.
[7,8,8] - [9,8,9] , A=[] , C=0 , 8-9-0+10=9 , A->9 , C->-1
[7,8] - [9,8] , A=[9] , C=-1 , 8-8-1+10=9 , A->9 , C->-1
[7] - [9] , A=[9,9] , C=-1 , 7-9-1=-3 , A->-3 , C->-1
[] - [] , A=[-3,9,9]
-300+99=-201 , END
</scratch>
-2 0 1
```

## J.3 MULTIPLICATION

### Multiplication Examples

| Plain | Detailed Scratchpad |
|---|---|

```
5*32=160
66*76=5016
67*74=4958
```

**Reverse**

```
$5*32=061$
$66*76=6105$
$67*74=8594$
```

```
Input:
22*52
Target:
<scratch>
[2,2] has 2 digits.
[5,2] has 2 digits.
[2,2] * 2 , A=[4,4] , k=1 , B=[4,4] , C=0+44=44
[2,2] * 5 , A=[1,1,0] , k=10 , B=[1,1,0,0] , C=44+1100=1144 , END
</scratch>
1 1 4 4
Input:
8*69
Target:
<scratch>
[8] has 1 digits.
[6,9] has 2 digits.
[8] * 9 , A=[7,2] , k=1 , B=[7,2] , C=0+72=72
[8] * 6 , A=[4,8] , k=10 , B=[4,8,0] , C=72+480=552 , END
</scratch>
5 5 2
Input:
52*34
Target:
<scratch>
[5,2] has 2 digits.
[3,4] has 2 digits.
[5,2] * 4 , A=[2,0,8] , k=1 , B=[2,0,8] , C=0+208=208
[5,2] * 3 , A=[1,5,6] , k=10 , B=[1,5,6,0] , C=208+1560=1768 , END
</scratch>
1 7 6 8
```

## J.4 SINE

### Sine Examples

| Plain | Detailed Scratchpad |
|---|---|

```
sin(1.0313)=0.8579
sin(-0.6909)=-0.6373
sin(-0.5719)=-0.5413
```

```
Input:
sin(1.0313)
Target:
<scratch>
x_0=1.0313
x_1:  x_0 - 1/3!  * (x^3) , x_1=0.8484
x_2:  x_1 + 1/5!  * (x^5) , x_2=0.8581
x_3:  x_2 - 1/7!  * (x^7) , x_3=0.8578
x_4:  x_3 + 1/9!  * (x^9) , x_4=0.8578 , END
</scratch>
0.8578
Input:
sin(-0.6909)
Target:
<scratch>
x_0=-0.6909
x_1:  x_0 - 1/3!  * (x^3) , x_1=-0.636
x_2:  x_1 + 1/5!  * (x^5) , x_2=-0.6374
x_3:  x_2 - 1/7!  * (x^7) , x_3=-0.6374
x_4:  x_3 + 1/9!  * (x^9) , x_4=-0.6375 , END
</scratch>
-0.6375
Input:
sin(-0.5719)
Target:
<scratch>
x_0=-0.5719
x_1:  x_0 - 1/3!  * (x^3) , x_1=-0.5408
x_2:  x_1 + 1/5!  * (x^5) , x_2=-0.5414
x_3:  x_2 - 1/7!  * (x^7) , x_3=-0.5414
x_4:  x_3 + 1/9!  * (x^9) , x_4=-0.5415 , END
</scratch>
-0.5415
```

## J.5 SQUARE ROOT

---

**Square Root Examples**

**Plain**

```
sqrt(7.2726)=2.6967
sqrt(3.6224)=1.9032
sqrt(1.0895)=1.0437
```

**Detailed Scratchpad**

```
Input:
sqrt(7.1042)
Target:
<scratch>
x_0=2
x_1:  1/2*(2+7.1042/2)=2.776, x_1=2.776
x_2:  1/2*(2.776+7.1042/2.776)=2.6675, x_2=2.6675
x_3:  1/2*(2.6675+7.1042/2.6675)=2.6653, x_3=2.6653
x_4:  1/2*(2.6653+7.1042/2.6653)=2.6653, x_4=2.6653 , END
</scratch>
2.6653
Input:
sqrt(6.2668)
Target:
<scratch>
x_0=2
x_1:  1/2*(2+6.2668/2)=2.5667, x_1=2.5667
x_2:  1/2*(2.5667+6.2668/2.5667)=2.5041, x_2=2.5041
x_3:  1/2*(2.5041+6.2668/2.5041)=2.5033, x_3=2.5033
x_4:  1/2*(2.5033+6.2668/2.5033)=2.5033, x_4=2.5033 , END
</scratch>
2.5033
Input:
sqrt(8.3216)
Target:
<scratch>
x_0=2
x_1:  1/2*(2+8.3216/2)=3.0804, x_1=3.0804
x_2:  1/2*(3.0804+8.3216/3.0804)=2.8909, x_2=2.8909
x_3:  1/2*(2.8909+8.3216/2.8909)=2.8847, x_3=2.8847
x_4:  1/2*(2.8847+8.3216/2.8847)=2.8847, x_4=2.8847 , END
</scratch>
2.8847
```

---

## J.6 NOISY SIMPLE SCRATCHPAD

We provide one example for each case of adding noise in the simplified scratchpad experiments discussed in Section B.5.

---

**Noisy Simple Scratchpad Examples**

```
We provide one example for each case of adding noise in the simplified
scratchpad experiments discussed in Section B.5.  The input prompt
is highlighted in light blue, while the remaining part is highlighted
in light green.  We construct the dataset to have either correct or
random digit-sum A and carry information C. For all cases, the final
answer remains accurate.
```

**Prompt:**

```
Input:
686+886
Target:
```

**Correct A & C**

```
A->2 , C->1
A->7 , C->1
A->5 , C->1.
1572
```

**Random C**

```
A->2 , C->0
A->7 , C->0
A->5 , C->1.
1572
```

**Random A**

```
A->0 , C->1
A->9 , C->1
A->9 , C->1.
1572
```

**Random A & C**

```
A->8 , C->1
A->1 , C->0
A->2 , C->1.
1572
```

---

### J.7 EXAMPLE DATA FOR GPT-3 FINE-TUNING

We provide an example from the training dataset consisting of one prompt-completion pair used for fine-tuning the GPT-3 model using OpenAI's API. The prompt is highlighted in light grey, while the completion is highlighted in light green. Note that for plain and reverse formatting, we include spacing between digits to ensure consistent tokenization of numbers. "###" is used as the stop sequence for generation.

### J.7.1 ADDITION

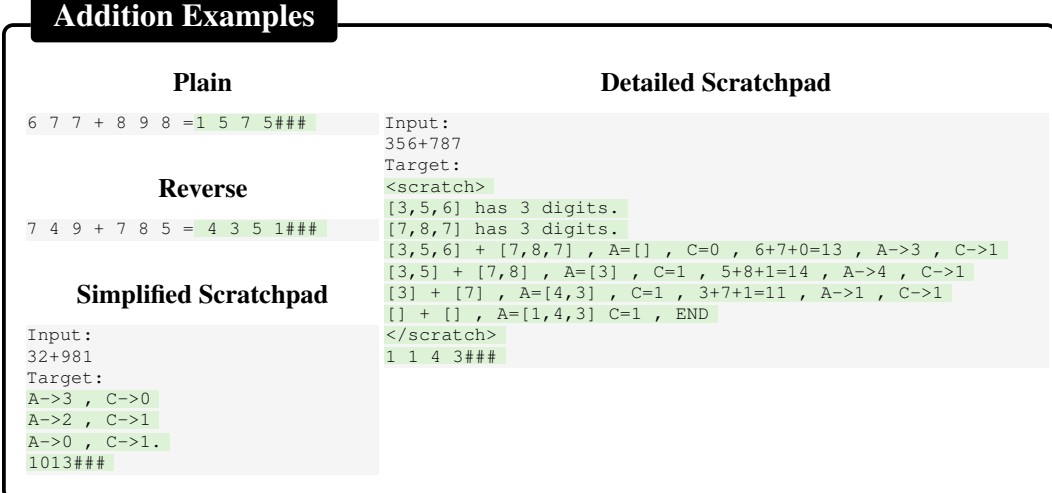

### J.7.2 SUBTRACTION

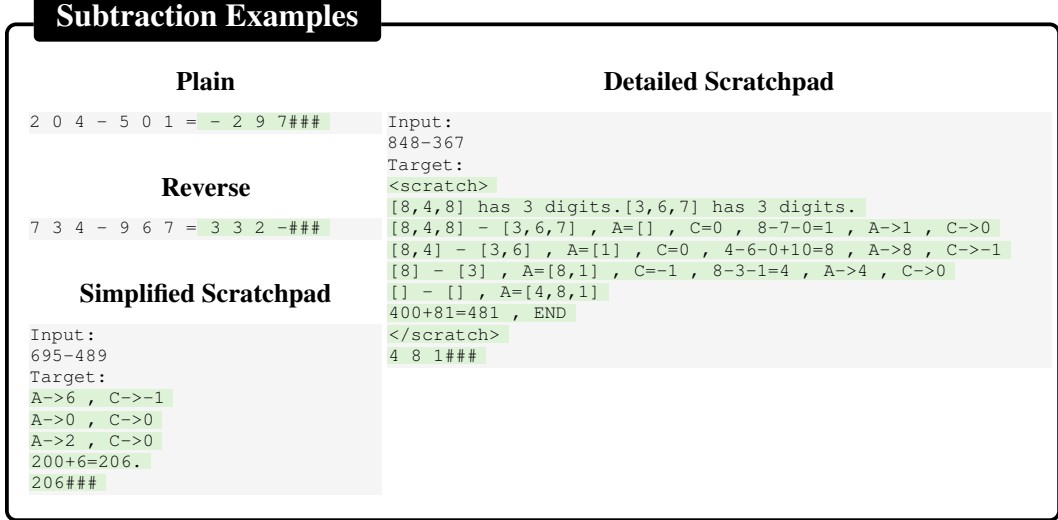

### J.7.3 SINE

**Sine Examples**

| Plain | Detailed Scratchpad |
|---|---|
| sin(-0.8649)
-0.7611### | Input:
sin(-1.3516)
Target:
x_0=-1.3516
x_1:  -1.3516 - 1/3!  * (x*x*x) , x_1=-0.9401
x_2:  -0.9401 + 1/5!  * (x*x*x*x*x) , x_2=-0.9777
x_3:  -0.9777 - 1/7!  * (x*x*x*x*x*x*x) , x_3=-0.9761
x_4:  -0.9761 + 1/9!  * (x*x*x*x*x*x*x*x*x) , x_4=-0.9762 , END
</scratch>
-0.9762### |

### J.7.4 SQUARE ROOT

**Square Root Examples**

| Plain | Detailed Scratchpad |
|---|---|
| sqrt(1.2178)
1.1035### | Input:
sqrt(5.5808)
Target:
<scratch>
x_0=2
x_1:  1/2*(2+5.5808/2)=2.3952, x_1=2.3952
x_2:  1/2*(2.3952+5.5808/2.3952)=2.3625, x_2=2.3625
x_3:  1/2*(2.3625+5.5808/2.3625)=2.3623, x_3=2.3623
x_4:  1/2*(2.3623+5.5808/2.3623)=2.3623, x_4=2.3623 , END
</scratch>
2.3623### |

