# TEACHING ARITHMETIC TO SMALL TRANSFORMERS

## ABSTRACT

Large language models like GPT-4 exhibit emergent capabilities across general-purpose tasks, such as basic arithmetic, when trained on extensive text data, even though these tasks are not explicitly encoded by the unsupervised, next-token prediction objective. This study investigates how even small transformers, trained from random initialization, can efficiently learn arithmetic operations such as addition, multiplication, and elementary functions like square root, using the next-token prediction objective. We first demonstrate that conventional training data is not the most effective for arithmetic learning, and simple formatting changes can significantly improve accuracy. This leads to sharp phase transitions as a function of training data scale, which, in some cases, can be explained through connections to low-rank matrix completion. Building on prior work, we then train on chain-of-thought style data that includes intermediate step results. Even in the complete absence of pretraining, this approach significantly and simultaneously improves accuracy, sample complexity, and convergence speed. We also study the interplay between arithmetic and text data during training and examine the effects of few-shot prompting, pretraining, and parameter scaling. Additionally, we discuss the challenges associated with length generalization. Our work highlights the importance of high-quality, instructive data that considers the particular characteristics of the next-word prediction loss for rapidly eliciting arithmetic capabilities.[1]

## 1 INTRODUCTION

Large language models like GPT-3/4, PaLM, LaMDA (Brown et al., 2020; Chowdhery et al., 2022; Thoppilan et al., 2022) have demonstrated general-purpose properties, often referred to as *emergent abilities* (Wei et al., 2022a), for a wide range of downstream tasks like language and code translation, compositional reasoning, and basic arithmetic operations (Webb et al., 2022; Nye et al., 2021; Wei et al., 2022b; Shi et al., 2022; Wang et al., 2022; Srivastava et al., 2022; Chen et al., 2023). What is perhaps surprising, is that these tasks are not explicitly encoded in the model's training objective, which typically is an auto-regressive, next-token-prediction loss.

Prior research has delved into exploring these capabilities and how they emerge as the scale and of training compute, type of data, and model size vary (Wei et al., 2022a; Chung et al., 2022; Tay et al., 2022). Untangling the factors, however, remains challenging due to the data complexity and the variety of tasks examined. Driven by the curiosity to understand the factors that elicit these capabilities in next-token predictors, we set out to pinpoint the key contributors that accelerate the emergence of such abilities. These contributors may include the format and scale of data, model scale, the presence of pre-training, and the manner of prompting.

To provide a more precise examination of these factors, our study is conducted in a controlled setting: we first focus on teaching arithmetic to small decoder-only transformer models, such as NanoGPT and GPT-2, when trained from random initialization. Starting with a model of 10.6M parameters and scaling up to 124M parameters, we use the standard autoregressive next-token prediction loss. Our objective is to understand if and to what degree these models can efficiently learn basic arithmetic operations like addition, subtraction, multiplication, square root, and sine, thereby providing a clearer lens through which to view the elicitation of emergent abilities. Below, we summarize our findings.

**Data format and sampling plays a significant role.** We first observe that teaching a model addition (or any other operation) using standard addition samples, *i.e.*, '$A_3A_2A_1 + B_3B_1B_1 = C_3C_2C_1$', is

---

[1]Our code is available at `https://anonymous.4open.science/r/nanoGPT-25D2`

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

**Teaching arithmetic operations beyond addition.** While our experiments so far were primarily focused on *addition*, we include other arithmetic operations to demonstrate the broader applicability of our insights. We consider a mix of arithmetic tasks – *subtraction*, *multiplication*, *sine*, and *square root*. Each operation entails its unique challenges and intricacies. For instance, subtraction introduces the concept of negative numbers, multiplication can generate significantly longer outputs, and sine and square root functions entail computations involving floating-point numbers.

The results depicted in Figure 6 indicate that similar to the findings of addition, the detailed scratchpad format significantly improves performance over *plain* or *reverse* formats and yields efficient results even with few samples for subtraction and multiplication tasks. Interestingly, we find *reverse* is not particularly effective in multiplication. On the other hand, the detailed scratchpad format exhibits reduced efficiency for $\sin$ and $sqrt$ compared to other operations. This discrepancy can be traced back to the complexity of the intermediate steps involved in the detailed scratchpad. While addition, subtraction, and multiplication are decomposed into simpler functions, sine and square root involve more intricate operations. See Appendix B.5 for a broader analysis of the error profile, and Appendix D for detailed experimental setup and results on jointly training on all five arithmetic tasks.

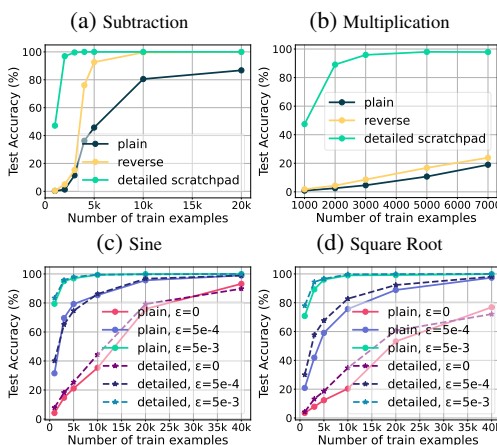

Figure 6: Performance of 3-digit subtraction, 2-digit multiplication, 4-digit precision sine and square root with varying data formats.

**Mixing Text with Arithmetic Data.** While the models so far were trained exclusively on arithmetic tasks, in practice, LLMs utilize a combination of arithmetic and *text* data for training. How does that affect the emergence of arithmetic skills? To explore that we incorporate both addition samples and text into our train data and evaluate the models with few-shot prompting (showing a few examples of addition in the prompt) to see if it is able to be effectively conditioned for the appropriate context (arithmetic/text generation). As we see in Figure 7, we find that few-shot prompting improves the performance of the model, allowing it to perform addition accurately even in the plain format.

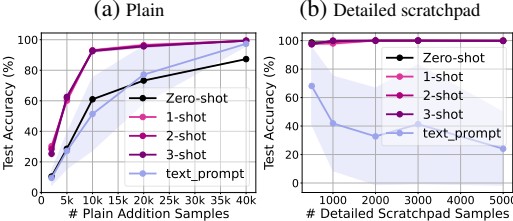

Figure 7: Performance of NanoGPT model trained with the Shakespeare dataset, addition dataset in plain, and detailed scratchpad format. The number of plain (left) and detailed scratchpad (right) formatted samples are varied. Performance is evaluated on zero-shot, few-shot, and text prompts, with the shaded area representing the standard deviation across various prompt exemplar sets.

Intriguingly, accuracy remains high using *plain* even with the inclusion of a text prompt preceding "A+B=". This is likely due to the structure of our mixed dataset where addition examples are interspersed within Shakespeare text. With the incorporation of more addition examples, instances where addition follows Shakespeare text increases, leading to a decrease in potential inconsistencies when text content is present during addition test queries. We further analyze the effect of text on prompting for both cases with and without text in the training data in Appendix E.

## 8 FINE-TUNING, SCALING, AND PRETRAINING IN LARGER MODELS

We extend our study from NanoGPT to larger models like GPT-2 and GPT-3 to explore the impact of pretraining and model size. Initially, we compare the performance of NanoGPT and GPT-2, both trained from scratch. This highlights the advantages of larger model scales, especially in zero-shot scenarios. We then fine-tune a pretrained GPT-3 on various arithmetic tasks using different data formats, reaffirming the importance of data formatting for larger pretrained models.

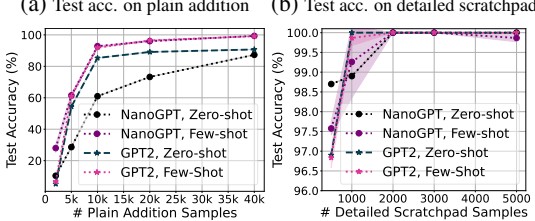

Figure 8: Comparing NanoGPT and GPT-2 trained jointly on the Shakespeare dataset and addition tasks using plain and detailed scratchpad formatting. Larger model sacale and using few-shot prompting enhances performance.

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

 "unique" tokens encountered during training. Each data sample is treated as a set of unique tokens, and the number of unique tokens is derived by *multiplying the number of samples with the tokens per sample*. For instance, the mean token count for a single training example in a 3-digit addition task is 13 for plain format, 15 for reverse format, 64 for simplified scratchpad format, and 281 for detailed scratchpad format. Note that this calculation does not evaluate uniqueness of tokens across samples *i.e.*, if the first sample is "112 + 129 = 241" and the second sample is "112 + 128 = 240", we will still consider that the model has seen 26 unique tokens even though only two tokens differ across samples.

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

2.3623### |