# OpenReview forum: "Teaching Arithmetic to Small Transformers"
_ICLR.cc/2024/Conference — ICLR 2024 poster_

### Official Review · Reviewer_twgj · 2023-10-25

**Soundness:** 3 good
**Presentation:** 3 good
**Contribution:** 1 poor
**Rating:** 6
**Confidence:** 3

**Summary:**

This paper investigates why arithmetic capabilities emerge from next-word prediction. The authors do several experiments with Transformer models between 10.6M and 124M parameters trained from scratch, and GPT-3 fine-tuned. The findings for the models trained from scratch is that for arithmetic they can generalise from relatively few examples (~6k), but generalise better when the output is reversed because this allows learning a simpler algorithm for arithmetic. The authors also find that holding out entire numbers from training doesn't decrease performance. Using scratchpads during training makes the models more sample efficient (even when accounting for the extra tokens this costs). The findings are similar for arithmetic with up to 10 digits and for different functions like subtraction. The authors also mix in text during training, and the produced model gets 100% arithmetic performance with few-shot prompting. In the third part the authors fine-tune GPT-3 and find further improved performance and sample-efficiency.

**Strengths:**

The paper is written well, a lot of experiments are done with some interesting findings (e.g. scratchpad training from scratch is more efficient even when accounting for extra tokens, holding out digits doesn't degrade performance).

**Weaknesses:**

The introduction says that the novelty of this work over prior work is to pinpoint the factors that contribute to the fast emergence of arithmetic capabilities through careful ablations, but I'm not sure how well the findings from this study would transfer to LLMs at scale. Additionally, I feel like some of the findings are not properly backed by the experiments:

- The Transformers in this work are almost exclusively trained on arithmetic data, and it's unclear whether the findings would transfer to models trained primarily on language with some arithmetic data weaved in. Indeed, you show that arithmetic can be learned from next-digit prediction, but that is unsurprising since the setup is simply a cross-entropy loss on the next "token"(=digit).

- The finding that reversing the output helps also most likely doesn't explain emergent abilities in LLMs, as this doesn't happen in natural text.

- You show that few-shot prompting can be used to improve performance when text is mixed in, but you don't try few-shot prompting when text is not mixed in, so it's unclear whether the increase in performance is due to text being mixed in, or simply because of few-shot prompting. Few-shot prompting should also work when you train the model without text, or am I missing something?

- The finding that sampling strategy is important is interesting, but on its own it's again unclear whether that transfers to emergent capabilities of larger models. For example, does the distribution of numbers in pretraining dataset follow a similar one as the one produced by your sampling strategy?

- You claim you try to disentangle the factor of pretraining, but compare NanoGPT and GPT-2 from scratch to a fine-tuned GPT-3, which doesn't disentangle pretraining from scale. Why not compare to GPT-2 fine-tuned?

In summary, I feel like the motivation of the study (to explain emergent arithmetic abilities of LLMs) does not match with the experiments.

**Questions:**

- I don't understand why you would call the learning curve between 1k and 4k examples a phase transition? You're learning a task with a deterministic output, and 100% accuracy is possible and this model happens to learn it between 1 and 4k examples. This doesn't seem to be a fast phase transition when the model is anyway only trained for 6k examples.

- It seems like the main reason for the model to learn arithmetic in this setup is because every example it sees is an entirely new one (new pair of operands), so the only strategy to achieve low training loss is actual arithmetic. It would be interesting to learn whether the generalisation behaviour is different when you train on the same set of examples for multiple epochs, which in turn could in fact explain some of the emergence of arithmetic in LLMs who are often trained for only a single epoch

- I'm not sure I understand the exclusion experiments; For example for 1st (LSD) digit exclusion, do you just exclude 5 in the 1st digit? meaning that something like 543 cannot occur, but 453 can? If yes, how is that exclusion? The numbers are represented as a sequence over absolute embeddings, so 5 is not excluded at all but simply held-out at a particular position in the number no?

- Typo end of page 8 (textand should be text and)

---

> ### Author Response · Authors · 2023-11-21
> **Response to Reviewer twgj (1/2)**
>
> Thank you for your valuable and overall positive feedback on our paper. We have responded to your concerns below and hope that this will convince you to elevate your score.
>
> > `scalability of the findings` The introduction says that the novelty of this work over prior work is to pinpoint the factors that contribute to the fast emergence of arithmetic capabilities through careful ablations, but I'm not sure how well the findings from this study would transfer to LLMs at scale. Additionally, I feel like some of the findings are not properly backed by the experiments:
>
> We apologize that this was not clear. We have a full table on scale ranging from NanoGPT, GPT2 and up to GPT3 models in Section 8 and also some additional experiments in Section F showing that most of the findings do indeed carry over when we scale up the models.
>
> > `transferability to language data` The Transformers in this work are almost exclusively trained on arithmetic data, and it's unclear whether the findings would transfer to models trained primarily on language with some arithmetic data weaved in. Indeed, you show that arithmetic can be learned from next-digit prediction, but that is unsurprising since the setup is simply a cross-entropy loss on the next "token"(=digit).
>
> While we agree that the loss is simple CE loss on next "token", it is unclear whether this is sufficient to learn the underlying algorithm that is necessary for performing arithmetic. Note that arithmetic in the token space is not equivalent to arithmetic in the embedding space. Moreover, this does not explain generalization to unseen examples of arithmetic. It is this aspect that is surprising. In fact, what is even more surprising is that without using careful data formatting and sampling, NanoGPT is unable to learn arithmetic with a reasonable number of samples.
>
> Furthermore, we have conducted experiments where arithmetic tasks are intermingled with textual data (refer to Section 7 and Appendix E). Although these experiments are on a small scale, the results suggest that our findings are transferable to a setting with the presence of text, and that few-shot promptining in these scenarios enhances the performance.
>
> > `practicality of reverse format` The finding that reversing the output helps also most likely doesn't explain emergent abilities in LLMs, as this doesn't happen in natural text.
>
> While we agree that reversing the output is not something that occurs in natural text, we use it as a mechanism to describe the importance of data formatting in learning compositional functions. Reversing the output is meant as an illustration of how even simple data formatting changes can elicit large improvements in performance.
>
> > `few-shot prompting on no-text model` You show that few-shot prompting can be used to improve performance when text is mixed in, but you don't try few-shot prompting when text is not mixed in, so it's unclear whether the increase in performance is due to text being mixed in, or simply because of few-shot prompting. Few-shot prompting should also work when you train the model without text, or am I missing something?
>
> Please refer to [Figure 21](https://imgur.com/GysI4fO) where we have results on few-shot prompting on models trained exclusively on text data. As the reviewer expects, few-shot prompting does work even when the model is trained without text.
>
> > `trasnferring the data sampling strategy` The finding that sampling strategy is important is interesting, but on its own it's again unclear whether that transfers to emergent capabilities of larger models. For example, does the distribution of numbers in pretraining dataset follow a similar one as the one produced by your sampling strategy?
>
> Is the reviewer referring to typical pretraining datasets such as C4? If so, then it is difficult to even extract the distribution of numbers from it due to scale. However, we would expect that it does not since the arithmetic performance of most typical LLMs trained on C4 and the Pile is quite poor. We hope that our work will encourage more research in the area of dataset creation and curation. As seen in Gunasekar et al. "Textbooks are all you need", high quality tokens are becoming more and more important in order to train models efficiently.

---

> > ### Author Response · Authors · 2023-11-21
> > **Response to Reviewer twgj (2/2)**
> >
> > > `GPT-2 fine-tuned` You claim you try to disentangle the factor of pretraining, but compare NanoGPT and GPT-2 from scratch to a fine-tuned GPT-3, which doesn't disentangle pretraining from scale. Why not compare to GPT-2 fine-tuned?
> > In summary, I feel like the motivation of the study (to explain emergent arithmetic abilities of LLMs) does not match with the experiments.
> >
> > We hope that the reviewer's concern regarding the comparison between NanoGPT, GPT-2 from scratch, and a fine-tuned GPT-3 is addressed in [Figure 25](https://imgur.com/xVCWPSl), where we present results for both NanoGPT and GPT-2 models initialized from scratch and with pretraining. Additionally, we use the TikToken tokenizer for GPT-2 to match GPT-3's tokenization approach. This comparison aims to isolate the effects of pretraining from model scale. We will emphasize these details in the revised manuscript.
> >
> > > `Number of samples in training data` It seems like the main reason for the model to learn arithmetic in this setup is because every example it sees is an entirely new one (new pair of operands), so the only strategy to achieve low training loss is actual arithmetic. It would be interesting to learn whether the generalisation behaviour is different when you train on the same set of examples for multiple epochs, which in turn could in fact explain some of the emergence of arithmetic in LLMs who are often trained for only a single epoch
> > >  I don't understand why you would call the learning curve between 1k and 4k examples a phase transition? You're learning a task with a deterministic output, and 100% accuracy is possible and this model happens to learn it between 1 and 4k examples. This doesn't seem to be a fast phase transition when the model is anyway only trained for 6k examples.
> >
> > We would like to clarify that for all experiments, we are training on a fixed set of example. The sample efficiency curve shows the performance varies when training for multiple epochs on training datasets of varying sizes. The term "phase transition" is used to describe the qualitative shift in the model's capability to perform addition—from an inability to grasp addition with an insufficient number of examples to learning perfect addition as the number of samples increase.
> >
> > Therefore, we refer to learning addition perfectly between 1k and 4k examples as a phase transition since the model goes from $0\%$ accuracy to almost $100\%$ in this range. We would also like to emphasize that this is an extremely small fraction of all possible 3-digit combinations. We stop at 6k since there is no point in training with more samples once the model has learned addition perfectly. However, the "phase transition" would look sharper if we extended the x-axis to the $10^6$ possible samples.
> >
> > > `unseen digits` I'm not sure I understand the exclusion experiments; For example for 1st (LSD) digit exclusion, do you just exclude 5 in the 1st digit? meaning that something like 543 cannot occur, but 453 can? If yes, how is that exclusion? The numbers are represented as a sequence over absolute embeddings, so 5 is not excluded at all but simply held-out at a particular position in the number no?
> >
> > That is exactly right. "Exclusion" here refers to its absence from a particular ordinal position, not its complete removal from the dataset. This approach allows us to investigate the model's ability to generalize arithmetic operations on digits from one position to another where it has never seen it. It is important to note that if a digit is completely excluded from the training data, the model cannot learn the representation of that digit and even its embedding would be random.

---

> > > ### Comment · Reviewer_twgj · 2023-11-22
> > > **Thanks for your response**
> > >
> > > Thank you for your responses. Although some of my concerns are adequately addressed (namely that some findings are not backed by experiments), my concern that these findings do not transfer to realistic settings remains. For example, you say yourself that it's unlikely that the datasets used for training models at scale have the right distribution, and your experiments do show that the right distribution is important for generalisation. I will increase my soundness score because the concerns regarding claims are mostly addressed, and with that my rating to a 5, but I still cannot recommend acceptance because of the claim that this paper looks at why arithmetic emerges at scale.

---

> > > > ### Author Response · Authors · 2023-11-22
> > > > **Transferability of our Results**
> > > >
> > > > Thank you for your feedback and for reconsidering some aspects of our work. We understand your concerns regarding the transferability of our findings to realistic settings and the emergence of arithmetic at scale. We have worked hard to conduct extensive experiments and provide insights into arithmetic learning in language models and would appreciate the opportunity to improve our manuscript within this review cycle. We hope that our work can be considered for ICLR, as we believe it is timely and relevant to the community. We are committed to addressing your concerns and are open to modifying our manuscript accordingly to ensure its scientific rigor and relevance.
> > > >
> > > > First, while it is true that the datasets used for training models at scale may not have exactly the same distribution we used in our experiments, our work highlights the importance of data formatting, sampling, and the inclusion of chain of thought steps for fast arithmetic learning. In fact, the improved abilities of the newer models such as GPT-4, Llama-2 highlight that, (as many suspect) these models are likely trained on carefully curated data. Moreover, with the growing token sparsity and prohibitively expensive training runs, it is essential that we choose the data as carefully as possible so as to ensure that these models achieve high quality with as few samples as possible. We also believe that the insights gleaned from our work can be applied to improve finetuning strategies and guide future research in synthetic dataset creation and curation, which has been an area of high interest in the related literature such as Gunasekar et al. "Textbooks are all you need".
> > > >
> > > > Second, we believe that our connection to matrix completion can help explain the sharpness of the phase transition observed in our experiments. Although our data may not resemble real-world datasets exactly, the core principles and mechanisms we uncover can still provide valuable insights into the emergence of arithmetic capabilities in large language models.
> > > >
> > > > We understand that some of our hypotheses may seem overreaching, and we are more than willing to tone down any claims that are not well-supported by the evidence. We would greatly appreciate any specific guidance from the reviewer on which parts of the manuscript should be modified to better align with the presented results.

---

> > > > > ### Comment · Reviewer_twgj · 2023-11-23
> > > > > **Thanks for the response**
> > > > >
> > > > > I’ve decided to increase my score, also after reading the other reviewers responses and rebuttals, because I agree that if you take the title of the paper at face value, the paper does exactly what it says it does, and additionally has insights about how exactly this can be done. I remain unclear on how much this explains anything related to LLMs, but I leave it to the authors to decide on where and if at all to tone down claims regarding that, or caveat that. I also remain with my point of calling the experiment leaving out a digit at a particular position digit exclusion way too strong. Of course an embedding cannot be learned without the digit entirely, but there are many works that have dealt with OOV problems. I strongly encourage the authors to write down more clearly what this experiment actually does. Because of these remaining (arguably maybe cosmetic or minor) points, I will raise to a 6.

---

### Official Review · Reviewer_sewW · 2023-10-31

**Soundness:** 2 fair
**Presentation:** 4 excellent
**Contribution:** 3 good
**Rating:** 5
**Confidence:** 4

**Summary:**

This paper focuses on an important problem of trying to understand how emergent capabilities like being able to solve arithmetic tasks arise while training transformer-based language models. Since it is difficult to decouple the various factors like compute, data, and model size to understand emergent capabilities, this paper conducts extensive experiments focusing on arithmetic tasks like addition, subtraction, multiplication, and unary operations like sine/sqrt on small models like GPT-2/nanoGPT. The paper presents various key findings:

1. Training data format and sampling is important to make the decoder models learn arithmetic tasks properly. Models can learn addition in reverse better compared to the standard addition (with most significant digit being predicted first auto-regressively). The training data should have a good distribution over the number of carry operations (for addition/subtraction) as random sampling leads to an imbalance and reduced performance.
2. Chain-of-thought style prompting / scratchpad based training leads to improved performance. Although there's a tradeoff, as the number of tokens to be generated to get the final answer have significantly increased.
3. Mixing text data with arithmetic task data during training does not lead to performance degradation, just that the number of samples required to achieve emergence for addition becomes higher.
4. Generalization beyond the digit lengths present during training is hard. If the model is not trained on a specific digit length, the model is not able to perform well even when it is trained on various digit lengths excluding this specific length.

**Strengths:**

I really liked reading this paper and going over the various results and ablations presented in the paper. I believe the paper has various strengths as listed below:

 - The paper tackles an important problem of understanding emergence in language models wrt arithmetic operations. Data with arithmetic operations is not inherently present in the pre-training corpus, but still the models can do some rudimentary level of arithmetic tasks with few-shot prompting. The paper has some good ideas on adding arithmetic task data to the pre-training, or adapting pre-trained models with supervised fine-tuning on arithmetic data with some caveats (adding spaces to ensure better tokenization).
- One of the important contributions of the paper is the structured data sampling for making models learn addition - ensuring good distribution over $n - 1$ digit operations for $n$ digit learning, and also ensuring equitable samples of the number of carry operations.
- The section on the equivalence of learning addition and low-rank matrix completion is insightful, and how transformers have generalization capabilities beyond matrix completion.
- The paper is well written and builds the story coherently, with lots and lots of ablation studies in the appendix and additional results.

**Weaknesses:**

- I was curious about the \\$ symbol bit present here and there in the paper, but it became clear after reading appendix B. In my opinion, the baseline used for making the strong claims of models being able to handle reverse addition better in the paper is wrong. The authors should have either done the reverse methodology without \\$ or used \\$ for the baseline too. This is a bit concerning as data formatting techniques highlighted in the paper are touted as one of the important contributions, and this finding makes that invalid.
- Expanding on the previous point, Figure 9 specifically highlights that even plain addition is able to reach almost 100% test accuracy with the addition of the \\$ symbol, and reverse without \\$ is almost at 90% (only 2/3% difference with the baseline).
- I believe some parts of the original manuscript is appendix material and some important bits in the appendix should be moved to the main paper. Specifically, the lemmas in section 4 seem a bit irrelevant given that difference of \\$ symbol in the baseline and reverse addition. Appendix section B.1 should be present in the main paper.
- Not a weakness since the paper is fairly well written, but here are some typos in the paper:
    1. I believe a latex shortcut is used to represent $A_{3}A_{2}A_{1} + B_{3}B_{2}B_{1} = C_{3}C_{2}C_{1}$ in the paper, because all occurrences of $B_{3}B_{2}B_{1}$ are represented as $B_{3}B_{1}B_{1}$.
    2. Figure 1, in the detailed scratchpad solution, I think a carry ($C = 1$) has been missed after $[1, 2] + [3, 6], A = [5]$.
    3. Figure 4b, it should be just Number of tokens on the x axis instead of Number of unique tokens?
    4. Figure 8: Larger model scale instead of sacale.
    5. No space between $\textit{text}$ and $\textit{and}$ in the last line on page 8.

**Questions:**

I have asked most of my questions in the weakness section, but here are a few more:

- Figure 1, $A$ is built incrementally, like $A = [5]$, then $A = [9,5]$, and so on. What is the effect of starting with $A = []$ (empty set) in the prompt, as the first line just starts with a carry.
- There's a slight confusion on the experiment setting in Figure 2. Is it plain addition with \\$ symbol and not the actual baseline? Also it seems that even with structured sampling, the overall performance on 2-digit addition is low when $n = 3$.
- Do you have an equivalent diagram for Figure 9 for GPT-2 and GPT-3?

---

> ### Author Response · Authors · 2023-11-21
> **Response to Reviewer sewW (1/1)**
>
> Thank you for your feedback on our paper. We appreciate your comments and would like to address your concerns.
>
> > `$-symbol` I was curious about the `$` symbol bit present here and there in the paper, but it became clear after reading appendix B. In my opinion, the baseline used for making the strong claims of models being able to handle reverse addition better in the paper is wrong. The authors should have either done the reverse methodology without `$` or used `$` for the baseline too. This is a bit concerning as data formatting techniques highlighted in the paper are touted as one of the important contributions, and this finding makes that invalid.
>
> We thank the reviewer for their careful reading of the paper and acknowledge that this is somewhat confusing. In order to simplify the message of the paper, we decided to subsume the `$` delimiter into the reverse data format. However, as the reviewer points out, even in Appendix B, it is clear that reverse does improve the performance significantly both with and without the `$` delimiter. We will modify the wording in the paper to make this more clear.
>
> > `Figure 9. plain reaching almost 100%` Expanding on the previous point, Figure 9 specifically highlights that even plain addition is able to reach almost 100% test accuracy with the addition of the \$ symbol, and reverse without `$` is almost at 90% (only 2/3% difference with the baseline).
>
> You are correct in observing that plain addition can approach 100% test accuracy with an adequate number of training examples. We have updated our manuscript to reflect that the plain format does not plateau at 85% as previously stated. We also found that introducing a `\n` character as a prefix while prompting the model significantly enhances performance for models trained without the `$` delimiter. This result is presented in [Figure 9](https://imgur.com/a/SKCkdPK), which can be viewed here. These results indicate that the inclusion of a delimiter during testing is a critical factor for model accuracy. We will add this result in our revision, and also include experiments on the plain format with this modification.
>
> > `Organization of paper` I believe some parts of the original manuscript is appendix material and some important bits in the appendix should be moved to the main paper. Specifically, the lemmas in section 4 seem a bit irrelevant given that difference of `$` symbol in the baseline and reverse addition. Appendix section B.1 should be present in the main paper.
>
> We maintain that the benefit of reversing the order of digits is significant even though it is reduced with the removal the `$` delimiter. However, we will include the contents of B.1 into the main paper as it serves to clarify the setting more clearly.
>
> > `Typos`
>
> Thank you for pointing these out. We have fixed it in the revised manuscript.
>
> > `scratchpad starting with A=[]` Figure 1, $A$ is built incrementally, like $A=[5]$, then $A=[9,5]$, and so on. What is the effect of starting with $A=[]$ (empty set) in the prompt, as the first line just starts with a carry.
>
> We apologize for the confusion. The detailed scratchpad example in Figure 1. initially contained a typo in the first intermediate step. All the detailed scratchpad experiments were actually conducted on training examples starting with `A=[]`. We have fixed Figure 1 with the correct example.
>
>
> > `Figure 2. Setting` There's a slight confusion on the experiment setting in Figure 2. Is it plain addition with `$` symbol and not the actual baseline? Also it seems that even with structured sampling, the overall performance on 2-digit addition is low when $n=3$.
>
> Yes that is indeed the setting we chose. We chose to use the `$`-wrapped plain as a baseline since we wanted to use a method that reaches nearly 100% accuracy within a reasonable number of samples. The reviewer is indeed correct that 2-digit addition is not as good as one would hope. The balanced digit sampling strategy involves selecting 100 instances of 1-digit, 900 instances of 2-digit (constituting 9% of the total 10,000 instances for this category), and 9,000 instances of 3-digit numbers. While our strategy performs significantly better than random sampling, it perhaps needs to be tweaked to reach 100% accuracy within each category.
>
> > `GPT-2, GPT-3` Do you have an equivalent diagram for Figure 9 for GPT-2 and GPT-3?
>
> We have added an additional result on GPT-2 in Figure 9 of the revised version, which shows a similar result as the NanoGPT experiments ([Updated Figure 9](https://imgur.com/BFNGBYO)).

---

> ### Comment · Reviewer_sewW · 2023-11-22
> **Response to Rebuttal**
>
> I thank the authors for their response to the review and appreciate the updates to the main manuscript. I still believe the authors have not addressed the mismatch in the baseline clearly, the plain should be with \\$ symbol too or reverse should not contain the $ symbol. This more fair baseline has very little difference between the plain and reverse. Due to this reason, I maintain my original rating of 5.
>
> Also, there's a typo in Figure 9 (without instead of wihout).

---

> > ### Author Response · Authors · 2023-11-22
> > **Addressing Baseline Mismatch and Improving the Manuscript**
> >
> > Thank you for your continued effort in providing thorough feedback for our manuscript. We appreciate the time you spent reviewing our work and apologize if some aspects of the presentation need calibration. We are more than eager to make the necessary adjustments to address your concerns.
> >
> > Regarding the mismatch in the baseline, we understand your concern and appreciate your attention to detail. We are currently working hard to modify the text and figures to better reflect the results. To provide a consistent comparison, we propose to change all results to plain format with the $ delimiter for all experiments. We are happy to run additional experiments and make the necessary changes in the manuscript to ensure a fair comparison between plain and reverse formats.
> >
> > While we still observe a difference between reverse and plain formats (see results with new baseline with $ symbol [here](https://anonymous.4open.science/r/nanoGPT-25D2/plots/figs/Figure1-withDollar.pdf)), we are currently running experiments to show that this difference increases with the number of digits for using the same $ delimiter. We hope that these additional results will help clarify the findings and address your concerns.
> >
> > We have invested considerable effort in our work, and we believe that it is possible to address your comments and improve the manuscript in this review cycle. We kindly ask for more clarity on any remaining concerns or suggestions for improvement, so we can address them promptly. We hope that our honest effort in addressing your feedback and our dedication to improving the manuscript will help you reconsider your rating and see the potential of our work.

---

### Official Review · Reviewer_ofJx · 2023-10-31

**Soundness:** 3 good
**Presentation:** 1 poor
**Contribution:** 3 good
**Rating:** 6
**Confidence:** 4

**Summary:**

The authors present an analysis of the performance of a small decoder-only transformer (NanoGPT), trained from scratch on arithmetic tasks (2 or 3-digit positive integer addition, mainly). They show that 3 digit addition can be learned to very high accuracy, from less than 5000 training examples, if the output digits are represented in "reverse order" (i.e. representing 256 as the sequence [6, 5, 2]), and less than 2000 if the model is provided chain of thought information (intermediary steps in the calculation) during training.

On two-digit addition, the authors observe that learning addition from a small training set amounts to completion of a low rank matrix (the addition table). They show that NanoGPT has the same data efficiency as classical completion algorithms, both needing about 2000 examples to "fill the table", but that NanoGPT overcomes one of the main limitations of classical algorithms: the need to have at least one example on every line and column.

The authors also present extensions of their approach to longer operands, and other mathematical operations (on integers and decimals). Finally, they present experiments with larger models (pre-trained or not), showing that whereas larger models, like GPT-3, achieve better performance with few-shot learning, their observations on the role of reversed digits in the output, and chain-of-thought prompting, remain valid.

**Strengths:**

The paper demonstrates that a basic arithmetic operation, like integer addition, can be learned by small transformers from a limited number of examples. They also show that techniques like chain-of-thought prompting, introduced as a method for finetuning pre-trained models on arithmetic tasks, also benefit small transformers, trained from scratch. This is an important result.

**Weaknesses:**

The submission is a compressed version of a very long paper. As a result, some of the claims in the introduction, e.g. those relative to length generalization and compositionality, are not discussed in the main paper. The results on other operators, and the impact of pre-training on text, are very hard to assess, because the paper provides almost no description of the experimental setting. Finally, some of the figures (e.g. fig. 5) are compressed beyond legibility. This makes the main paper difficult to read, especially starting with section 7 (unless one is ready to read 25 additional pages).

I would recommend that the authors either submit the full version of their paper to a journal, or limit it to their results on addition, which are significant enough, and deserve a longer discussion.

**Questions:**

* To which extent is your finding on reversing digit order in the output specific to the decoder-only architecture you use? I believe output order would be irrelevant in an encoder-only setting (possible here because the output sequence is guaranteed to be shorter than the input sequence), what about an encoder-decoder architecture?
* In the paragraph on balancing digit, you say : "For instance, in the case of 3-digit addition, random sampling results in a meager 0.01% probability of selecting a 1-digit number." The probability of selecting a 1-digit operand should be 1%, right?
* What are the performances of the model when operands have different lengths (e.g. 2 + 312, or 546 + 7)?
* Does the model learn the properties of addition, e.g. commutativity? This could be done by testing whether model predictions for A+B and B+A are the same, even early during training.
* Can those results generalize to decimal numbers (e.g. 1.21+13.12)?
* Can those results generalize to signed numbers, by adding a sign token to all three integers?
* There is a tension between your results from section 3 and 4, which suggest that models are learning the same algorithm as us (quickly memorizing the 10x10 table, then adding successive digits and propagating carries), and the results from section 5, which frame addition as a memorization+interpolation problem (learning to interpolate a large but low rank matrix). Is there a way to decide what algorithm the model is actually learning?
 * Is there a chain-of-thought approach suitable to low-rank matrix completion? If so, it would greatly improve its data efficiency, and perhaps allow it to scale to larger operands.

---

> ### Author Response · Authors · 2023-11-21
> **Response to Reviewer ofJx (1/3)**
>
> Thank you for your appreciative feedback on our paper. Please find the responses to your concerns inline below.
>
> > `Paper being too compressed` The submission is a compressed version of a very long paper. As a result, some of the claims in the introduction, e.g. those relative to length generalization and compositionality, are not discussed in the main paper. The results on other operators, and the impact of pre-training on text, are very hard to assess because the paper provides almost no description of the experimental setting. Finally, some of the figures (e.g. fig. 5) are compressed beyond legibility. This makes the main paper difficult to read, especially starting with section 7 (unless one is ready to read 25 additional pages). I would recommend that the authors either submit the full version of their paper to a journal, or limit it to their results on addition, which are significant enough, and deserve a longer discussion.
>
> We appreciate the reviewer's feedback and are encouraged by the recognition of the significance of our results on addition. We understand that the scope of the paper is extensive, and some content might be compressed, making it difficult to read. In light of the reviewer's suggestion, we would be happy to reduce the scope of this submission to focus solely on addition, moving relevant results from the appendix to the main text to provide a clearer and more comprehensive discussion.
>
> We believe that our work is timely and relevant to the current research landscape, and we believe that presenting our findings at this year's ICLR would be beneficial to the community. We will revise Sections 7 and 8 to limit their reliance on the appendix and improve the overall legibility and coherence of our paper.
>
> > `different architectures` To which extent is your finding on reversing digit order in the output specific to the decoder-only architecture you use? I believe output order would be irrelevant in an encoder-only setting (possible here because the output sequence is guaranteed to be shorter than the input sequence), what about an encoder-decoder architecture?
>
> We intentionally focus on decoder-only architectures in order to focus on understanding the emergent properties in LLMs which are typically decoder-only transformers. For encoder-only models, is the author referring to BERT style architectures? If so, since the model is producing all the digits "simultaneously", the reviewer is right in that the reverse will likely not make any difference. However, in the case of encoder-decoder architectures, since the model is still producing digits from left-to-right, reversing the digit order will give it the opportunity to learn a simpler function (see Lemma 1 and 2). We are currently running experiments on both these architectures and hope to report the results by the end of the rebuttal period.
>
> > In the paragraph on balancing digit, you say : "For instance, in the case of 3-digit addition, random sampling results in a meager 0.01% probability of selecting a 1-digit number." The probability of selecting a 1-digit operand should be 1%, right?
>
> It is $0.01\%$ since there are only $100$ samples containing 1-digit numbers among the $10^6$ 3-digit addition samples. The reviewer is right in that the probability of selecting a single 1-digit operand is $1\%$, but the probability that either operand is 1-digit is significantly lower.
>
> > `different lengths` What are the performances of the model when operands have different lengths (e.g. 2 + 312, or 546 + 7)?
>
> We report the following result on different digit length combinations. Each row and column represent the number of digits in the first and second operand, respectively.
>
> For plain, performance is lower for lower-digit combinations. It is also interesting to note that the performance is not necessarily symmetric in the lengths of the operands. We conjecture that this is because despite our balanced sampling approach, the number of samples of $(1, 3)$ digit or $(2,1)$ digit sums are relatively few in our training data.
>
> | Plain   | 1-digit | 2-digit | 3-digit |
> | ----   | ----    | ----    | ----    |
> | 1-digit| 72.8%   | 42.2%   | 77.0%   |
> | 2-digit| 27.8%   | 60.0%   | 91.4%   |
> | 3-digit| 38.6%   | 61.2%   | 97.0%   |
>
> For reverse, performance is fairly consistent for different digit combinations since the model learns addition almost perfectly.
>
> | Reverse| 1-digit | 2-digit | 3-digit |
> | ----   | ----    | ----    | ----    |
> | 1-digit| 100.0%  | 99.8%   | 99.0%   |
> | 2-digit| 96.4%   | 100.0%  | 100.0%  |
> | 3-digit| 94.0%   | 99.4%   | 100.0%  |

---

> ### Author Response · Authors · 2023-11-21
> **Response to Reviewer ofJx (2/3)**
>
> > `commutativity` Does the model learn the properties of addition, e.g. commutativity? This could be done by testing whether model predictions for A+B and B+A are the same, even early during training.
>
> This is an excellent question! We find that while the model doesn't learn commutativity perfectly, it definitely learns it to a large extent, especially after being fully trained. We use NanoGPT fully trained on the plain data format and check if $(A+B) = (B+A)$. Out of 9900 test examples, only 8799 samples commute. While most of these samples are trivially commutative (since the model gets the answer correct), we find that out of the 164 samples where the model gets both $(A+B)$ and $(B+A)$ wrong, it still commutes on 99 of them! We summarize these findings in the table below.
>
> |        | P(A+B=B+A, both are correct answer) | P(A+B and B+A are both incorrect) | P(A+B=B+A \| both are incorrect) |
> | ----   | ----    | ----      | --- |
> |  | 0.878 (8700 samples)   | 0.016 (164 samples)  | 0.603 (99/164 samples) |
>
> Therefore, the model learns to commute 60.3\% of the time even when it gets the answers wrong!
>
> > `decimal numbers` Can those results generalize to decimal numbers (e.g. 1.21+13.12)?
>
> We observe that our results do generalize to addition of two decimal numbers (See [Decimal Results](https://imgur.com/a/uGwqxpl)). We convert each integer number (up to 3-digit) to decimal numbers with 1 digit + 2 floating points (ex. 1-digit number `a -> 0.0a`, 2-digit number `ab -> 0.ab`, 3-digit number `abc -> a.bc`) to train the model on decimal numbers. The obtained results align with our findings on integer addition, where addition is learned more efficiently with the reversed output. Somewhat surprisingly, the overall performance is better than 3-digit integer addition. However, this is likely because the decimal number formatting behaves like zero-padding since we ensure fixed digit length (Appendix B.1).
>
>
> > `signed numbers` Can those results generalize to signed numbers, by adding a sign token to all three integers?
>
> We suspect that our results can be generalized to signed numbers as well, which is similar to training the model on both addition and subtraction. Our preliminary experiment on training the model with all signed combinations of two 3-digit numbers show the following [results](https://imgur.com/a/2mTeH9m). We find that training with signed numbers is a more difficult task than training on addition or subtraction alone. Interestingly, the accuracy of $(+,+)$ and $(-,-)$ sign combinations is higher than $(+,-)$ or $(-,+)$ sign combination, suggesting that (as expected) learning addition is easier than learning subtraction.
>
> > `learned algorithm` There is a tension between your results from sections 3 and 4, which suggest that models are learning the same algorithm as us (quickly memorizing the 10x10 table, then adding successive digits and propagating carries), and the results from section 5, which frame addition as a memorization+interpolation problem (learning to interpolate a large but low-rank matrix). Is there a way to decide what algorithm the model is actually learning?
>
> This is a keen observation and an excellent point! In fact, most of our work is aimed at trying to glean some insights into what algorithm the model is actually learning. In fact, the second part of Section 5 suggests that the generalization properties of NanoGPT when certain numbers are excluded from the training data, surpass Matrix Completion and therefore are not simply interpolating the low-rank addition matrix. While the results from Section 3 and 4 do indicate that the model learns our addition algorithm when presented with reversed outputs, the limited length generalization indicates that it does not "truly" learn this algorithm. In the end, while we do not precisely characterize the algorithm learned by the model, we do characterize its tendencies and behaviors, particularly in regard to its sensitivity to the input data format.

---

> > ### Author Response · Authors · 2023-11-22
> > **Response to Reviewer ofJx (3/3)**
> >
> > > `chain-of-thought for larger operands` Is there a chain-of-thought approach suitable to low-rank matrix completion? If so, it would greatly improve its data efficiency, and perhaps allow it to scale to larger operands.
> >
> > Perhaps the reviewer can clarify what they mean by chain-of-thought approach suitable to low-rank matrix completion? For larger operands, we find that the simplified scratchpad is a more token efficient version that is more scalable to larger operands. Furthermore, since the sample complexity improves significantly with chain-of-thought data, it remains efficient to scale to larger operands (see Appendix C. where we train on samples up to 10 digits in length)
> >
> > One possible option for designing a chain-of-thought approach suitable to low-rank matrix completion is to decompose the two operands to their 1s, 10s, 100s and so on. For example $abc + def = a00 + b0 + c + d00 + e0 + f = ghi$. We tried retraining the model with data in this format and the [results](https://imgur.com/a/rBaza0R) show that this works extremely well. While it is still slightly worse than the (longer) simple scratchpad format, it far outperforms the plain format. The high performance of this format suggests that the tensor completion argument presented in our [Response to Reviewer 9wwh (2/3)](https://openreview.net/forum?id=dsUB4bst9S&noteId=5SmGFcc0O8) might be representative of the way the model learns addition.

---

> > > ### Comment · Reviewer_ofJx · 2023-11-22
> > >
> > > Thank you very much for your responses, and for making the main part of paper easier to read.
> > >
> > > I am happy to raise my note from 3 to 6.

---

> > > > ### Author Response · Authors · 2023-11-22
> > > > **Thank you for your feedback, and happy to address more concerns**
> > > >
> > > > Thank you for all your feedback and time you spent reading and providing feedback on our work, and also for updating your assessment. We'd be happy to address any further concerns you may have. Please do let us know.

---

### Official Review · Reviewer_UmA7 · 2023-10-31

**Soundness:** 3 good
**Presentation:** 3 good
**Contribution:** 2 fair
**Rating:** 5
**Confidence:** 3

**Summary:**

In this study, the researchers explore the ability of small transformers, to grasp basic arithmetic tasks using the next-token prediction objective. The research reveals that simple alterations in the format of the training data, such as inverting the results or incorporating step-by-step breakdowns, can markedly enhance model accuracy. Furthermore, the paper delves into the intricate relationship between arithmetic and text data in training, as well as the length generalization challenges encountered by these models. This research underscores the significance of refined and targeted data, keeping in mind the unique traits of the next-token prediction objective, in swiftly fostering arithmetic capabilities.

**Strengths:**

Well-written, technically sound paper

**Weaknesses:**

The study's main shortcoming in terms of novelty stems from its reliance on previously established methods and datasets, particularly the use of reasoning-augmented data, which has been prevalent in enhancing model performance. The authors explicitly acknowledge that their work doesn't break new ground in terms of the types of training data or in achieving peak performance with minimal model parameters. However, the research sets itself apart through its meticulous ablation studies and in-depth exploration of various sampling techniques, training data formats, data source mixing ratios, and model scales. Additionally, while they provide certain novel theoretical explanations for observed phenomena, their primary emphasis on arithmetic isn't for its intrinsic importance but as an easily testable emergent skill to better understand emergent phenomena in models.

**Questions:**

How do the authors envision scaling their methodology beyond GPT-3? Additionally, do they believe their approach is compatible with subquadratic transformer variants?

---

> ### Author Response · Authors · 2023-11-21
> **Response to Reviewer UmA7 (1/1)**
>
> We appreciate the reviewer's thoughtful feedback on our paper. We are glad to hear that you find our paper is well-written, and is technically sound. We respond to your questions inline below.
>
> > `Shortcoming in terms of Novelty` The study's main shortcoming in terms of novelty stems from its reliance on previously established methods and datasets, particularly the use of reasoning-augmented data, which has been prevalent in enhancing model performance. The authors explicitly acknowledge that their work doesn't break new ground in terms of the types of training data or in achieving peak performance with minimal model parameters. However, the research sets itself apart through its meticulous ablation studies and in-depth exploration of various sampling techniques, training data formats, data source mixing ratios, and model scales. Additionally, while they provide certain novel theoretical explanations for observed phenomena, their primary emphasis on arithmetic isn't for its intrinsic importance but as an easily testable emergent skill to better understand emergent phenomena in models.
>
> We acknowledge the reviewer's concern regarding the novelty of our work. While we agree that our study does not introduce new datasets or techniques, we would like to emphasize that the novelty of our research lies in the extensive ablation studies and the thorough exploration of various factors influencing model performance in a well-specified setting. Our work's contribution is the comprehensive understanding gained through dissecting the emergence of arithmetic in small transformer models, particularly highlighting the importance of data sampling and data formatting, including seemingly simple methods like reversing the outputs. Our findings contribute to a more in-depth study of how transformers develop arithmetic skills, and have a broader implications on synthetic training data generation for LLM training, which we believe is valuable for the research community. This work is particularly timely given the current state of research in this area and will hopefully encourage further exploration and development of more sophisticated techniques to better understand emergent phenomena in models.
>
>
> > `Q1. scaling methodology` How do the authors envision scaling their methodology beyond GPT-3?
>
> We would like to emphasize that since our approach primarily focuses on data sampling and formatting techniques, it is directly applicable to models of different scales. Our experiments on finetuning GPT-3 (Table 2) show that our findings extend from NanoGPT to GPT-2 and even GPT-3. We see no reason why it should not extend beyond that as well. We are currently running experiments on finetuning GPT-4 and hope to include the results by the end of the rebuttal period.
>
> > `Q2. compatibility with subquadratic variants` Additionally, do they believe their approach is compatible with subquadratic transformer variants?
>
> We have not yet explored subquadratic Transformer variants such as sparse attention, or linformer.  Since our analysis is primarily focussed on the pre-training data, we expect that it should extend to those variants as well. We are currently running our experiments for these variants and hope to include the results by the end of the rebuttal. We will surely include it in the final manuscript.

---

### Official Review · Reviewer_9wwh · 2023-11-02

**Soundness:** 3 good
**Presentation:** 3 good
**Contribution:** 2 fair
**Rating:** 8
**Confidence:** 4

**Summary:**

The paper investigates the performance of Transformers (small scale) on a set of arithmetic tasks when trained from scratch with next token prediction objective. The study is motivated by the emergent ability of LLMs in solving arithmetic problems even though they are not directly trained on these tasks. The main task is multi digit arithmetic (-,+,*), most of the paper is focused on setting where the task is only multi digit addition.

In the paper,
1. They study the impact of input representation (reversing the output digits helps the models learn the task faster).
   - In the setting where they reverse the output digits, the model learns the task but there is a sharp phase transition as they increase the number of training samples. They try to explain this by the argument that learning a map on `n` digits from random samples is equivalent to completing a low-rank matrix (but they mention that this doesn't explain the generalisation behaviour of the models).
  - For the multiplication operation, reversing the output does not have a positive effect.
2. They study the impact of data distribution (balanced vs non-balanced)
3. They show that using chain of thought during training helps (the more detailed the better).
  - In this context, they compare the models in terms of sample efficiency and token efficiency, and show that models trained with CoT are more sample efficient but in total they require more number of tokens.
  - They show that a detailed scratch-pad doesn't help with operations like sine and square-root.
4. They show that the techniques of reversing and using CoT during training, stay as sample efficient when the complexity of the task grows in terms of the number of digits, where is training the models on the plain format of the task becomes harder (requires more samples) as the number of digits increases.
5. They investigate the effect of training the model on a mixture of text and arithmetic data.
6. They investigate the effect of models size (comparing nano-gpt and gpt-2, and pre-trained/fine-tuned GPT-3).

**Strengths:**

- Lot's of interesting analysis.
- Maybe a difference with some of the previous work on this topic is that here the objective, similar to language models, is the next token prediction, as opposed to modelling the task as a classification task. This is basically training language models on arithmetic data. The paper aims to reveal the factors that lead to emergence of arithmetic capabilities in a minimal setting.

**Weaknesses:**

While it is very interesting to understand if, how and under which settings language models learn simple arithmatics, It's not clear to me how the findings in the paper can be generalised.
- For example, even in case of the ability of models to learn arithmetics, as mentioned in the paper, these results are based on using character level tokeniser which simplifies things a lot when it comes to such tasks and is potentially one of the biggest challenges for LLMs to perform well on these tasks. (A parallel work that looks into this: https://openreview.net/forum?id=OinvjdvPjp).
- Methods like reversing the output seem a bit tacky. I agree it is interesting to see that these types of modification to the input/output impacts the results, making it easier for the model to learn the task, but I am not sure if they can have any value beyond analysis/understanding purposes.
- While some of the experiments presented in the paper are conceptually very interesting they seem to be exploring the space a bit sparsely which makes it harder to make any firm conclusions.

**Questions:**

1. When chain of thought is applied during training, does the model also generate the chain of thought during inference? Is there any correlation between the validity of the chain of thought (during inference) and the correctness of the answer?
2. Could the reason for better performance of the detailed chain of thought model simply be its length?
3. In Table 1, could you report a confidence interval? In the caption, you say in some cases it improves (when some numbers are excluded from the training data)? If the difference here is significant (it actually is a an improvement), what is the intuitive explanation? You mention a regularisation effect, could you elaborate on that?
4. In Figure 4, are the models with scratch pad also with reversed output?
5. The paper argues that the sudden jump in accuracy can be explained if addition is formulated as 2-rank matrix completion, but the generalization behaviour of the model can not be explained by this. Could you elaborate how this explanation holds even though it is not fully consistent with the behaviour of the model?
6. When comparing models with different sizes, is the smaller model trained for longer?
7. Is there a reason why the number of digits for experiments in Figure 6 are different for different operations?
8. What is the number of digits for experiments presented in Figures 7 and 8 and Table 2.
9. Do you have any experiments where the different operations are not split into different tasks (where the task contains the mixture of operation)?
10. Is there a failure point as you increase the number of digits?

---

> ### Author Response · Authors · 2023-11-21
> **Response to Reviewer 9wwh (1/3)**
>
> We thank the reviewer for their detailed feedback and for acknowledging that our experiments and analyses are extensive and interesting. Please find our responses to each of the concerns you have raised below.
>
> > `Generalizability` These results are based on using character level tokenizer which simplifies things a lot when it comes to such tasks and is potentially one of the biggest challenges for LLMs to perform well on these tasks.
>
> We agree that the character-level tokenizer simplifies the setting, but the main reason we chose arithmetic, is to find a setting where it is simple enough to ablate all the different factors. Note that for GPT-2, we also ran experiments with BPE tokenization (tiktoken), as shown in [Figure 25](https://imgur.com/xVCWPSl).
> While the results may not completely generalize, our experiments on Chain-of-Thought data and sampling on models of varying scales illustrate that most of our findings remain applicable. We hope that this will encourage more research on systematically ablating individual components.
>
> > `Value beyond analysis/understanding purposes` Methods like reversing the output seem a bit tacky. I agree it is interesting to see that these types of modification to the input/output impacts the results, making it easier for the model to learn the task, but I am not sure if they can have any value beyond analysis/understanding purposes.
>
> We appreciate your perspective on the value of our work and would like to emphasize that our primary objective is to analyze and understand how these models learn arithmetic skills and, if possible, determine more efficient ways to elicit such learning. While reversing may seem somewhat tacky, the importance of coming up with data formatting techniques to accelerate the learning process is more important than ever. We believe that work along this line is especially important now with growing model scales and growing scarcity of tokens. We hope that our work encourages more research along the lines of Gunasekar et al. "Textbooks are all you need" and leads to creation of small, well curated datasets for fast LLM training.
>
>
> > `Sparse exploration space` While some of the experiments presented in the paper are conceptually very interesting they seem to be exploring the space a bit sparsely which makes it harder to make any firm conclusions.
>
> We appreciate the reviewer's observation regarding the sparse exploration space in our study. We acknowledge that it might be challenging to draw firm conclusions, but we argue that this is primarily due to the inherent complexity of studying LLMs. Given the scale and the multitude of variables involved, isolating individual aspects of the problem is a daunting task. This is the reason we focused on a simple problem like arithmetic and conducted extensive ablations to gain insights into data formatting, sampling, and their impact on teaching skills to small transformer models. However, we would like to point out that we have results on GPT-2 and various scales of GPT-3 models in Sections 8. and Appendix F. We are currently running experiments on finetuning GPT-4 and hope to post those by the end of the rebuttal period.
>
> In our study, we aimed to examine parameters that are commonly used in practice, such as training data formatting, data scale, and few-shot prompting. We understand the reviewer's concern about the scope of our experiments and would be open to reducing it if deemed valuable. However, we believe that each experiment in our study offers useful insights, which we have attempted to summarize in each subsection of the paper. Our work serves as a foundation for future research in understanding the learning process of language models and optimizing their performance using data formatting and other techniques.
>
> > `Q1. CoT generated data` When chain of thought is applied during training, does the model also generate the chain of thought during inference? Is there any correlation between the validity of the chain of thought (during inference) and the correctness of the answer?
>
> When we train the model with CoT data from scratch, it does generate output in a chain-of-thought style during inference. As expected, there is indeed a strong correlation between the validity of the chain of thought (during inference) and the correctness of the answer. We evaluate the `scratchpad match score` to be the fraction of intermediate computations that are correct and plot it with the final accuracy. We observe a strong correlation between the two, as can be seen [here](https://imgur.com/a/WLdu5Ln). We observe that errors in intermediate steps propagate to subsequent steps, leading to incorrect answers. However, for some of the less accurate models (say NanoGPT trained on 250 samples), the model chances upon the correct answer even after getting some of the intermediate computations wrong.

---

> ### Author Response · Authors · 2023-11-21
> **Response to Reviewer 9wwh (2/3)**
>
> > `Q2. Length of CoT` Could the reason for better performance of the detailed chain of thought model simply be its length?
>
> This is a great question! We have several results that demonstrate that the improved performance of the detailed scratchpad format is attributed to the intermediate steps rather than the length of the data format. For instance, in Section B.3. in the Appendix, we highlight the importance of designing intermediate steps by comparing two different versions of the detailed scratchpad format for subtraction. In Section B.4, we examine the effect of recording correct digit-wise sum `(A)` and carry `(C)` information versus randomizing `A` and `C` in the simplified scratchpad format, showing that addition is best learned with accurate `A` and `C` values.
>
> Furthermore, in the revised version of Section B.3., we have included an additional experiment in which we replace the intermediate steps of the detailed scratchpad format with random tokens, keeping the overall data format length constant and find that the performance is significantly worse than using the correct intermediate steps. This indicates that the intermediate steps play a crucial role in the model's performance, rather than the mere length of the data format.
>
> > `Q3. Table 1.` In Table 1, could you report a confidence interval? In the caption, you say in some cases it improves (when some numbers are excluded from the training data)? If the difference here is significant (it actually is a an improvement), what is the intuitive explanation? You mention a regularisation effect, could you elaborate on that?
>
> By "some cases", we refer to the columns where we exclude 100 and 500 numbers for plain and all 3 cases for reverse, since the overall accuracy is higher than in the case when we exclude no numbers. We refer to this as a regularization effect since it can be thought of as analogous to data augmentation with cropped images used while training vision models. However, as the reviewer points out, this is somewhat handwavy.
>
> A more reasonable explanation of this is a generalization of the matrix completion argument that we discuss in Section 5. Note that learning how to add two-digit numbers could be viewed as completing a fourth-order, rank-4 tensor (rank-4 in the CP sense). $ab + cd = 10a + b + 10c + d$, which can then be represented as the sum of fourth order rank-1 tensors (rank-1 in the CP sense): $10* T_1 + T_2 + 10*T_3 + T_4$ where $T_1$ is $\mathbf{N}\circ \mathbf{1} \circ \mathbf{1} \circ \mathbf{1}$, $T_2$ is $\mathbf{1}\circ \mathbf{N}\circ \mathbf{1}\circ \mathbf{1}$ and so on. Note that $\mathbf{N} \in \mathbb{R}^{10}$ here denotes the vector $[0, 1, 2, \dots, 9]$ and $\mathbf{1} \in \mathbb{R}^{10}$ denotes the vector of 1s. While this argument is still informal, it now shows that the sample complexity of learning the "addition-map" is $O(\\#digits)$ rather than $O(10^{\\#digits})$. It also reveals that seeing each digit in each position enough times to learn $T_i$ might be sufficient.
>
> We have updated [Table 1.](https://imgur.com/a/6zHX9xs) and [Table 4.](https://imgur.com/a/6zHX9xs) to contain the confidence interval on the performance of 5 models trained with different seeds.
>
> > `Q4. Figure 4. scrachpad output` In Figure 4, are the models with scratch pad also with reversed output?
>
> No. The models with scratchpad do not have reversed output. Only the reverse format is trained with reversed output.
>
> > `Q5. Matrix Completion` The paper argues that the sudden jump in accuracy can be explained if addition is formulated as 2-rank matrix completion, but the generalization behaviour of the model can not be explained by this. Could you elaborate how this explanation holds even though it is not fully consistent with the behaviour of the model?
>
> We use the phase transition as a hint that matrix completion (MC) could explain part of the behavior of the model learning addition. As the reviewer correctly points out, our discussion in section 5 on generalization shows that NanoGPT is not performing exact matrix completion. We merely remark that the $O(n)$ phase transition of NanoGPT is almost identical to that of the phase transition in MC, and therefore provides some insights into its learning behavior.
>
> > `Q6. Training iterations for smaller model` When comparing models with different sizes, is the smaller model trained for longer?
>
> For plain and reverse formats, NanoGPT and GPT-2 are both trained for the same number of iterations - sufficient for the training loss to converge. The learning rate is chosen from {1e-3, 5e-4, 1e-4, 5e-5} based on validation loss. For the scratchpad format, NanoGPT is trained longer since the number of tokens per sample is higher and it requires more iterations to converge. For the precise details of hyperparameter choices, we refer the reviewer to Table 13 and 14 in the Appendix.

---

> > ### Author Response · Authors · 2023-11-21
> > **Response to Reviewer 9wwh (3/3)**
> >
> > > `Q7. Number of digits for different operations` Is there a reason why the number of digits for experiments in Figure 6 are different for different operations?
> >
> > The choice of 3-digit subtraction was to match that of addition. Multiplication is a harder task and the small NanoGPT model struggles to learn it with plain format, even with only 2-digit multiplication - thus, we settled on 2-digit multiplication. Furthermore, unlike addition and subtraction, the size of the output explodes with longer inputs for multiplication.
> >
> > > `Q8. Figure 7, 8, and Table 2.` What is the number of digits for experiments presented in Figures 7 and 8 and Table 2.
> >
> > The number of digits considered for Figure 7, 8 and Table 2 are all 3-digit additions. All experiments on addition focuses on 3-digit addition, unless otherwise stated.
> >
> > > `Q9. Mixture of Operations` Do you have any experiments where the different operations are not split into different tasks (where the task contains the mixture of operation)?
> >
> > Yes, jointly training on all five different operations are presented in the Appendix D.2. We also ran an experiment where we took the model trained purely on addition and then finetuned it on samples of subtraction to test whether the model is able to transfer some of its learnings. Interestingly, this model converges much faster than a randomly initialized model albeit to a slightly worse solution. We conjecture that the embeddings learned for addition transfer effectively to subtraction. (See [Figure](https://imgur.com/VdxWE3B))
> >
> > > `Q10. More digits` Is there a failure point as you increase the number of digits?
> >
> > We extend our experiments up to 10 digit numbers in Appendix C. and observe no real failure point i.e., we are always able to learn addition within a reasonable number of samples. In fact, the sample efficiency gap between reverse/scratchpad and plain tends to increase with number of digits!

---

> > > ### Comment · Reviewer_9wwh · 2023-11-22
> > >
> > > Thank you very much for the detailed response and clarifications.
> > >
> > > Reviewing the responses and all the other reviews. I believe this paper provides interesting insights on learnability of arithmetic tasks with language models as we scale the number of training samples. While not all the findings in this paper might hold anymore for LLMs (trained on larger scale text data with different properties and limitations). They do show that the task is learnable under specific settings in a somewhat generalisable manner (and hint the factors (like scratch pad or specific ways of formatting data) that have a positive contribution in achieving this).
> > >
> > > - If I am getting this right, the empirical results provided in this paper show that under specific setting we can learn task such as arithmetics in generalisable manner with only training the models on task specific examples (e.g., we. do not need language data).
> > >
> > > - The experiments in Appendix C. are indeed very interesting. It would be more interesting if we could also see the failure point and how the performance compares to LLMs in those cases.
> > >
> > > - Thank you for updating Table 1 and 4 to include the confidence intervals.
> > >
> > > - That would be really nice if you can include the results also for the GPT-4 model in the next version of the paper (as the authors  mentioned it them-selves).
> > >
> > > - Thanks for pointing out to the experiments comparing different formatting of the scratch pad and also including experiments with random CoT. Even though it is intuitive to expect the more informative the chain of thought the better, I find it more convincing to support this with some empirical evidence.
> > >
> > > - I do understand that this paper has not aimed to address the length generalisation challenge, but as the authors point out to previous related work that study mechanisms (scratch-pad + few-shot prompting) that helps LLMs better generalise over length, it would have been interesting to see if these finding hold in case where the models are trained purely on the arithmetics data.
> > >
> > >
> > > P.S., apologies for the typo in my review, I meant the reversing technique seems "hacky" not "tacky" :D

---

> > > > ### Author Response · Authors · 2023-11-22
> > > > **Thank you for the immediate response!**
> > > >
> > > > We thank the reviewer for going through our rebuttal and responding so quickly! We are heartened to see that the reviewer believes that our work is interesting and recommends acceptance. As the reviewer suggests, we will include the GPT-4 results and an additional discussion elaborating on the failure cases of length generalization presented in Appendix C. While we were able to show that these models are able to still learn arithmetic even in the presence of natural language (Appendix E), it is still an interesting question to see if language can help, as shown by prior related work on scratchpad and few-shot prompting. We hope to explore this and length generalization in more detail in the future.
> > > >
> > > > Please let us know if there are any other ways in which we can improve our manuscript!

---

### Author Response · Authors · 2023-11-21
**General Response to AC and Reviewers**

We thank the reviewers for their constructive comments and insightful questions. We appreciate that all reviewers acknowledge that we have "extensive experimental results and interesting analysis" (Reviewer 9wwh, Reviewer sewW, Reviewer twgj). We are also glad to hear that the reviewers find our work "well written and an important result" (Reviewer UmA7 , Reviewer ofJx, Reviewer sewW, and Reviewer twgj). The reviewers are all appreciative of our work and find it interesting, however, we are a little disappointed that the scores are somewhat low (5-5-3-5-3) and hope that by resolving the concerns raised by the reviewers, we can encourage them to increase their scores.

The main concerns/questions raised are:

> **1. Limited Applicability**: Some reviewers mention that the offered findings are on too simple a setting and do not generalize to the case of natural language.

We would like to emphasize that our primary objective is to analyze and understand how these models learn arithmetic skills and, if possible, determine more efficient ways to elicit such learning. We hope that this will help better understand general case of emergent properties in large language models. While we agree that ideas of reversing the input/output may seem specific and not directly applicable to real data, our findings reveal the importance of data formatting in accelerating the learning process. Demonstrating the impact of a simple method like reversing highlights the significance of properly structuring data to enable the language model to learn a compositional function using its individual components more effectively. We also believe that these findings can have significant implications for *synthetic training data generation* for LLMs - a topic that is rapidly growing in importance. We also analyze the token-efficiency of these different formats in Section 6, which is particularly relevant for practical applications. To the best of our knowledge, we are the first work to study these aspects of CoT-style data for arithmetic applications.

We acknowledge that while it might be challenging to draw firm conclusions, this is primarily due to the inherent complexity of studying LLMs. Given the scale and the multitude of variables involved, isolating individual aspects of the problem is a daunting task. This is the reason we focused on a simple problem like arithmetic and conducted extensive ablations to gain insights into data formatting, sampling, and their impact on teaching skills to small transformer models.

**Concluding remarks:** As for the concerns/questions raised, we believe that we have addressed all of them sufficiently and replied in line with each review. We believe that now is a critical and timely moment to share our work with the ML/LLM community, as it not only presents valuable insights that can foster discussions among researchers but also provides novel insights for next-token predictors based on transformer models. We believe that our findings shed light on intricate learning mechanisms behind small transformers and provide explanations on why various data formatting strategies optimize the learning of various arithmetic skills. As the interest in emergent properties and efficient transformers continues to grow, our work carries the potential to inspire the community and stimulate further exploration in this realm. We are appreciative of the opportunity to contribute to this thriving area of research and eagerly anticipate further feedback from reviewers and AC.

---

### Meta-Review · Area_Chair_sVzs · 2023-12-05

**Metareview:**

This work studies the learning behavior of small decoder-only Transformers trained from scratch on arithmetic tasks (primarily several-digit addition). They demonstrate that Transformers are fairly sample-efficient in learning short addition, and they investigate factors which affect the learning speed and robustness (e.g. data format, data distribution, pretraining). For example, they find that formatting the answer in reverse-order significantly improves Transformer learning. Some techniques which were known to help in large-scale LLMs also reproduce in this simple setting, such as Chain-of-Thought.
They consider the toy model of matrix-completion (MC), and find that while MC can reproduce some aspects of Transformer learning (e.g. the sample efficiency), it cannot reproduce all aspects (e.g. OOD robustness to excluding train summands).

The reviews were borderline, with one reviewer strongly supporting the work, and other four reviewers split between weak accept / weak reject.
All reviewers agreed that the paper is interesting and insightful, and the main experiments are thorough. The primary reviewer concerns were:
1. Paper organization & focus: The paper is not well-adapted to the 9-pg ICLR format. It includes many auxiliary discussions, which are not supported/elaborated in the paper body. This distracts from the main focus of the paper (studying addition).
2. Motivation / Practicality: The results may not carry over to large-scale LLMs. This is not necessarily a problem, but the motivation of the paper should be clarified. All reviewers got the impression that this work intended to shed light on LLMs, but the nature of this contribution was unclear. Some of the results (e.g. reversal) do not hold for pretrained LLMs, as reviewers noted.
3. “Polish”: Certain experimental choices are left unexplained (perhaps for lack of space), and sometimes inconsistent in minor ways (e.g. the “$” token, and the details of which factors were controlled when comparing models).

In the rebuttal, the authors partly addressed these points, and several reviewers increased score.
I recommend this paper for acceptance, because I believe these concerns are fixable in the camera-ready, and reviewers agreed the results are insightful and important.
However, I expect the authors to revise the presentation for the camera-ready, in order to fully address all of the reviewers’ concerns.

In addition to the above, here are some of the most significant requested revisions:
* Elaborate further on the experimental setup for all experiments, especially for the LRMC-comparison of Table 1 (all reviewers were confused about this).
* Fix the apparently inconsistent use of “$” in experiments.
* Clarify which parts of the paper are your main claims, vs. Discussion/Speculation. There are several paragraphs (e.g. “Compositional and length generalization” that are not elaborated upon further, and better suited for Discussion).
* Strongly consider removing parts of the paper in order to focus on the main addition experiments (other operations are not discussed in enough detail to be convincing in the body).  The experiments in Section 8 seem somewhat scattered / incomplete in comparison to other sections.
* Elaborate more on Limitations, especially to clarify the relation to production LLMs.
* Avoid imprecise language. E.g. “phase transition” is used to describe plots which do not have any apparent discontinuities (Figure 1). I strongly recommend removing the phrase entirely — it is an over-used phrase in the current literature, which rarely contains meaning. If you are using “phase transition” in a formal sense (e.g. in the LRMC section), clarify that you are using a particular formal definition.

**Justification For Why Not Higher Score:**

Reviewers observed various concerns with the presentation and clarity of the paper (described above).

**Justification For Why Not Lower Score:**

All reviewers agreed the results are insightful and important. The concerns were partially addressed in the rebuttal, and could be fully addressed by the camera-ready. These concerns were ultimately not serious enough to outweigh the paper's strengths.

---

### Decision · Program_Chairs · 2024-01-16

Accept (poster)